# Learning Partitions from Context

**Simon Buchholz**
Department for Empirical Inference
Max Planck Institute for Intelligent Systems
Tübingen AI Center
Tübingen, Germany
sbuchholz@tue.mpg.de

## Abstract

In this paper, we study the problem of learning the structure of a discrete set of $N$ tokens based on their interactions with other tokens. We focus on a setting where the tokens can be partitioned into a small number of classes, and there exists a real-valued function $f$ defined on certain sets of tokens. This function, which captures the interactions between tokens, depends only on the class memberships of its arguments. The goal is to recover the class memberships of all tokens from a finite number of samples of $f$. We begin by analyzing this problem from both complexity-theoretic and information-theoretic viewpoints. We prove that it is NP-complete in general, and for random instances, we show that on the order of $N \ln(N)$ samples, implying very sparse interactions, suffice to identify the partition. We then investigate the conditions under which gradient flow dynamics of token embeddings can reveal the class structure, finding that this is achievable in certain settings when given on the order of $N^2 \ln^2(N)$ samples.

## 1 Introduction

Modern machine learning systems are able to learn extremely complicated relations from data. They often rely on learned embeddings of discrete tokens in a continuous space. This is notably true for Large Language Models (LLMs) [9, 17, 25, 23] which encode their input by converting text into a sequence of discrete tokens that are embedded in a high dimensional embedding space and those embeddings are fed into, e.g., a transformer architecture [28] which allows predicting the next token. But also in the other domains, e.g., in vision, discrete embeddings are frequently used as a component of deep learning architectures [27, 10] as this enables capturing complex concepts that are often discrete.

It was observed that after training, these word embeddings exhibit many interesting structures. The most prominent example probably is the observation that the difference of the Word2Vec embedding vectors of the nouns 'king' and 'queen' approximately equals the difference of the embeddings of 'man' and 'woman' [21, 22]. Similarly, it was found that using word similarity as an inductive bias to structure latent spaces helps downstream performance [4]. Thus, a properly structured latent space seems to be an important ingredient to capture the intricate correlations in complex data.

A proper theoretical understanding of such complex models currently remains an elusive goal. However, there have been various attempts to understand various components of deep learning models. Many works investigated the behavior of feedforward-networks in particular focusing on shallow networks [2, 8] and asymptotic regimes [15, 29]. More recently, several works investigated transformer architectures (often focusing on the one layer case with linearized attention mechanism) [16, 1, 30, 12]. On a technical level [11] is closely related as they study the large depth limit of transformers through the lens of particle systems (however in their case time corresponds to depth, while in our case it corresponds to training time). A feature shared by many of those works is that

little structure is assumed on the input, i.e., fixed token embeddings are assumed and often those are even assumed to be isotropic Gaussian.

Here we instead focus on the dynamics of the token embeddings and study how these can recover structure present in the data. There is no ground truth target for the embeddings for large scale models used in deep learning and their embeddings need to capture a variety of nuanced correlations and relations that are hard to formalize. Therefore, we focus on a simplified problem that nevertheless shares important features with more complex real world settings. One general heuristic is that the embeddings contain information about the similarity of tokens. We focus here on the strongest form of similarity, namely equality. Indeed, our central assumption is that tokens can be clustered in a small number of groups such that tokens within a cluster behave exactly the same, i.e., they interact in the same manner with other tokens.

Then the central questions is what assumptions allow us to recover a hidden structure. The crucial feature of this setting is that we only get information about a token by its interaction with other tokens about which we also only learn through their interaction behavior. Moreover, those interactions are typically sparse, i.e., we observe only a small subset of all possible interactions. Note that this setting resembles observations made in the context of collective behavior where a global structure emerges from local interactions [13, 24, 3]. A related question was investigated in [6] where they study associative memory and also want to identify a hidden structure, however, they learn the class memberships directly through the interaction with a class embedding (and they train the interaction instead of the embedding). In [19] the dynamics of word embeddings and transformer layers was investigated when the data follows a topic model. This work shares the crucial feature that membership of a word in a certain topic is only transmitted through the co-occurrence with other words from the same topic. In contrast to their work, we here do not focus on learning class membership from token frequencies and in fact consider uniform token distribution. Instead, we view this problem as a logical inference problem: Given a set of facts about a set of tokens can the hidden structure of the tokens discovered.

We summarize the main contributions of the paper as follows:

- We introduce a learning problem that shares important features with learning through the interaction behavior that is crucial for LLMs and complex systems.

- We analyze this problem from a complexity-theoretic viewpoint where we show that it is in general hard, and from an information-theoretic viewpoint where we show, roughly, that for $N$ different tokens in the alphabet order $N \ln(N)$ samples are sufficient to identify the latent structure.

- We then carefully investigate the gradient dynamics of token embeddings, finding local recovery of the cluster structure and global recovery for tensor-product functions on the tokens if we have more than $N^2 \ln(N)$ samples for an alphabet with $N$ tokens.

**Notation.** We write $[N] = \{1, \ldots, N\}$ for the set of the first $N$ integers. The cardinality of a finite set $A$ is denoted by $|A|$ and we also denote the standard Euclidean norm of any vector $v \in \mathbb{R}^d$ by $|v|$. The expressions $\lambda_{\max}(A)$ and $\lambda_{\min}(A)$ denote the largest and smallest eigenvalue of a symmetric matrix $A$ respectively. We denote the uniform distribution over a set $A$ by $\mathcal{U}(A)$. For two subsets $A, B \subset \mathbb{R}^d$ we denote by $A + B = \{a + b : a \in A, b \in B\}$ their Minkowski sum. We denote the permutation group on $k$ elements by $\mathfrak{S}_k$. An overview of the used variable names can be found in Appendix A.

## 2 Setting and Motivation

In this section, we illustrate our problem with an example and define the setup more formally. Consider the set of all animals. Those can be grouped into classes such as mammals, birds, or reptiles (in fact there is a rich hierarchical structure which we ignore here). Those groups were conceived by findings sets of animals that share many properties. Once these groups are found, we can predict unobserved properties by first identifying the cluster to which an animal belongs and then predict that the property is shared with animals in the same cluster. Note that this is a specific instance of a general problem in scientific inference, where we want to uncover a hidden grouping of similar entities from sparse observations about these entities.

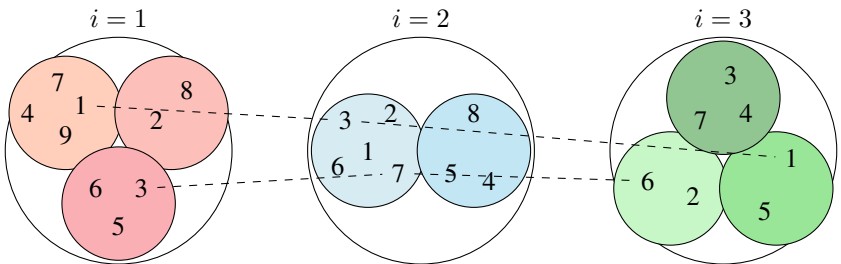

Figure 1: Illustration of the setting for $I = 3$ different groups clustered in 3, 2, and 3 subgroups respectively. Samples consist of one element of each group, the dashed lines indicate samples $(1, 3, 1)$ and $(3, 7, 6)$.

Here our main motivation, however, stems from the analysis of large language models where a similar problem arises implicitly during training. They are trained by next token prediction, so we do not expect them to learn structure by deductive reasoning such as cows are mammals, and mammals have lungs, so cows have lungs. Instead, their learning signal is whether a token can be replaced by another token for a given context. Thus, it is a natural question whether gradient descent-based training on token embeddings can uncover a hidden cluster structure of the data. Note that if the hidden structure is recovered, then generalization to unseen prompts is possible.

We now introduce our formal setup that captures key aspects of the discussion. We consider $I$ sets of $N_1, N_2, \ldots, N_I$ tokens or entities (such as words). For simplicity, we identify these with tokens from the set $[N_i]$. For each of the sets $[N_i]$ there is a partition $\mathcal{P}_i$ in $K_i$ classes which we can identify with the set $[K_i]$. Then we can encode the partitions through maps $\Pi_i : [N_i] \to [K_i]$ so that the partition is given by $\mathcal{P}_i = (\Pi_i^{-1}(k_i))_{k_i \in [K_i]}$, i.e., $\Pi_i$ encode the class membership. We consider the map

$$\Pi = \Pi_1 \otimes \ldots \otimes \Pi_I, \text{ i.e., } \quad \Pi(n_1, \ldots, n_I) = (\Pi_1(n_1), \ldots, \Pi_I(n_I)). \tag{1}$$

This structure is illustrated in Figure 1 Now we assume that there is a function $g : [K_1] \times [K_2] \times \cdots [K_I] \to \mathbb{R}$ which depends only on the classes. The map $f = g \circ \Pi$ extends this map to the tokens such that it only depends on the group a token belongs to. In the case where $g$ (and thus $f$) maps to $\{0, 1\}$ this can be interpreted as truth assignments, i.e., a statement consisting of a sequence $(n_1, \ldots, n_I)$ is true if $f(n_1, \ldots, n_I) = 1$ and false otherwise and this case is the main motivation of our work. More generally, $f$ could output the index of a suitable next token where in the 0, 1 case 0 could correspond to a negation while 1 to an end of sentence token. Our goal is to learn the partitions $\mathcal{P}_i$ or, equivalently, $\Pi$ up to a permutation and thereby identify the hidden structure.

We assume that we are given data in the form of samples $(\boldsymbol{n}^s, f(\boldsymbol{n}^s))$ where $\boldsymbol{n}^s = (n_1^s, \ldots, n_I^s)$ and $n_i^s \in [N_i]$. In other words, we try to learn the underlying structure from the interactions of a token with the other tokens, which is the same for every element of the partition. To simplify the notation and statements, we assume in the following that $N_i = N$ and $K_i = K$ for some $N$, $K$ and all $1 \le i \le I$. Our main interest concerns the case where $N$ is large, i.e., there are many entities and $K$ and $I$ are small, i.e., the number of groups is small.

Let us summarize several features that this model shares with real world problem, such as learning suitable embeddings in latent space.

- Hierarchical structures, i.e., groups of objects that share certain features, as discussed here, are abundant in language and science.
- We only receive an indirect learning signal for the value of $\Pi_i(n)$ through its interaction with other tokens.
- Interactions can be very complex, i.e., here the output depends on the interaction of $I$ different tokens and ignoring parts of the context makes learning infeasible.

On the other hand, many important features are abstracted away, e.g.:

- Here we assume that tokens from the same element of the partition interact in exactly the same way with other tokens while in reality there are many different partitions of the tokens depending on the broader context (e.g., we can group species by habitat, color, or, size each

resulting in different partitions) or there are exceptions, e.g., mammals generally do not lay eggs but the platypus does.

- Many more complex notions of similarity or further properties of embeddings such as a vector space structure are not covered. Also, there can be many uninformative features.
- We do not consider noisy data or errors in this work, which is crucial for real world applications.

# 3 Complexity-Theoretic and Information-Theoretic Analysis

We now study this learning problem in different settings. Let us first briefly discuss complexity-theoretic and information-theoretic properties of the learning problem to understand the general boundaries of this learning task. We first study the information-theoretic viewpoint, i.e., the question of how many samples are necessary to identify $\Pi$ (and potentially $g$). We focus on the case where we sample $\Pi$ and the data-samples uniformly at random. To learn an unstructured map $[N]^I \to \mathbb{R}$ we generally need of the order $N^I \ln(N^I)$ independent samples ($N^I$ when sampling without replacement).

For the structured setting we show that if $\Pi$ is drawn uniformly at random then generally order $K^I N \ln(N)$ samples are sufficient to learn $f$ and the partition induced by $\Pi$. In other words, for every token $n_i$ and each of the $K^I$ classes $\Pi^{-1}(\boldsymbol{k})$ we need of the order of $\ln(N)$ samples $\boldsymbol{n}$ such that $\Pi(\boldsymbol{n}) = \boldsymbol{k}$ and $\boldsymbol{n}_i = n_i$. In particular, for $N \gg K^I$ any token will interact only with $K^I \ln(N) \ll N$ other tokens, i.e., a very sparse subset of the other tokens.

We require the following necessary condition for identifiability: For every $k_i \neq k_i'$ there are $k_1, \ldots, k_{i-1}, k_{i+1}, \ldots, k_I \in [K]$ such that

$$g(k_1, \ldots, k_{i-1}, k_i, k_{i+1}, \ldots, k_I) \neq g(k_1, \ldots, k_{i-1}, k_i', k_{i+1}, \ldots, k_I). \tag{2}$$

Note that if this condition is indeed necessary because if it is not satisfied then it is not possible to distinguish $\Pi_i^{-1}(k_i)$ and $\Pi_i^{-1}(k_i')$. Clearly, we can generally only identify $\Pi$ up to a permutation symmetry, i.e., we can only find $\tilde{\Pi}$ such that there are permutations $\pi_i \in \mathfrak{S}_K$ such that $\Pi_i = \pi_i \tilde{\Pi}_i$. We have the following result.

**Theorem 1.** *Assume that $g : [K]^I \to \mathbb{R}$ is a function satisfying the assumption* (2)*. Assume we randomly sample maps $\Pi_i$ such that $\Pi_i(n_i) = k_i$ with probability $K^{-1}$ for all $i$, $n_i$, and $k_i$ and such that $(\Pi_i(n_i))_{i \in I, n_i \in [N]}$ are independent. Assume we are given $S$ samples $(\boldsymbol{n}, g \circ \Pi(\boldsymbol{n}))$ where $\boldsymbol{n} \sim \mathcal{U}([N]^I)$. Then there is a constant $N_0(I, K, \eta)$ such that with probability at least $1 - 2e^{-\eta}$ for $N \geq N_0(I, K, \eta)$ and*

$$S \geq 2^{2I+3} I K^I N \ln(N) \tag{3}$$

*we can recover $\Pi$ and $g$ up to permutations of $[K]$.*

This result is a special case of Theorem 6 in Appendix B which shows similar bounds for arbitrary maps $\Pi$ that are not necessarily random. In the more general setting, there are additional dependencies on the size of the preimages $\Pi_i^{-1}(k)$. Note that this dependency cannot be avoided because if there is a $\boldsymbol{k}$ such that $|\Pi^{-1}(\boldsymbol{k})| = 1$ and $g(\boldsymbol{k}) = 1$ and $g(\boldsymbol{k}') = 0$ for $\boldsymbol{k} \neq \boldsymbol{k}'$ then order $N^I$ samples are necessary to find $\boldsymbol{n}$ such that $\Pi(\boldsymbol{n}) = \boldsymbol{k}$ and thus $\Pi$. The general proof idea is to bound the probability that any fixed $\Pi' \neq \Pi$ is compatible with the dataset. It turns out that it is possible to bound this probability in terms of the partitions distance of the partitions induced by $\Pi$ and $\Pi'$. Then we are left with bounding the number of partitions, and we conclude with the union bound. We now show that this bound is essentially tight.

**Theorem 2.** *Let $g : [K]^I \to \mathbb{R}$ be a function such that $g(1, k_2, \ldots, k_I) = g(2, k_2, \ldots, k_I)$ for all $k_2, \ldots, k_I \in [K]$ except when $k_2 = k_3 = \ldots = k_I = 1$. Assume that $N$ is divisible by $K$ and that $|\Pi_i^{-1}(k)| = N/K$ for all $i \in [I]$ and $k \in [K]$. Given $3 \leq S \leq NK^{I-1} \ln(N/K)/4$ samples $(\boldsymbol{n}^s, g \circ \Pi(\boldsymbol{n}^s))$ where $\boldsymbol{n} \sim \mathcal{U}([N]^I)$ i.i.d. Then the function $\Pi$ is identifiable with probability at most $2e^{-\sqrt{N/K}}$.*

The proof of this result can be found in Appendix B. Next, we emphasize that while typically a rather small number of samples is sufficient to learn $\Pi$ it can generally be very hard to do this in practice. More concretely, we show that for $I \geq 3$ even deciding whether there is a map

$\Pi = \Pi_1 \otimes \ldots \otimes \Pi_I : [N]^I \to [K]^I$ such that $f = g \circ \Pi$ given access to samples of the form $(\boldsymbol{n}^s, t^s)$ is NP-complete. We show that this is true even if $g$ is known, $I = 3$ and $K = 2$.

**Theorem 3.** *Consider the map* $g : \{0, 1\}^3 \to \{0, 1\}$ *given by*

$$g(k_1, k_2, k_3) = \mathbf{1}_{k_1+k_2+k_3=2}. \tag{4}$$

*Then it is an NP-complete problem to decide given samples of the form* $(n_1^s, n_2^s, n_3^s, t^s) \in [N]^3 \times \{0, 1\}$ *whether there is a map* $\Pi = \Pi_1 \otimes \Pi_2 \otimes \Pi_3$ *with* $\Pi_i : [N] \to \{0, 1\}$ *such that* $t^s = g \circ \Pi(n_1^s, n_2^s, n_3^s)$ *for all samples.*

The proof of this result can be found in Appendix C. Now that we established under what the conditions $\Pi$ can in principle be learned, and clarified that this might be hard in general, we next discuss how we can find $\Pi$ in practice. First, we remark that Theorem 3 rules out the existence of any general fast algorithms to learn $\Pi$. Given the combinatorial nature and the hardness of the problem, it is natural to reformulate the task as a constraint satisfaction problem which can then be solved using standard SAT solvers (see, e.g., [14] for a review). Indeed, we can introduce Boolean variables $t_{kn}^i$ for $i \in [I]$, $k \in [K]$, and $n \in [N]$ which encode whether $\Pi_i(n) = k$ and $r_{\boldsymbol{k}v}$ for every $\boldsymbol{k} \in [K]^I$ and $v \in \mathrm{Im}(f)$ in the (finite) image of $f$ that encode whether $g(\boldsymbol{k}) = v$. It is then relatively straightforward to then express the conditions for the map $\Pi$ as a constraint satisfaction problem which is satisfiable if and only if there are maps $\Pi$ and $g$ such that $t^s = g(\Pi(\boldsymbol{n}^s))$ holds for all samples. We outline the construction in more detail in Appendix C. We leave the task of developing and studying efficient algorithms for the considered problem for future work because the main motivation of this paper is rather to understand how the complex statistical patterns can be extracted using simple gradient based algorithms. This will be investigated in the next section.

## 4 Analysis of Gradient Descent Dynamics

In this section, we investigate under what conditions the clustering induced by $\Pi$ can be learned using gradient descent on token embeddings. Our main finding is that for uniformly random $\Pi$ and $S$ sufficiently large gradient descent can be used to uncover or at least preserve the cluster structure of the embeddings. This shows that while the general problem is NP hard typical random instances with sufficiently many samples can be solved with straightforward algorithms quickly. This is in spirit similar to the results found in, e.g., [5]. Let us start by introducing the setting.

**Setting.** We assume that we have token embeddings for each of the $I$ sets $[N_1]$ to $[N_I]$, i.e., we assume that there are vectors $v(i, n) \in \mathbb{R}^D$ for some $D$ and all $i \in [I]$, $n \in [N]$. Based on these embeddings, we assume that we are given a function $\hat{f} : \mathbb{R}^{ID} \to \mathbb{R}$ that transforms the embeddings into a prediction. We will abuse notation and write for $\boldsymbol{n} \in [N]^I$

$$\hat{f}(\boldsymbol{n}) = \hat{f}(v(1, \boldsymbol{n}_1), \ldots, v(I, \boldsymbol{n}_I)), \tag{5}$$

i.e., we will suppress the map from tokens to embeddings in the notation.

Now we consider gradient descent for the embedding vectors using the least square loss on training samples, i.e., the loss of a sample $(\boldsymbol{n}, t = f(\boldsymbol{n}))$ is $(\hat{f}(\boldsymbol{n}) - f(\boldsymbol{n}))^2$.

We assume that we are given a dataset $\mathcal{D} = \{\boldsymbol{n}^1, \ldots, \boldsymbol{n}^S\} \sim \mathcal{U}([N]^I)^S$. Then the empirical loss reads (the division by 2 is convenient for the gradient evaluation later)

$$\hat{\mathcal{L}}((v(i, n))_{i \in [I], n \in [N]}) = \frac{1}{2} \sum_{s=1}^{S} (\hat{f}(\boldsymbol{n}^s) - f(\boldsymbol{n}^s))^2. \tag{6}$$

We also define shorthands for certain concatenations of embeddings. For a sample $\boldsymbol{n} \in [N]^I$ we denote the collection of embeddings by $\boldsymbol{v}(\boldsymbol{n}) = (v(1, \boldsymbol{n}_1), \ldots, v(I, \boldsymbol{n}_I))$. Using the convention (5) we can then write $\hat{f}(\boldsymbol{v}(\boldsymbol{n})) = \hat{f}(\boldsymbol{n})$.

Moreover, we define $\bar{\boldsymbol{v}}(i) \in \mathbb{R}^{DN}$ as the concatenation of the vectors $v(i, 1), \ldots, v(i, n)$, i.e., the combined embedding for the $i$-th slot and $\bar{\boldsymbol{v}} \in \mathbb{R}^{DNI}$ as the concatenation of the vectors $\bar{\boldsymbol{v}}(i)$ for $1 \leq i \leq I$, i.e., all token embeddings concatenated. We consider the regularized loss given by

$$\hat{\mathcal{R}}^\lambda(\bar{\boldsymbol{v}}) = \frac{N}{S} \hat{\mathcal{L}}(\bar{\boldsymbol{v}}) + \frac{\lambda}{2} |\bar{\boldsymbol{v}}|^2. \tag{7}$$

Note that the scaling by $N/S$ (instead of usual $1/S$) is natural because every token embedding $v(i, n)$ occurs in approximately $S/N$ of the samples and so the scaling ensures that the gradient of $\hat{\mathcal{R}}^\lambda$ with respect to the token embedding $v(i, n)$ is of order one. Now, we consider the continuous gradient descent of the loss with respect to the embeddings, i.e., we consider (omitting the time variable from the notation)

$$\dot{v}(i, n) = -\frac{\mathrm{d}}{\mathrm{d}v(i, n)} \hat{\mathcal{R}}^\lambda(\bar{\boldsymbol{v}}) = -\frac{N}{S} \frac{\mathrm{d}}{\mathrm{d}v(i, n)} \hat{\mathcal{L}}(\bar{\boldsymbol{v}}) - \lambda v(i, n). \tag{8}$$

This introduces a time dynamics on the token embeddings. We indicate the time dependence by $v(i, n, t)$ but we drop $t$ if not necessary. Our main goal is to understand which conditions ensure that the token embeddings $v(i, n, t)$ and $v(i, n', t)$ converge to each other if $\Pi_i(n) = \Pi_i(n')$ as $t \to \infty$. To investigate this we define the center of the class embeddings by

$$w(i, k, t) = \frac{1}{|\Pi_i^{-1}(k)|} \sum_{n \in \Pi_i^{-1}(k)} v(i, n, t) \tag{9}$$

and we consider the deviations from the class centers given by

$$\delta(i, n, t) = v(i, n, t) - w(i, \Pi_i(n), t). \tag{10}$$

Thus the vectors $\delta(i, n, t)$ capture whether we recover the cluster structure, in particular if all norms $|\delta(i, n, t)|$ are small then we essentially recovered the hidden structure. Therefore, we define

$$\delta_{\max}(t) = \max_{i \in [I]} \max_{n \in [N]} |\delta(i, n, t)|. \tag{11}$$

Similarly, to the notation introduced before we consider for $\boldsymbol{k} \in [K]^I$ the vector $\boldsymbol{w}(\boldsymbol{k}) = (w(1, \boldsymbol{k}_1), \ldots, w(I, \boldsymbol{k}_I))$. As in (5) we abuse notation and write

$$\hat{g}(\boldsymbol{k}) = \hat{f}(\boldsymbol{w}(\boldsymbol{k})) = \hat{f}(w(1, \boldsymbol{k}_1), \ldots, w(I, \boldsymbol{k}_I)). \tag{12}$$

Similar to $\bar{\boldsymbol{v}}(i)$ and $\bar{\boldsymbol{v}}$ we introduce $\bar{\boldsymbol{w}}(i)$ as the concatenation of $(w(k, i))_{k \in [K]}$ and $\bar{\boldsymbol{w}}$ as the concatenation of $\bar{\boldsymbol{w}}(i)$.

**Assumptions.** Our first result for the gradient dynamics states that clusters are stable under the dynamics if the initial loss is sufficiently small. More precisely, this means that we assume that $v(i, n)$ and $v(i, n')$ are close initially whenever $\Pi_i(n) = \Pi_i(n')$, i.e., $\delta_{\max}(0)$ is small. In addition, we assume that $|\hat{g}(\boldsymbol{k}) - g(\boldsymbol{k})|$ is small. To capture this we define

$$r_{\max}(t) = \max_{\boldsymbol{k} \in [K]^I} |\hat{g}(\boldsymbol{w}(\boldsymbol{k}, t)) - g(\boldsymbol{k})|. \tag{13}$$

Then the result shows that $\delta_{\max}$ stays small for all times if $S \gtrsim N^2$ under mild additional assumptions. In other words, if we start from the correctly learned substructures and $\hat{g}(\boldsymbol{k}) \approx g(\boldsymbol{k})$ for all $\boldsymbol{k} \in [K]^I$ then this remains true for all times. Note that while we phrase smallness as an assumption on the mean embeddings $w(i, k)$ this is generally a consequence of $\delta_{\max}$ small and a small empirical loss $\hat{\mathcal{L}}(\bar{\boldsymbol{v}})$. Let us now state the required assumptions.

**Assumption 1.** *We assume that the map* $\Pi : [N]^I \to [K]^I$ *is approximately balanced which means that for all* $i \in [I]$, $k \in [K]$

$$\frac{N}{2K} \leq \Pi_i^{-1}(k) \leq \frac{2N}{K}. \tag{14}$$

This assumption ensures that clusters are of approximately equal size. We have already seen in Section 3 that different cluster sizes increase the sample complexity of learning $\Pi$.

**Assumption 2.** *We assume that there is a convex set* $\Omega \subset \mathbb{R}^D$ *and a constant* $M' \geq 1$ *such that the following bound holds*

$$\sup_{\boldsymbol{v} \in \Omega^I} \sup_{i_1, i_2, i_3 \in [DI]} \max\left(|\hat{f}(\boldsymbol{v})|, |\partial_{i_1} \hat{f}(\boldsymbol{v})|, |\partial_{i_1} \hat{f}(\boldsymbol{v})|, |\partial_{i_1} \partial_{i_2} \partial_{i_3} \hat{f}(\boldsymbol{v})|\right) \leq M' = \frac{\sqrt{M}}{4}. \tag{15}$$

*Here it is convenient to introduce* $M = 16M'^2$ *so that later certain errors in a Taylor approximation are bounded by* $M$. *We also assume that*

$$\max_{\boldsymbol{k} \in [K]^I} |g(\boldsymbol{k})| \leq M' = \frac{\sqrt{M}}{4}. \tag{16}$$

This is a rather mild assumption. For $C^3$ functions $\hat{f}$ and $\Omega$ bounded this is always true. The next assumption entails a rigidity of approximate minimizers of the loss.

**Assumption 3.** *We assume that for all embeddings $w(i, k) \in \mathbb{R}^D$ for $i \in [I]$, $k \in [K]$ that satisfy*

$$r_{\max} = \max_{\boldsymbol{k} \in [K]^I} |\hat{f}(\boldsymbol{w}(\boldsymbol{k})) - g(\boldsymbol{k})| \leq 1 \tag{17}$$

*the bound*

$$\omega_0 = \min_{k,i} \lambda_{\min} \left( \sum_{\boldsymbol{k} \in [K]^I, \boldsymbol{k}_i = k} \nabla_{w(i,k)} \hat{f}(\boldsymbol{w}(\boldsymbol{k})) \otimes \nabla_{w(i,k)} \hat{f}(\boldsymbol{w}(\boldsymbol{k})) \right) > 0 \tag{18}$$

*holds for some positive constant $\omega_0$. Of course, the bound $r_{\max} \leq 1$ could be replaced by any other constant.*

Note that this condition can only hold if $D \leq K^{I-1}$, i.e., the latent space dimension cannot be too large. The high level intuition of this assumption is essentially that (at least if $\sum_{\boldsymbol{k}}(\hat{f}(\boldsymbol{w}(\boldsymbol{k})) - g(\boldsymbol{k}))^2$ is small) there is no direction $v \in \mathbb{R}^D$ such that $v \cdot \nabla_{w(i,k)}\hat{f}(\boldsymbol{w}(\boldsymbol{k})) \approx 0$ for all $\boldsymbol{k}$ such that $\boldsymbol{k}_i = k$, i.e., we cannot move one single embedding without changing the output $\hat{f}(\boldsymbol{w}(\boldsymbol{k}))$ for at least one $\boldsymbol{k}$. If this condition does not hold, then we cannot guarantee that $\sum_{\boldsymbol{k}}(\hat{f}(\boldsymbol{w}(\boldsymbol{k})) - f(\boldsymbol{k}))^2$ is minimized for a unique $w(i, k)$ (for all other embeddings fixed). This generally prevents concentration of $v(i, n)$. Note that this condition does not ensure that there is a unique minimizer $\bar{w}$, in particular there could still be a rotationally invariant family of embeddings $w(i, k)$ such that $\hat{f}(\boldsymbol{w}(\boldsymbol{k})) = g(\boldsymbol{k})$ for all $\boldsymbol{k} \in [K]^I$. Finally, we need a further mild assumption that ensures that mean token embeddings $w(i, k)$ stay bounded in some set if the loss is small. This can be achieved, e.g., if $\hat{f} \to \infty$ if $|\boldsymbol{w}(\boldsymbol{k})| \to \infty$.

**Assumption 4.** *We assume that for all collections of mean embeddings $w(i, k) \in \mathbb{R}^D$ for $i \in [I]$, $k \in [K]$ that satisfy*

$$r_{\max} = \max_{\boldsymbol{k} \in [K]^I} |\hat{f}(\boldsymbol{w}(\boldsymbol{k})) - g(\boldsymbol{k})| \leq 1 \tag{19}$$

*there is a convex set $\Omega_0 \subset \mathbb{R}^d$ such that $w(i, k) \in \Omega_0$ for all $i \in [I]$, $k \in [K]$. Again, the right-hand side of the bound $r_{\max} \leq 1$ could be replaced by any other constant.*

**Results.** The first stability theorem can then be stated as follows.

**Theorem 4.** *Let $\Pi : [N]^I \to [K]^I$ be approximately balanced as stated in Assumption 1 Assume that the functions $g : [K]^I \to \mathbb{R}$ and $\hat{f} : \mathbb{R}^{ID} \to \mathbb{R}$ satisfy Assumption 3 for some $\omega_0 > 0$ and Assumption 4 for some convex set $\Omega_0$. Assume that Assumption 2 holds for some $M$ and the set $\Omega = \Omega_0 + B_2(0)$. Then there are constants $c_1, C_2, C_3, C_4 > 0$ depending on $I$, $M$, $D$, and $\omega_0$ such that for all initial embeddings $v(i, n, t = 0) \in \mathbb{R}^D$ for $i \in [I]$ and $n \in [N]$ satisfying*

$$\delta_{\max}(0) \leq C_2 K^{-3I/2}, \quad r_{\max}(0) \leq C_3 K^{-3I/2} \tag{20}$$

*and sample size*

$$S \geq c_1 \max\left(K^{3I} N^2 \ln^2(N), N \ln(N) K^{9I/2}\right) \tag{21}$$

*the following holds with probability at least $1 - S^{-1}$ over the randomness of the dataset. When considering the gradient dynamics of the embeddings given by (8) the bound*

$$R = \limsup_{t \to \infty} r_{\max}(t) \leq 1 \tag{22}$$

*holds and moreover*

$$\limsup_{t \to \infty} \delta_{\max}(t) \leq C_4 K^{3I/2} \sqrt{\ln(S) \frac{N}{S}} R. \tag{23}$$

*In particular $\delta_{\max}(t) \to 0$ if $r_{\max}(t) \to 0$, i.e., all token embeddings for one fixed class converge to the same point.*

This result shows that for order $N^2 \ln(N)$ samples and initialization sufficiently close to a global minimum the cluster structure remains stable. We do not provide conditions that ensure $r_{\max}(t) \to 0$ which guarantees convergence to 0 loss and perfect recovery of the clusters.

Next we note that we cannot expect the clustering to be stable in general even if $\delta_{\max}(0)$ is arbitrarily small if initialization is not close to a minimum. This is true even in the simplest case where $I = K = 1$, i.e, we consider gradient descent for a single function value. Then gradient descent does not necessarily converge to a global minimum and close by points do not necessarily stay close because gradient descent is not well posed. Let us clarify this by an example.

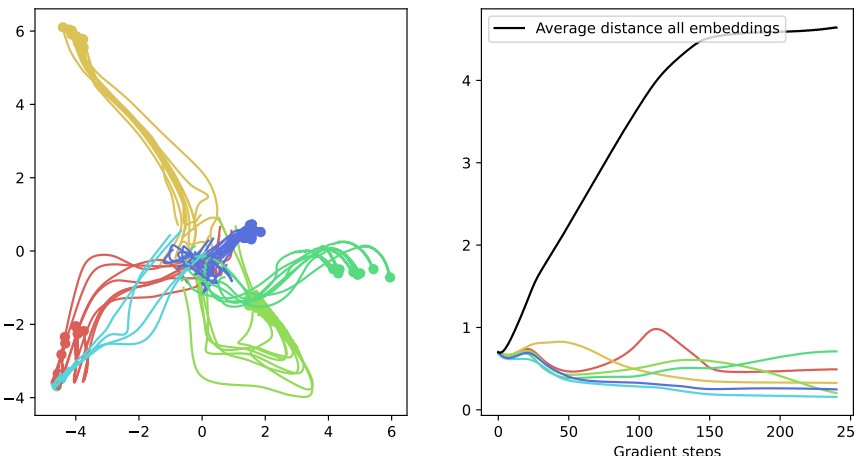

Figure 2: Simulation of the setting in Theorem 5 with $N = 1000$, $K = I = 3$, $D = 2$, $\lambda = 0$, $S = 100.000$. (left) trajectories of 50 randomly sampled tokens from 6 different classes. (right) Average distance of token embeddings within a class for different classes (colored) and average distance between all pairs of embeddings (black).

**Example 1.** Assume $I = K = D = 1$, $N > 1$, $\hat{f}(x) = -x^2 + x^3$ and $f(n) = -2$. Consider the dataset $\mathcal{D} = \{1, \dots, n\}$. Assume that $v(1, n, t = 0) \sim N(0, \sigma^2)$ for any $\sigma^2 > 0$. Then the gradient dynamics introduced in (8) reads

$$\dot{v}(1, n, t) = \hat{f}'(v(1, n, t))(\hat{f}(v(1, n, t)) - f(n)) \approx 4v(1, n, t) \quad \text{if } |v(1, n, t)| \text{ is small.} \quad (24)$$

We find that $v(1, n, t) \to 2/3$ (which is a local minimum of $\hat{f}$ as $t \to \infty$ if $v(1, n, t = 0) > 0$. On the other hand $v(1, n, t) \to -1$ if $v(1, n, t = 0) < 0$. So in this case $\delta_{\max}(0) = O(\sigma^2 \sqrt{\ln(N)})$ but $\delta_{\max}(t) \to 5/3$ as $t \to \infty$. Slight modifications show that also $\delta_{\max} \to \infty$ is possible.

The previous example shows that without additional assumption, we cannot expect to recover the structure of the data. Therefore, we impose additional restrictions on the function $\hat{f}$. As apparent from Example 1 and also from the bound in Lemma 1 in Appendix E, it is the curvature of the function $\hat{f}$ that can push token embeddings of the same class apart. We therefore consider the function class of slow-wise linear functions defined as follows.

**Definition 1.** We call $\hat{f}$ *slot-wise linear* if for every $\boldsymbol{v} = (v(1), \dots, v(I)) \in \mathbb{R}^{D \times I}$ and any $i \in [I]$, $\alpha, \beta \in [D]$ the relation

$$\frac{\mathrm{d}}{\mathrm{d} v_\alpha(i)} \frac{\mathrm{d}}{\mathrm{d} v_\beta(i)} \hat{f}(\boldsymbol{v}) = 0 \quad (25)$$

holds.

Let us denote for $v \in \mathbb{R}^D$ by $\tilde{v} \in \mathbb{R}^{D+1}$ the vector $v$ where we append a 1. The most general slot-wise linear function is then of the form

$$f(\boldsymbol{v}) = T(\tilde{v}(1) \otimes \tilde{v}(2) \otimes \dots \otimes \tilde{v}(I)) \quad (26)$$

where $T : \mathbb{R}^{(D+1)^I} \to \mathbb{R}$ is a linear map. Note that this class covers linearized attention where the embeddings $v(i,n)$ are split in three separate parts that are used to form key, query, and value vectors. For this function class we can show stronger clustering results.

**Theorem 5.** *Let $\Pi$ be approximately balanced, i.e., assume that Assumption 1 holds. Let $\hat{f} : \mathbb{R}^{ID} \to \mathbb{R}$ be a slot-wise linear. Assume that $v(i,n,t)$ follow the gradient dynamics (8) and that $v(i,n,t) \in \Omega$ for all $i \in [I]$, $n \in [N]$ and $t > 0$ for some convex set $\Omega$. Assume that Assumption 2 holds with constant $M$ for the set $\Omega$. Let $C_1$ be the constant from Lemma 1 (which depends on I, D, M). Assume that at initialization*

$$|v(i,n,t=0)| \leq \frac{\lambda}{8C_1}. \tag{27}$$

*Then there are constant $C_2, C_3 \geq 0$ depending on M, I, D, such that for*

$$S \geq \max\left(C_2 \frac{N^2 K^{I-1}}{\lambda^2} \ln^2(N/\lambda), C_3 \frac{NK^{I-1}}{\lambda^4} \ln(N/\lambda)\right) \tag{28}$$

*the bound*

$$\delta_{\max}(t) \leq \max\left(\delta_{\max}(0)e^{-\lambda t/8}, \frac{4C_1 K^{(I-1)/2}}{\lambda}\sqrt{\ln(S)\frac{N}{S}}\right) \tag{29}$$

*holds for all $t \geq 0$.*

The high level summary of this result is that for order $N^2 \ln(N)$ samples the clusters can be recovered up to an error of order $\lambda^{-1}\sqrt{\ln(S)N/S}$. Note that the a-priori assumption that the gradient flow is restricted to the set $\Omega$ might appear difficult to guarantee in practice. However, we conjecture that the results extend to gradient dynamics clamped at the boundary. Moreover, in Lemma 2 we prove that the mean embeddings $w(i,k,t)$ stay within a ball of radius $R = O(\sqrt{\lambda}^{-1})$ for all times. This allows us to prove Theorem 8 which does not require any a-priori bounds on the evolution of $v(i,n,t)$ but comes at the price that the constants $C_1$, $C_2$, and $C_3$ depend on $\lambda$ so we cannot infer the explicit $\lambda$ dependence. Let us make an important remark.

*Remark* 1. While we state our results for a fixed function $\hat{f}$ this function could in principle be time-dependent, e.g., $\hat{f}$ could be given by a neural network, and we consider gradient descent not only on the token embeddings but also on the network parameters. The only requirement is that the assumptions hold uniformly for all times $t$. In particular, for Theorem 5 we only need to ensure that the derivatives of $\hat{f}$ stay uniformly bounded in time. This can, e.g., be guaranteed by clipping the parameters of the slot-wise linear map $\hat{f}$.

Our results so far show that we can recover the right cluster structure in the sense that the embeddings from the same group cluster. However, this leaves open whether there is any non-trivial dynamics at all, i.e., all embeddings could cluster at the same point. This is in general not the case as can be seen from Corollary 1 which states that in the setting of Theorem 5 and for large times the dynamics of the cluster-means follow the equation

$$\dot{w}(i,k) = -\sum_{\boldsymbol{k}\in[K]^I, \boldsymbol{k}_i = k} \frac{\alpha_{\boldsymbol{k},i}}{2}\nabla_{w(i,\boldsymbol{k}_i)}(\hat{g}(\boldsymbol{k}) - g(\boldsymbol{k}))^2 - \lambda w(i,k) + O\left(\sqrt{\ln(S)\frac{N}{S}}\right) \tag{30}$$

where $(2K)^{-I} \leq \alpha_{\boldsymbol{k},i} \leq (2/K)^I$ are positive numbers. This shows that $w(i,k)$ follow generally a non-trivial dynamic (and this also justifies the scaling as this expression is of order 1). So in the generic case the cluster structure will be revealed if the numbers of samples is sufficiently large, however, there is no general guarantee that the clusters are well separated. As an illustration of this result we refer to Figure 2 where the clustering of the embeddings becomes apparent.

**Proof idea and overview.** Let use here give a quick summary of the main steps of the proof and where to find them. The first important ingredient is an expansion of the loss gradient. We Taylor expand the loss of a sample $\boldsymbol{v}(\boldsymbol{n})$ around the point $\boldsymbol{w}(\Pi(\boldsymbol{n}))$ to second order with remainder terms, the relevant calculations can be found in Appendix D (see Proposition 1 for the outcome). A second ingredient are concentration bounds for certain datapoint-statistics random variables and

random matrices. Those are derived in Section G with the necessary results collected in Theorem 9. Combining the Taylor expansion with the concentration result, we can extract the dominant terms of the expansion (see Appendix E). Moreover, we obtain such an expansion for $\dot{w}(i, k)$ (see Corollary 1) and thus the displacements $\dot{\delta}(i, n)$ (see Corollary 2). This expansion can then be used to control $\partial_t |\delta(i, n)|^2$ (see Lemma 1) which is sufficient to control $\delta_{\max}$.

**Discussion of assumptions**  Let us contemplate the differences and similarities to training token embeddings in neural networks.

- For the first main result Theorem 4 we make minimal assumptions on $\hat{f}$ so this could in principle be a neural network applied to the token embeddings. The second result, Theorem 5, is more restrictive but covers subclasses of linearized attention.
- An important feature of the results is that $\hat{f}$ itself could be time dependent (see Remark 1).

Differences to standard training of neural networks are:

- We use continuous time gradient descent instead of stochastic gradient descent. This is a frequently used modification and in suitable limits those converge (see, e.g., [20]).
- We use mean squared error, while sequence modelling usually relies on cross entropy loss. This simplification is frequently used in theoretical analysis, but it is expected that results generally extend to the non-convex cross entropy loss.
- A more crucial difference is that the embedding space dimension in practice is usually chosen large to provide large representation capacity. Here $D$ has to be rather small to allow a unique optimal solution of the token embeddings that allows us to recover the cluster structure.

## 5   Conclusion

In this paper, we considered a learning problem where we try to recover a partition of tokens from their interaction with other tokens. This can be seen as a toy problem for next token prediction in LLMs, but also more broadly as a problem in scientific inference. We studied this problem from different perspectives, namely from an information-theoretic, complexity-theoretic, and gradient descent based viewpoint. We found that order $N \ln(N)$ samples are sufficient to recover the partition for $N$ tokens, while we showed that $N^2 \ln(N)$ samples are sufficient for gradient based methods. There are several natural open follow-up questions. First, there are some open questions regarding the tightness of our analysis of the gradient descent. In particular, it is a natural question whether already $\Omega(N \ln(N))$ samples are sufficient to control $\delta_{\max}$ which is the information-theoretic threshold and would be similar to the optimal results for matrix completion [18, 7]. Another interesting question for future research is whether Theorem 5 holds for standard initialization schemes for the token embeddings. Secondly, it is of interest to relax the notion of clustering of embeddings to more general notions that still allow recovering some structure but are also applicable to high dimensional latent spaces and potentially to multiple partitions and relations on the same tokens (e.g., tokens belonging to different clusters). Thirdly, it is a natural question whether this work can be connected more closely to empirical findings.

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

# Supplementary Material

This supplementary material is structured as follows. We first review the notation used in the paper in Appendix A. Then we provide the proofs for the information-theoretic results in Section 3 in Appendix B and the proof of Theorem 3 and a reduction to a constraint satisfaction problem in Appendix C. The proofs of the main results concerning the gradient flow rely on a careful Taylor expansion of the loss gradient. This expansion can be found in Appendix D and bounds for this expansion are derived in Appendix E. Based on these bounds, we can prove our main results in Appendix F. An important ingredient in bounding the Taylor expansion are concentration results for the dataset statistics that can be found in Appendix G. Finally, we review some results on random matrices in Appendix H which are necessary for the concentration bounds and we review the definition of Kronecker products in Appendix I.

## A   Overview of Notation used

Let us here collect important notation used throughout the paper and the proofs.

**General notation.**

- Numbers up to $n$: $[n] = \{1, \ldots, n\}$
- Eigenvalues of a matrix: $\lambda_i(A)$
- Largest eigenvalue: $\lambda_{\max}(A)$
- Operator norm of a matrix: $\|A\|$

**Notation used in the learning problem.**

- Number of slots: $I$
- Number of tokens for each slot: $N$
- Number of classes: $K$
- Number of samples: $S$
- Map defining the partition in subclasses: $\Pi = \Pi_1 \otimes \ldots \otimes \Pi_I : [N]^I \to [K]^I$
- Function on classes: $g : [K]^N \to \mathbb{R}$
- Induced function on tokens: $f : [N]^I \to \mathbb{R}, f = g \circ \Pi$

**Notation used in the gradient descent analysis.**

- Dimension of latent space: $D$
- Embedding for token $n$ of slot $i$: $v(i, n)$
- Token embeddings for a sample $\boldsymbol{n}$: $\boldsymbol{v}(\boldsymbol{n}) = (v(1, \boldsymbol{n}_1), \ldots, v(I, \boldsymbol{n}_I))$
- Token embeddings of a slot $i$: $\bar{\boldsymbol{v}}(i) = (v(i, n))_{n \in [N]}$
- All token embeddings: $\bar{\boldsymbol{v}} = (\bar{\boldsymbol{v}}(i))_{i \in [I]}$
- Mean of cluster embeddings: $w(i, k)$
- Sample version: $\boldsymbol{w}(\boldsymbol{k}) = (w(1, \boldsymbol{k}_1), \ldots, w(I, \boldsymbol{k}_I))$
- Mean token embeddings for slot $i$: $\bar{\boldsymbol{w}}(i) = (w(i, k))_{k \in [K]}$
- All mean embeddings: $\bar{\boldsymbol{w}} = (\bar{\boldsymbol{w}}(i))_{i \in [I]}$
- Displacements from cluster center: $\delta(i, n) = v(i, n) - w(i, \Pi_i(n))$
- Displacements of a sample: $\boldsymbol{\delta}(\boldsymbol{n}) = (\delta(1, \boldsymbol{n}_1), \ldots, \delta(I, \boldsymbol{n}_I))$
- Displacements of all tokens for slot $i$: $\bar{\boldsymbol{\delta}}(i) = (\delta(i, n))_{n \in [N]}$
- Function on token embeddings: $\hat{f} : \mathbb{R}^{DI} \to \mathbb{R}$, identified with $\hat{f} : [N]^I \to \mathbb{R}$, $\hat{f}(\boldsymbol{n}) = \hat{f}(\boldsymbol{v}(\boldsymbol{n}))$
- Function on classes: $\hat{g} : [K]^I \to \mathbb{R}$, $\hat{g}(\boldsymbol{k}) = \hat{f}(\boldsymbol{w}(\boldsymbol{k}))$
- Regularization: $\lambda$

# B Proofs for Information-Theoretic Results

In this section, we provide the proofs and additional results for Section 3. Let us first start to state the general information-theoretic bound that handles the case where $|\Pi_i^{-1}(k)|$ might be of arbitrary size.

**Theorem 6.** *Let $g : [K]^I \to \mathbb{R}$ satisfy (2) and there is a projection $\Pi = \Pi_1 \otimes \ldots \otimes \Pi_I : [N]^I \to [K]^I$. Assume there is $L > 0$ such that for every $\boldsymbol{k} \in [K]^I$ we have $\Pi^{-1}(\boldsymbol{k}) \geq N^I/L$. Moreover, assume for all $i \in [I]$ and $k \in [K]$ the bound $|\Pi_i^{-1}(k)| \geq M$, i.e., every class has at least $M$ members. Assume we are given $S$ samples $\{(\boldsymbol{n}^s, g \circ \Pi(\boldsymbol{n}^s)) : s \in [S]\}$ where $\boldsymbol{n} \sim \mathcal{U}([N]^I)$. If*

$$S \geq 2 \cdot \max \left( \frac{K^{I+1}L}{M} \max \left( \ln(K) I N, \eta \right), 2^I L N \max(2 \ln(NK)I, \eta) \right) \tag{31}$$

*for some $\eta \geq 2$ then with probability at least $1 - 6e^{-\eta}$ we can recover $\Pi$ and $g$ up to permutations of $[K]$. In other words, for any map $\Pi' = \Pi_1' \otimes \ldots \otimes \Pi_I'$ and $g'$ such that $g \circ \Pi(\boldsymbol{n}^s) = g' \circ \Pi'(\boldsymbol{n}^s)$. There are permutations $\pi_i \in \mathfrak{S}_K$ such that $\Pi_i = \pi_i \circ \Pi_i'$ and the corresponding relation holds for $g$, $g'$.*

A few remarks to explain this result are in order.

*Remark* 2. The scaling of $S$ might appear slightly complicated, so let us comment on this. We are mostly interested in the case where $N$ large and $K$ stays bounded. In addition, we are primarily interested in the regime where $L \leq CK^I$ bounded and $M \geq cN/K$ (which holds for random $\Pi$), i.e., the sampling probability of each class $\bar{k} \in [K]^I$ is of similar size. In this case, the first term in the condition (31) for $S$ stays constant as $N \to \infty$. The second term dominates, and we see that we need only $O(N \ln(N))$ samples to identify $\Pi$ and $g$. This is the setting studied in Theorem 1 in the main text. The result is essentially tight in this limit (up to the dependence on $I$) as stated in Theorem 2. Note that as remarked in the main text, the dependence on $L$ cannot be avoided. Indeed, consider the extreme case where $\Pi_i(n) = 1$ for $n = 1$ and $\Pi_i(n) = 2$ for $n > 1$ and $g(k_1, \ldots, k_I) = 1$ iff $k_1 = \ldots = k_I = 0$ and $g(k_1, \ldots, k_I) = 1$ otherwise. Then we have $L = N^I$ and we also need of order $N^I$ samples to sample the point $(0, 0, \ldots, 0) \in [N]^I$ which is necessary to identify $\Pi$.

*Proof.* Let us first introduce some notation. We denote the set of samples

$$\mathcal{D} = \{\boldsymbol{n}^1, \ldots, \boldsymbol{n}^S\}. \tag{32}$$

Consider two partitions $\mathcal{P}, \mathcal{Q}$ of a set $[N]$. We denote by $\mathcal{P}^A$ for $A \subset [N]$ the restriction of a partition to a subset (i.e., $\{P \cap A : P \in \mathcal{P}\}$). The partition distance is defined by

$$D(\mathcal{P}_1, \mathcal{P}_2) = \min\{|A^c| : \mathcal{P}_1^A = \mathcal{P}_2^A, A \subset [N]\}, \tag{33}$$

i.e., the minimal number of elements that need to be removed such that the partitions agree.

We call $\Pi'$ compatible with the datapoints $\mathcal{D}$ if there is a $g'$ such that $g' \circ \Pi' = g \circ \Pi$ on the data. Now the general strategy is to consider any other candidate map $\Pi'$ and upper bound the probability that is compatible, i.e., for some function $g'$ the functions $f' = g'\Pi$ and $f$ agree on the data. We will then conclude by applying the union bound over maps $\Pi'$. Thus, we need to prove an upper bound on the number of partitions with partition distance at most $\Delta$ and a lower bound on the error probability for a given $\Delta$. Let us start with the latter. Denote by

$$\mathcal{P}_i = \{\Pi_i^{-1}(k) : k \in [K]\} \tag{34}$$

the partition generated by $\Pi_i$. Consider any other map $\Pi' = \Pi_1' \otimes \ldots \otimes \Pi_I' : [N]^I \to [K]^I$ with corresponding partitions $\mathcal{P}_i'$. We define for any such $\Pi'$ the quantity

$$\Delta(\Pi') = \max_{i \in [I]} D(\mathcal{P}_i, \mathcal{P}_i') \tag{35}$$

Assume now that

$$\Delta = \Delta(\Pi') = D(\mathcal{P}_1, \mathcal{P}_1') \leq \frac{M}{2}, \tag{36}$$

and in particular $D(\mathcal{P}_i, \mathcal{P}_i') \leq \Delta$. Let $\mathcal{P}_1 = \{P_1, \ldots, P_K\}$ where $P_k = \Pi_1^{-1}(k)$. Let $A$ be a set of maximal size such that $\mathcal{P}_1^A = \mathcal{P}_1'^A$. After relabeling we can assume that $\mathcal{P}_1' = \{P_1', \ldots, P_K'\}$ and

$P_k \cap A = P'_k \cap A$. By composing $\Pi'_1$ with a permutation we can also assume that $P'_k = \Pi'^{-1}_1(k)$. Moreover,

$$|P'_k \cap A| = |P_k \cap A| \geq |P_k| - |A^c| \geq |P_k| - |P_k|/2 = |P_k|/2. \tag{37}$$

The same applies to all partitions $\mathcal{P}_i$ and $\mathcal{P}'_i$. Next, we claim that if $f = g \circ \Pi$ and $f' = g' \circ \Pi'$ agree on all samples then with probability $K^I e^{-\frac{S}{2^I L}}$ over the randomness of the samples $g = g'$. Define $E_{\boldsymbol{k}} := \Pi'^{-1}(\boldsymbol{k}) \cap \Pi^{-1}(\boldsymbol{k})$. Using (37) and the assumption on $\Pi$ we have that for $\boldsymbol{k} \in [K]^N$

$$|E_{\boldsymbol{k}}| := |\Pi'^{-1}(\boldsymbol{k}) \cap \Pi^{-1}(\boldsymbol{k})| \geq 2^{-I} |\Pi^{-1}(\boldsymbol{k})| \geq \frac{N^I}{2^I L}. \tag{38}$$

The probability that none of the $S$ samples is in $E_{\boldsymbol{k}}$ can then be bounded as follows

$$\mathbb{P}(\mathcal{D} \cap E_{\boldsymbol{k}} = \emptyset) = \left(1 - \frac{|E_{\boldsymbol{k}}|}{N^I}\right)^S \leq \left(1 - \frac{1}{2^I L}\right)^S \leq e^{-\frac{S}{2^I L}}. \tag{39}$$

Applying the union bound over $\boldsymbol{k}$ we find that

$$\mathbb{P}_{\mathcal{D}}(\Pi' \text{ is consistent with } \mathcal{D} \text{ for some } g' \neq g) \leq K^I e^{-\frac{S}{2^I L}}. \tag{40}$$

Now we bound the probability that $g' = g$ is compatible, i.e., $g \circ \Pi(\boldsymbol{n}^s) = g \circ \Pi'(\boldsymbol{n}^s)$ for all $s \in [S]$. Let now $n_1 \in [N]$. Denote by $E_{n_1} \subset [N]^I$ the set of all vectors $\boldsymbol{n} = (n_1, n_2, \ldots, n_I)$ such that

$$g \circ \Pi(\boldsymbol{n}) \neq g \circ \Pi'(\boldsymbol{n}) \tag{41}$$

We now lower bound the size of $E_{n_1}$ under the assumption that there is $k \in [K]$ such that $n_1 \in P'_k \cap A^c$ (where $A$ is as above a set such that $\mathcal{P}_1^A = \mathcal{P}'^A_1$). Then $\Pi'_1(n_1) = k$ and (by minimality of $A^c$) we find $n_1 \notin P_k$ and thus $\Pi_1 n_1 = \bar{k} \neq k$ for some $\bar{k}$. By assumption, we can find $k_2, \ldots, k_I$ such that $g(k, k_2, \ldots, k_I) \neq g(\bar{k}, k_2, \ldots, k_I)$. Consider now any vector $\boldsymbol{n} = (n_1, n_2, \ldots, n_I)$ such that $n_i \in \Pi_i^{-1}(k_i) \cap \Pi'^{-1}_i(k_i)$. Then, for any such $\boldsymbol{n}$

$$g \circ \Pi(\boldsymbol{n}) = g(\bar{k}, k_2, \ldots, k_I) \neq g(k, k_2, \ldots, k_I) = g \circ \Pi'(\boldsymbol{n}) \tag{42}$$

we conclude that all such $\boldsymbol{n}$ are in $E_{n_1}$ By assumption we have

$$\left| [N] \times \prod_{i=2}^{I} (\Pi_i^{-1}(k_i) \cap \Pi'^{-1}_i(k_i)) \right| \geq \frac{N^I}{2^I L}. \tag{43}$$

This implies that

$$|E_{n_1}| \geq \frac{N^{I-1}}{2^I L} \tag{44}$$

and therefore

$$\mathbb{P}(E_{n_1}) \geq \frac{1}{2^I L N}. \tag{45}$$

The sets $E_{n_1}$ are disjoint so we conclude that $E = \cup_{n_1 \in [N]} E_{n_1}$ satisfies

$$\mathbb{P}(E) = \sum_{n_1 \in [N]} \mathbb{P}(E_{n_1}) \geq |A^c| \frac{1}{2^I L N} = \frac{\Delta}{2^I L N}. \tag{46}$$

Now we can upper bound the probability that $\mathcal{D}$ is compatible with $\Pi'$ and $g$ by

$$\mathbb{P}_{\mathcal{D}}(\mathcal{D} \text{ is compatible with } \Pi', g) \leq \mathbb{P}_{\mathcal{D}}(E \cap \mathcal{D} = \emptyset) \leq \left(1 - \frac{\Delta}{2^I L N}\right)^S \leq e^{-\frac{S\Delta}{2^I L N}}. \tag{47}$$

Clearly the same reasoning applies to any other index instead of 1.

The next step is to upper bound the number of such candidate partitions. The number of partitions such that $D(\mathcal{P}_1, \mathcal{P}'_1) \leq \Delta$ can be bounded by $(NK)^\Delta$, i.e., we $\Delta$ times select one of $N$ tokens and assign it to another (or the same) class. This bound can be applied to all indices $1 \leq i \leq I$ and by the union bound, we find that

$$|\{(\mathcal{P}'_1, \ldots, \mathcal{P}'_I) : \max D(\mathcal{P}'_i, \mathcal{P}_i) \leq \Delta\}| \leq (NK)^{I\Delta}. \tag{48}$$

Thus we can bound the the probability that any of the maps $\Pi'$ (there are more such maps than partitions because there is a label assigned to each class, but consistency with the data depends only on the underlying permutation) with $0 < \Delta(\Pi') \leq M/2$ is consistent with the data is bounded by

$$\mathbb{P}_{\mathcal{D}}(\exists \Pi' \text{ with } \Delta(\Pi') \leq M/2 \text{ consistent with } \mathcal{D}) \leq I \sum_{\Delta=1}^{M/2} (NK)^{I\Delta} \left( e^{-\frac{S\Delta}{2^I LN}} + K^I e^{-\frac{S}{2^I L}} \right). \quad (49)$$

Here the first term corresponds to the upper bound for the probability that (after permutation) $g = g'$ is compatible with the data and the second term bounds the probability that any other function is compatible with the data and $I$ accounts for the fact that any $i \in [I]$ can be $\arg\max_i D(\mathcal{P}_i, \mathcal{P}'_i)$.

We now consider the remaining case that there is an index $i \in [I]$ such that $D(\mathcal{P}_i, \mathcal{P}'_i) \geq M/2$ where we can assume w.l.o.g. that $i = 1$. As above, let $\mathcal{P}_1 = \{P_1, \ldots, P_K\}$ and $\mathcal{P}'_1 = \{P'_1, \ldots, P'_K\}$ such that $P_k \cap A = P'_k \cap A$ and $A$ is a set of maximal size. We now claim that we can find an index $k \neq \bar{k}$ such that

$$|P'_k \cap P_k| \geq \frac{M}{2K} \quad \text{and} \quad |P'_k \cap P_{\bar{k}}| \geq \frac{M}{2K^2}. \quad (50)$$

Indeed, first assume that there is an index $k$ such that $|P'_k \cap P_k| \leq M/K$. Since $|P'_k| \geq M$ there is another index $\bar{k}$ such that $|P'_{\bar{k}} \cap P_k| \geq M/K$. By maximality of $A$ we have $P_k \cap A = P'_k \cap A = P_k \cap P'_k$ and moreover

$$|P'_k \cap P_k| + |P'_{\bar{k}} \cap P_{\bar{k}}| \geq |P'_{\bar{k}} \cap P_k| + |P'_k \cap P_{\bar{k}}| \quad (51)$$

because otherwise exchanging $P'_k$ and $P'_{\bar{k}}$ would allow picking a larger $A$. By our assumption, we conclude from here that

$$|P_{\bar{k}} \cap P'_k| \geq |P_k \cap P'_k| + |P_{\bar{k}} \cap P'_k| - |P_k \cap P'_k| \geq \frac{M}{K} - \frac{M}{2K} = \frac{M}{2K}. \quad (52)$$

Thus, in this case $|P'_{\bar{k}} \cap P_{\bar{k}}| \geq M/(2K)$ and $|P'_{\bar{k}} \cap P_k| \geq M/K$ and (50) holds. Assume now to the contrary that there is no index $k$ such that $|P'_k \cap P_k| < M/K$. The assumption $|A^c| \geq M/2$ implies that there is $k$ such that $|A^c \cap P'_k| \geq M/(2K)$ and by minimality of $A^c$ we find $A^c \cap P'_k \cap P_k = \emptyset$ and thus there is $\bar{k} \neq k$ such that $|P'_k \cap P_{\bar{k}}| = |A^c \cap P'_k \cap P_{\bar{k}}| \geq M/(2K^2)$. This finishes the proof of (50).

We now fix $k$ and $\bar{k}$ such that $|P_k \cap P'_k| \geq M/(2K^2)$ and $|P_{\bar{k}} \cap P'_k| \geq M/(2K^2)$. Then, by assumption, we can find $k_2, \ldots, k_I \in [K]$ such that $\boldsymbol{k} = (k, k_2, \ldots, k_I)$ and $\bar{\boldsymbol{k}} = (\bar{k}, k_2, \ldots, k_I)$ satisfy $g(\boldsymbol{k}) \neq g(\bar{\boldsymbol{k}})$.

Moreover $|\Pi^{-1}(\boldsymbol{k})| \geq N^I/L$. Now for every $k_i$ there is $k'_i$ such that $|\Pi_i^{-1}(k_i) \cap \Pi'^{-1}_i(k'_i)| \geq K^{-1}|\Pi_i^{-1}(k_i)|$. Thus, there is $\boldsymbol{k}' \in [K]^I$ such that

$$\prod_{i=2}^{I} \left| \Pi'^{-1}_i(k'_i) \cap \Pi_i^{-1}(k_i) \right| \geq K^{-I+1} \left| \prod_{i=2}^{I} \Pi_i^{-1}(k_i) \right| \geq \frac{N^I}{K^{I-1}L}. \quad (53)$$

Define $A_i = \Pi'^{-1}_i(k'_i) \cap \Pi_i^{-1}(k_i)$ and $A_1 = P_k \cap P'_k$ and $\bar{A}_1 = P_{\bar{k}} \cap P'_k$. Define $\boldsymbol{A} = A_1 \times A_2 \times \ldots \times A_I$ and $\bar{\boldsymbol{A}} = \bar{A}_1 \times A_2 \times \ldots \times A_I$. Note that by (50)

$$|\boldsymbol{A}|, |\bar{\boldsymbol{A}}| \geq N^I \frac{M}{2K^{I+1}L}. \quad (54)$$

Clearly $\Pi'$ is constant (and equal to $(k, k'_2, \ldots, k'_I)$) on $\boldsymbol{A} \cap \bar{\boldsymbol{A}}$ and therefore also $g' \circ \Pi'$ is constant for any $g'$. On the other hand, $\Pi(\boldsymbol{A}) = \{\boldsymbol{k}\}$ is constant and $\Pi(\bar{\boldsymbol{A}}) = \bar{\boldsymbol{k}}$ is constant but

$$g \circ \Pi(\boldsymbol{n}) = g(\boldsymbol{k}) \neq g(\bar{\boldsymbol{k}}) = g \circ \Pi(\bar{\boldsymbol{n}}) \quad (55)$$

for $\bar{n} \in \boldsymbol{A}$ and $\bar{n} \in \bar{\boldsymbol{A}}$. Now $\Pi'$ can only be consistent with the data (for any $g'$) if there is no sample in $\boldsymbol{A}$ or no sample in $\bar{\boldsymbol{A}}$, i.e., and thus

$$\mathbb{P}_{\mathcal{D}}((\Pi', g') \text{ consistent with } \mathcal{D} \text{ for a } g') \leq \mathbb{P}_{\mathcal{D}}(\mathcal{D} \cap \boldsymbol{A} = \emptyset \text{ or } \mathcal{D} \cap \bar{\boldsymbol{A}} = \emptyset)$$
$$\leq \mathbb{P}_{\mathcal{D}}(\mathcal{D} \cap \boldsymbol{A} = \emptyset) + \mathbb{P}_{\mathcal{D}}(\mathcal{D} \cap \bar{\boldsymbol{A}} = \emptyset) \quad (56)$$

where we used the union bound in the last step. Then we find using (54) and (56)

$$\mathbb{P}_{\mathcal{D}}\left((\Pi', g') \text{ consistent with } \mathcal{D} \text{ for a } g'\right) \leq 2\left(1 - \frac{M}{2K^{I+1}L}\right)^S \leq 2e^{-\frac{SM}{2K^{I+1}L}}. \qquad (57)$$

Finally, we can bound the number of partitions by the number of maps $\Pi_i : [N] \to [K]$ which implies that

$$|\{\mathcal{P}_1, \ldots, \mathcal{P}_I : \mathcal{P}_i \text{ partition of } [N] \text{ in at most } K \text{ classes}\}| \leq (K^N)^I = K^{NI}. \qquad (58)$$

So we get the upper bound

$$\mathbb{P}_{\mathcal{D}}(\exists \Pi' \text{ such that } \Delta(\Pi') \geq M/2 \text{ and } \Pi' \text{ compatible with } \mathcal{D}) \leq 2IK^{NI}e^{-\frac{MS}{K^{I+1}L}}. \qquad (59)$$

Combining (49) and (59) and using the union bound we find

$$\mathbb{P}_{\mathcal{D}}((\Pi, g) \text{ not identifiable up to permutations}) \leq \mathbb{P}_{\mathcal{D}}(\exists \Pi' \text{ with } \Delta(\Pi') > 0 \text{ is compatible with } \mathcal{D})$$

$$\leq 2K^{NI}Ie^{-\frac{MS}{K^{I+1}L}} + I\sum_{\Delta=1}^{M/2}(NK)^{I\Delta}\left(e^{-\frac{S\Delta}{2^I LN}} + K^Ie^{-\frac{S}{2^I L}}\right). \qquad (60)$$

Note that clearly the assumptions imply $M \leq N/K < N$ and thus $e^{-S/(2^I L)} \leq e^{-S\Delta/(2^I LN)}$ for $\Delta \leq M/2$. We then find (using the simple bound $IK^I \leq (2K)^I \leq (NK)^{I\Delta}$ that

$$\sum_{\Delta=1}^{M/2}(NK)^{I\Delta}\left(e^{-\frac{S\Delta}{2^I LN}} + K^Ie^{-\frac{S}{2^I L}}\right) \leq 2\sum_{\Delta=1}^{M/2}e^{\left(2\ln(NK)I - \frac{S}{2^I LN}\right)\Delta} \leq 4e^{-\eta} \qquad (61)$$

for

$$S \geq \max(4\ln(NK)I2^I LN, 2^{I+1}\eta LN) \qquad (62)$$

and $\eta > 2$ where we bound the geometric sum by twice its largest term. The first summand in (60) can be bounded by

$$2IK^{NI}e^{-\frac{MS}{K^{I+1}L}} = 2e^{2\ln(K)NI - \frac{MS}{K^{I+1}L}} \leq 2e^{-\eta} \qquad (63)$$

for

$$S \geq \max\left(\frac{4\ln(K)IK^{I+1}LN}{M}, \frac{2\eta K^{I+1}L}{M}\right). \qquad (64)$$

$\square$

The proof of Theorem 1 is now a direct consequence of the previous result because when assuming that a uniformly random map $\Pi$ is chosen we can estimate the quantities $M$ and $L$ in the previous theorem with high probability.

*Proof of Theorem 1.* We observe that $|\pi_i^{-1}(k_i)| \sim \text{Bin}(N, K^{-1})$. Applying a Chernoff bound on the tail of the binomial variable we obtain

$$\mathbb{P}\left(|\pi_i^{-1}(k_i)| < \frac{N}{K}(1 - \tfrac{1}{2})\right) \leq e^{-\frac{1}{2^2}\frac{N/K}{2}} = e^{-\frac{N}{8K}}. \qquad (65)$$

By the union bound we get

$$\mathbb{P}\left(\min_{i\in[I], k\in[K]}|\pi_i^{-1}(k)| < N/(2K)\right) \leq K^Ie^{-\frac{N}{8K}} = e^{\ln(K)I - \frac{N}{8K}} \leq e^{-\eta} \qquad (66)$$

if $N \geq N_0(\eta, K, I)$. Note that $M \geq N/(2K)$ implies that $L \leq (N/M)^I \leq (2K)^I$. Assuming that $\Pi$ is such that the bounds for $M$ and $L$ hold we can apply Theorem 6 and find that for $N \geq N_0$ sufficiently large (depending on $I, K, \eta$) and

$$S \geq 2^{2I+3}IK^I N \ln(N) \qquad (67)$$

(we bounded $\ln(NK) \leq 2\ln(N)$) the maps $\Pi$ and $g$ are identifiable with probability at least $1 - e^{-\eta}$. Here we used that as $N \to \infty$ the term indicated above is dominating in (31). Now the union bound over the bad events ends the proof. $\square$

*Proof of Theorem 2.* As before, we note $\mathcal{D} = \{\boldsymbol{n}^1, \ldots, \boldsymbol{n}^S\}$. Assume there is $n_1$ such that $\Pi_1(n_1) = 1$ and such that

$$\mathcal{D} \cap \{n_1\} \times \Pi_2^{-1}(1) \times \ldots \times \Pi_I^{-1}(1) = \emptyset. \tag{68}$$

Then $\Pi' \neq \Pi$ is compatible with $\mathcal{D}$ where $\Pi_i' = \Pi_i$ for $i \geq 2$ and $\Pi_1(n) = \Pi_1'(n)$ for $n \neq n_1$ and $\Pi_1'(n_1) = 2 \neq 1 = \Pi_1(n_1)$ (by assumption on $g$). Let us denote

$$A_{n_1} = \{n_1\} \times \Pi_2^{-1}(1) \times \ldots \times \Pi_I^{-1}(1) \subset [N]^I. \tag{69}$$

Then

$$\mathbb{P}(\boldsymbol{n}^s \in A_{n_1}) = \frac{1}{NK^{I-1}}. \tag{70}$$

To estimate the probability of the event $\bigcup_{n_1} \{A_{n_1} \cap \mathcal{D} = \emptyset\}$ we use Poissonization to make the events independent. Consider datasets $\tilde{\mathcal{D}}$ whose distribution is generated by first sampling $\bar{S} \sim \text{Poi}(2S)$ and then conditional on $\bar{S}$ sample a dataset $\hat{\mathcal{D}}$ as before, i.e., $\boldsymbol{n}^s \sim \mathcal{U}([N]^I)$ for $s \in [\bar{S}]$. Then we find that

$$|A_{n_1} \cap \tilde{\mathcal{D}}| \sim \text{Poi}\left(\frac{2S}{NK^{I-1}}\right) \tag{71}$$

and those events are independent for $n_1 \neq n_1'$. Thus, we find that

$$\mathbb{P}_{\tilde{\mathcal{D}}}(|A_{n_1} \cap \tilde{\mathcal{D}}| = 0) = e^{-\frac{2S}{NK^{I-1}}}. \tag{72}$$

By independence we now find that

$$\mathbb{P}_{\tilde{\mathcal{D}}}\left(\bigcap_{n_1 \in \Pi_1^{-1}(1)} \{A_{n_1} \cap \tilde{\mathcal{D}} \neq \emptyset\}\right) = \prod_{n_1 \in \Pi_1^{-1}(1)} \mathbb{P}_{\tilde{\mathcal{D}}}\left(A_{n_1} \cap \tilde{\mathcal{D}} \neq \emptyset\right)$$

$$= \left(1 - e^{-\frac{2S}{NK^{I-1}}}\right)^{N/K}$$

$$\leq \left(1 - e^{-\frac{1}{2}\ln(N/K)}\right)^{N/K} = \left(1 - \frac{\sqrt{K}}{\sqrt{N}}\right)^{N/K} \leq e^{-\sqrt{N/K}} \tag{73}$$

where we used the upper bound for $S$ and $(1 - x) \leq e^{-x}$. Assume $\mathcal{D}_s \sim \mathcal{U}([N]^I)^s$ and define

$$p_s = \mathbb{P}_{\mathcal{D}_s}\left(\bigcap_{n_1 \in \Pi_1^{-1}(1)} \{A_{n_1} \cap \mathcal{D}_s \neq \emptyset\}\right). \tag{74}$$

Note that $p_S$ is an upper bound on the probability that $\Pi$ is identifiable as explained above. We have shown that

$$\mathbb{E}(p_{\text{Poi}(2S)}) \leq e^{-\sqrt{N/K}}. \tag{75}$$

Clearly $p_s$ is decreasing in $s$. This implies that

$$e^{-\sqrt{N/K}} \geq \mathbb{E}(p_{\text{Poi}(2S)}) \geq p_S \cdot \mathbb{P}(\text{Poi}(2S) \geq S). \tag{76}$$

A Chernoff bound for the Poisson distribution reads

$$\mathbb{P}(\text{Poi}(\lambda) \leq x) \leq \frac{(e\lambda)^x e^{-\lambda}}{x^x} \tag{77}$$

which implies with $\lambda = 2S$ and $x = S$ that

$$\mathbb{P}(\text{Poi}(2S) \leq S) \leq \frac{(2Se)^S e^{-2S}}{S^S} = \left(\frac{2}{e}\right)^S. \tag{78}$$

We thus conclude that

$$p_S \leq 2e^{-\sqrt{N/K}} \tag{79}$$

for $S \geq 3$ (implying $1 - (2/e)^S \geq 1/2$). $\qquad\square$

# C Proofs of Complexity-Theoretic Analysis and Constraint Satisfaction Reduction

In this section, we provide the proof of Theorem 3 and a general reduction to a constraint satisfaction problem. We start with the proof of Theorem 3. An important property we will use frequently in the proof is that $T$ has the property that for all values $x_1$ and $x_2$ there is $x_3(x_1, x_2)$ such that $T(x_1, x_2, x_3(x_1, x_2)) = 0$. Indeed, $x_3(x_1, x_2) = 0$ except for $x_1 = x_2 = 1$ where $x_3(1, 1) = 0$ has this property.

*Proof.* We reduce the problem to 3SAT. We consider an arbitrary formula

$$\phi(x) = \bigwedge C_i(x) \tag{80}$$

on $\ell$ variables with $m$ clauses $C_i$. We denote $f = g \circ \Pi$. To improve readability, we consider a symbol set $\mathcal{S}$ instead of $[N]$ which is used for all 3 slots. We proceed in three steps. First we show that we can write down a set of equations that ensures that two symbols $s_0, s_1 \in \mathcal{S}$ satisfy $\Pi_i(s_j) = j$ for all $i$ (if $f = g \circ \Pi$). Suppose the following relations hold

$$\begin{aligned} f(s_0, s_0, s_1) = f(s_0, s_1, s_0) = f(s_1, s_0, s_0) = 0, \\ f(s_0, s_1, s_1) = f(s_1, s_0, s_1) = f(s_1, s_1, s_0) = 1. \end{aligned} \tag{81}$$

Then it is easy to conclude that $\Pi_i(s_j) = j$ holds. Indeed, suppose that $\Pi_1(s_1) = 0$. Then the lower part of the previous display combined with the definition of $g$ imply $\Pi_2(s_0) = \Pi_3(s_1) = \Pi_2(s_1) = \Pi_3(s_0) = 1$. But this is a contradiction to $f(s_1, s_0, s_0) = 0$. Thus $\Pi_1(s_1) = 1$ and the same reasoning implies $\Pi_i(s_1) = 1$. Then the second equation in (81) directly implies $\Pi_i(s_0) = 0$.

We now consider a symbol set

$$\mathcal{S} = \{s_0, s_1\} \cup \mathcal{X} \cup \bar{\mathcal{X}} \cup \mathcal{C} \cup \bar{\mathcal{C}} \cup \mathcal{T} \cup \mathcal{T}' \cup \mathcal{F} \tag{82}$$

where

$$\mathcal{X} = \{X_1, \dots, X_\ell\} \tag{83}$$

will encode the variables in the formula $\phi$ and

$$\mathcal{C} = \{C_{1,1}, C_{1,2}, C_{2,1}, C_{2,2}, \dots, C_{m,1}, C_{m,2}\} \tag{84}$$

are auxiliary variables for the clauses. In addition, we need further auxiliary variables

$$\begin{aligned} \bar{\mathcal{X}} &= \{\bar{X}_1, \dots, \bar{X}_\ell\}, \\ \bar{\mathcal{C}} &= \{\bar{C}_{1,1}, \bar{C}_{1,2}, \bar{C}_{2,1}, \bar{C}_{2,2}, \dots, \bar{C}_{m,1}, \bar{C}_{m,2}\}, \quad \text{and} \\ \mathcal{O} &= \{o_1, \dots, o_m\}. \end{aligned} \tag{85}$$

We now add a set of equations that will ensure that for some value $x_i \in \{0, 1\}$

$$\begin{aligned} \Pi_1(X_i) = \Pi_2(X_i) = \Pi_3(X_i) = x_i \\ \Pi_1(\bar{X}_i) = \Pi_2(\bar{X}_i) = \Pi_3(\bar{X}_i) = 1 - x_i. \end{aligned} \tag{86}$$

This can be achieved by adding the following relations for all $i$

$$\begin{aligned} f(X_i, \bar{X}_i, s_1) = f(\bar{X}_i, X_i, s_1) = f(s_1, \bar{X}_i, X_i) = 1, \\ f(s_1, X_i, \bar{X}_i) = f(X_i, s_1, \bar{X}_i) = f(\bar{X}_i, s_1, X_i) = 1. \end{aligned} \tag{87}$$

Note that the these relations indeed imply that $\Pi_a(X_i) \neq \Pi_b(\bar{X}_i)$ for $a \neq b$, $a, b \in \{1, 2, 3\}$. This implies then that (86) holds. We add the similar relations for $C_{i,k}$ which again ensure that

$$\Pi_j(C_{i,k}) = \Pi_{j'}(C_{i,k}) \neq \Pi_j(\bar{C}_{i,k}). \tag{88}$$

Now we encode the clauses of the formula. Consider a clause $C_l$ of the form $x_{i_1} \vee x_{i_2} \vee x_{i_3}$. Then we add the relations

$$\begin{aligned} f(\bar{X}_{i_1}, \bar{X}_{i_2}, C_{l,1}) &= 0 \\ f(\bar{X}_{i_2}, \bar{X}_{i_3}, C_{l,2}) &= 0 \\ f(\bar{C}_{l,1}, \bar{C}_{l,2}, o_l) &= 1. \end{aligned} \tag{89}$$

For a given choice of $\Pi$ we set $x_i = \Pi_1(X_i)$. We now claim that the relations in the last display can hold if and only if $x_{i_1} \vee x_{i_2} \vee x_{i_3}$ evaluates to true. Suppose the formula evaluates to false. Then

$$\Pi_j(X_{i_1}) = \Pi_j(X_{i_2}) = \Pi_j(X_{i_3}) = 0 \tag{90}$$

and therefore

$$\Pi_j(\bar{X}_{i_1}) = \Pi_j(\bar{X}_{i_2}) = \Pi_j(\bar{X}_{i_3}) = 1. \tag{91}$$

Then relation (89) implies

$$\Pi_3(C_{l,i}) = 1 \tag{92}$$

for $i = 1, 2$. Using (88) we find that $\Pi_1(\bar{C}_{l,1}) = \Pi_2(\bar{C}_{l,2}) = 0$ which implies

$$f(\bar{C}_{l,1}, \bar{C}_{l,2}, o_l) = f(0, 0, o_l) = 0 \tag{93}$$

for any $\Pi_3(o_l)$. Therefore, the equations cannot all hold.

Suppose to the contrary that the formula evaluates to true. Let us first assume that $\Pi_1(X_{i_1}) = 1$. Then $\Pi_1(\bar{X}_{i_1}) = 0$ and we can set $\Pi_3(C_{l,1}) = 0$ which ensures that the first equation is satisfied (for any $\Pi_2(\bar{X}_{i_2})$). By definition of $g$ we can also find a value $c_{l,2}$ such that for $c_{l,2} = \Pi_3(C_{l,2})$ the relation $f(\bar{X}_{i_2}, \bar{X}_{i_3}, C_{l,2}) = f(x_{i_2}, x_{i_3}, c_{l,2}) = 0$ holds. Now $\Pi_1(\bar{C}_{l,1}) = 1 - \Pi_3(C_{l,1}) = 1$ ensures that for any value of $\bar{c}_{l,2} = \Pi_2(\bar{C}_{l,2})$ we can choose $\Pi_3(o_l)$ such that the relation $f(\bar{C}_{l,1}, \bar{C}_{l,2}, o_l) = f(1, \bar{c}_{l,2}, o_l) = 1$ holds. The same reasoning applies for $\Pi_1(X_{i_3}) = 1$ and a similar argument applies if $\Pi_1(X_{i_2}) = 1$. For clauses containing negations the same construction works except that $\bar{X}_{i_j}$ has to be replaced by $X_{i_j}$ for the negated variables in equation (89).

Putting everything together, we have shown that for a given formula $\phi$ there is a set of relations of the form $f(\boldsymbol{n}^s) = t^s$ where $\boldsymbol{n}^s \in \mathcal{S}^3$ given by (81), (87) (plus similar equations for $C_{l,i}$), and (89) which has the following properties. For an assignment $x = (x_1, \ldots, x_n) \in \{0, 1\}^n$ such that $\phi(x) = 1$ evaluates to true, there is a map $\Pi$ such that for all $s$ the relations $t^s = f(\boldsymbol{n}^s)$ hold and where $\Pi_1(X_i) = x_i$. On the other hand, if for some $\Pi$ all the relations $t^s = f(\boldsymbol{n}^s)$ hold, then the Boolean variables $x_i = \Pi(X_i)$ will satisfy the formula $\phi$. This ends the proof. $\qquad\square$

We now show how the problem of finding $\Pi$ and $g$ can be generally expressed as a constraint satisfaction problem. Recall that we want to find (if they exist) for a given $K$ a projection map $\Pi : [N]^I \to [K]^I$ and $g : K^I \to \mathbb{R}$ such that for all given samples $(\boldsymbol{n}^s, t^s)$ the relation $t^s = g \circ \Pi(\boldsymbol{n}^s)$ holds. The general strategy is to introduce Boolean variables encoding the maps $\Pi$ and $g$ and then express all conditions as suitable constraints for these variables. We first introduce Boolean variables $t_{kn}^i$ for $i \in [I]$, $k \in [K]$, and $n \in [N]$ which are 1 if the token $n$ is in cluster $k$, i.e., $\Pi_i(n) = k$ and 0 otherwise. Then the expression

$$t_{1n}^i \vee t_{2n}^i \vee \ldots \vee t_{kn}^i \tag{94}$$

for $i \in [I]$ and $n \in [N]$ is true if and only if $n$ is assigned to to at least one cluster. In addition, we consider variables $r_{\boldsymbol{k}v}$ for every $\boldsymbol{k} \in [K]^I$ and $v \in \mathrm{Im}(f)$ in the (finite) image of $f$. These variables shall encode whether $g(\boldsymbol{k}) = v$ is true or not. Then the constraints

$$\neg r_{\boldsymbol{k}v} \vee \neg r_{\boldsymbol{k}v'} \tag{95}$$

for $v \neq v'$ and all $\boldsymbol{k}$ ensure that each cluster is assigned at most one value $v$. Finally, we add for every datapoint $(\boldsymbol{n}^s, f(\boldsymbol{n}^s))$ and every $\boldsymbol{k}$ the constraint

$$\neg t_{\boldsymbol{k}_1 \boldsymbol{n}_1^s}^1 \vee \ldots \vee \neg t_{\boldsymbol{k}_I \boldsymbol{n}_I^s}^I \vee r_{\boldsymbol{k}f(\boldsymbol{n}^s)}. \tag{96}$$

This ensures that if $\Pi(\boldsymbol{n}^s) = \boldsymbol{k}$ then this cluster must be assigned the value $f(\boldsymbol{n}^s)$.

We then consider the constraint satisfaction problem consisting of the $\bigwedge$ of all the conditions in (94), (95), and (96). Then any satisfying assignment gives rise to a map $\Pi$ and $g$ such that $g \circ \Pi(\boldsymbol{n}^2) = t^s$. Indeed, we set $g(\boldsymbol{k}) = v$ if $s_{\boldsymbol{k}v} = 1$ if such a $v$ exists and arbitrarily otherwise. Note that for every satisfying assignment and every $\boldsymbol{k}$ there is at most one such $v$ so $g$ is well-defined. Moreover, we set $\Pi_i(n) = k$ for any $k$ such that $t_{nk}^i = 1$ and at least one such $k$ exists. On the other hand, we can easily construct a satisfying assignment given $\Pi$ and $g$ so that there are no solutions $\Pi$ and $g$ if the constraint satisfaction problem has no solution.

Note that we did not ensure that each token is assigned only to a single cluster but this can be achieved in post-processing or by adding additional constraints (such as $\neg t_{k_1 n}^i \vee \neg t_{k_1 n}^i$).

## D  Taylor Expansion of the Loss Gradient

The goal of this and the following section is to lay the groundwork for the proof of the main results.

The general strategy of our proofs is to Taylor expand the loss of each term around the mean token embedding $w(i, \Pi_i(\boldsymbol{n}_i))$ to first order plus a remainder term. We can then extract the dominating terms of these expansions using concentration results for the datapoint statistics. It turns out that the linearized dynamics has favorable properties, while the remainder terms can be bounded. In this section, we derive the expansion of the loss gradient while the required bounds, in particular Theorem 7 and Lemma 1, are derived in the next section, Appendix E (they rely on concentration results which are deferred to Appendix G).

As pointed out above, the goal of this section is to Taylor expand the sample loss for one sample, where we expand the loss around the point $\boldsymbol{w}(\Pi(\boldsymbol{n}))$. Let us first introduce some notation. Recall that we denoted by $\boldsymbol{v}(\boldsymbol{n}) = (v(1, \boldsymbol{n}_1), \ldots, v(I, \boldsymbol{n}_I))$ and by $\bar{\boldsymbol{v}}(i)$ the concatenation of the embeddings $(v(i, n))_{n \in [N]}$. We define similarly

$$\boldsymbol{\delta}(\boldsymbol{n}) = (\delta(1, \boldsymbol{n}_1), \ldots, \delta(I, \boldsymbol{n}_I)) \tag{97}$$

and we denote by $\bar{\boldsymbol{\delta}}(i) \in \mathbb{R}^{DN}$ the concatenation of $\delta(i, 1), \ldots, \delta(i, n)$, i.e.,

$$\bar{\boldsymbol{\delta}}(i) = (\delta(i, n))_{n \in [N]} \tag{98}$$

and $\bar{\boldsymbol{\delta}} \in \mathbb{R}^{IDN}$ as the concatenation of $\bar{\boldsymbol{\delta}}(1), \ldots, \bar{\boldsymbol{\delta}}(I)$.

Consider a data-point $\boldsymbol{n}$ with $\Pi(\boldsymbol{n}) = \boldsymbol{k}$. We now consider the derivative of the mean squared error of this term, i.e.,

$$L(\boldsymbol{n}) = \frac{1}{2}(\hat{f}(\boldsymbol{n}) - f(\boldsymbol{n}))^2 = \frac{1}{2}(\hat{f}(\boldsymbol{n}) - g(\boldsymbol{k}))^2. \tag{99}$$

We use Greek-indices for the latent space dimension. Fix an $i \in [I]$ as the slot with respect to which we take the derivative. We then find for $\alpha \in [D]$

$$\frac{\mathrm{d}}{\mathrm{d}v_\alpha(i, \boldsymbol{n}_i)} \frac{1}{2} \left( \hat{f}(\boldsymbol{v}(\boldsymbol{n})) - f(\boldsymbol{k}) \right)^2 = \frac{\mathrm{d}}{\mathrm{d}v_\alpha(i, \boldsymbol{n}_i)} \frac{1}{2} \left( \hat{f}(\boldsymbol{v}(\boldsymbol{n})) - f(\boldsymbol{k}) \right)^2$$

$$= \left( \frac{\mathrm{d}}{\mathrm{d}v_\alpha(i, \boldsymbol{n}_i)} \hat{f}(\boldsymbol{v}(\boldsymbol{n})) \right) \cdot \left( \hat{f}(\boldsymbol{v}(\boldsymbol{n})) - f(\boldsymbol{k}) \right) =: h(\boldsymbol{v}(\boldsymbol{n})). \tag{100}$$

Here we introduced the shorthand $h(\boldsymbol{v}(\boldsymbol{n}))$ for this function which also depends on $i$, $\boldsymbol{k}$, and $\alpha$. Now we estimate this function by Taylor expanding it around the point

$$\boldsymbol{w}(\boldsymbol{k}) = (w(1, \boldsymbol{k}_1), \ldots, w(I, \boldsymbol{k}_I))). \tag{101}$$

Then we find the following expansion to second order

$$h(\boldsymbol{v}(\boldsymbol{n})) = h(\boldsymbol{w}(\boldsymbol{k})) + \sum_{j \in [I]} \sum_{\beta \in [D]} \frac{\mathrm{d}}{\mathrm{d}w_\beta(j, \boldsymbol{k}_j)} h(\boldsymbol{w}(\boldsymbol{k})) \delta_\beta(j, \boldsymbol{n}_j)$$

$$+ \sum_{j_1, j_2 \in [I]} \sum_{\beta_1 \beta_2 \in [D]} R^{n,i,\alpha}_{(j_1,\beta_1),(j_2,\beta_2)}(\boldsymbol{w}(\boldsymbol{k})) \delta_{\beta_1}(j_1, \boldsymbol{n}_{j_1}) \delta_{\beta_2}(j_2, \boldsymbol{n}_{j_2}) \tag{102}$$

where $R_{(j_1,\beta_1),(j_2,\beta_2)}$ denotes the remainder terms. We remark that if we assume that $v(i, n) \in \Omega$ for some convex set $\Omega$ and all $i \in [I]$, $n \in [N]$ then (by convexity of $\Omega$) also $w(i, \boldsymbol{k}_i) \in \Omega$ and

$$|R^{n,i,\alpha}_{(j_1,\beta_1),(j_2,\beta_2)}| \leq \max_{\boldsymbol{v} \in \Omega^I} \frac{1}{2} \left| \frac{\mathrm{d}}{\mathrm{d}v_{\beta_1}(j_1, \boldsymbol{n}_{j_1})} \frac{\mathrm{d}}{\mathrm{d}v_{\beta_2}(j_2, \boldsymbol{n}_{j_2})} h(\boldsymbol{v}) \right|. \tag{103}$$

Let us now (abusing notation again, it will be clear from context which function we refer to) write

$$L(\boldsymbol{k}) = \frac{1}{2}(\hat{f}(\boldsymbol{w}(\boldsymbol{k})) - g(\boldsymbol{k}))^2. \tag{104}$$

We introduce some more quantities to get a concise representation of the Taylor expansion. We define the matrices $D_{i,j}L(\boldsymbol{k}) \in \mathbb{R}^{D \times D}$ containing the derivatives of a function $L$ with respect to $w(j_1, \boldsymbol{n}_{j_1})$ and $w(j_2, \boldsymbol{n}_{j_2})$, i.e., we consider

$$(D_{j_1,j_2}L(\boldsymbol{k}))_{\alpha,\beta} = \frac{\mathrm{d}}{\mathrm{d}w_\alpha(j_1, \boldsymbol{k}_{j_1})} \frac{\mathrm{d}}{\mathrm{d}w_\beta(j_2, \boldsymbol{k}_{j_2})} L(\boldsymbol{k}). \tag{105}$$

Similarly we define the vector $D_j L(\boldsymbol{k}) \in \mathbb{R}^D$ by

$$(D_j L(\boldsymbol{k}))_\alpha = \frac{\mathrm{d}}{\mathrm{d}w_\alpha(j, \boldsymbol{k}_j)} L(\boldsymbol{k}). \tag{106}$$

For the diagonal entries $D_{j,j}$ we need a more fine-grained decomposition. Note that by the product rule we have

$$\begin{aligned}
\frac{\mathrm{d}}{\mathrm{d}w_\alpha(j, \boldsymbol{k}_j)} \frac{\mathrm{d}}{\mathrm{d}w_\beta(j, \boldsymbol{k}_j)} L(\boldsymbol{k}) =& \left( \frac{\mathrm{d}}{\mathrm{d}w_\alpha(j, \boldsymbol{k}_j)} \frac{\mathrm{d}}{\mathrm{d}w_\beta(j, \boldsymbol{k}_j)} \hat{f}(\boldsymbol{w}(\boldsymbol{k})) \right) \cdot \left( \hat{f}(\boldsymbol{w}(\boldsymbol{k})) - g(\boldsymbol{k}) \right) \\
&+ \left( \frac{\mathrm{d}}{\mathrm{d}w_\alpha(i, \boldsymbol{k}_i)} \hat{f}(\boldsymbol{w}(\boldsymbol{k})) \right) \left( \frac{\mathrm{d}}{\mathrm{d}w_\beta(i, \boldsymbol{k}_i)} \hat{f}(\boldsymbol{w}(\boldsymbol{k})) \right).
\end{aligned} \tag{107}$$

Using the notations we introduced we can therefore write (recall $\hat{g}(\boldsymbol{k}) = \hat{f}(\boldsymbol{w}(\boldsymbol{k}))$)

$$D_{j,j}L(\boldsymbol{k})) = (D_{j,j}\hat{g}(\boldsymbol{k})) (\hat{g}(\boldsymbol{k}) - g(\boldsymbol{k})) + D_j\hat{g}(\boldsymbol{k}) \otimes D_j\hat{g}(\boldsymbol{k}). \tag{108}$$

Collecting finally all remainder terms as

$$R_\alpha(i, \boldsymbol{n}) = \sum_{j_1, j_2 \in [I]} \sum_{\beta_1 \beta_2 \in [D]} R^{n,i,\alpha}_{(j_1,\beta_1),(j_2,\beta_2)}(\boldsymbol{w}(\boldsymbol{k})) \delta_{\beta_1}(j_1, \boldsymbol{n}_{j_1}) \delta_{\beta_2}(j_2, \boldsymbol{n}_{j_2}) \tag{109}$$

we can thus summarize the Taylor expansion result as follows

$$\begin{aligned}
D_i L(\boldsymbol{n}) =& D_i L(\boldsymbol{k}) + ((D_{i,i}\hat{g}(\boldsymbol{k})) (\hat{g}(\boldsymbol{k}) - g(\boldsymbol{k})) + D_i\hat{g}(\boldsymbol{k}) \otimes D_i\hat{g}(\boldsymbol{k})) \delta(i, \boldsymbol{n}_i) \\
&+ \sum_{j \neq i} (D_{i,j}L(\boldsymbol{k})) \delta(j, \boldsymbol{n}_j) + R(i, \boldsymbol{n})
\end{aligned} \tag{110}$$

Based on the expansion (110) we now want to get an expression for the gradient of the total loss on the entire dataset. We decompose the dataset as follows

$$\mathcal{D}^{\boldsymbol{k},n,i} = \{\boldsymbol{n}^s \in \mathcal{D} | \, \Pi(\boldsymbol{n}^s) = \boldsymbol{k}, \, \boldsymbol{n}_i^s = n\}. \tag{111}$$

Note that if $\Pi_i(n) \neq \boldsymbol{k}_i$ then $\mathcal{D}^{\boldsymbol{k},n,i} = \emptyset$. We also define similarly

$$\mathcal{D}^{n,i} = \{\boldsymbol{n}^s \in \mathcal{D} | \, \boldsymbol{n}_i^s = n\}. \tag{112}$$

We find the following expansion

$$\begin{aligned}
\frac{\mathrm{d}}{\mathrm{d}v(i,n)} \hat{\mathcal{L}}(\bar{\boldsymbol{v}}) =& \frac{\mathrm{d}}{\mathrm{d}v(i,n)} \sum_{s=1}^S L(\boldsymbol{n}^s) = \sum_{\boldsymbol{k} \in [K]^I} \sum_{\boldsymbol{n} \in \mathcal{D}^{\boldsymbol{k},n,i}} \frac{\mathrm{d}}{\mathrm{d}v(i,n)} L(\boldsymbol{n}) \\
=& \sum_{\boldsymbol{k} \in [K]^I} \sum_{\boldsymbol{n} \in \mathcal{D}^{\boldsymbol{k},n,i}} D_i L(\boldsymbol{k}) + ((D_{i,i}\hat{g}(\boldsymbol{k})) (\hat{g}(\boldsymbol{k}) - g(\boldsymbol{k})) + D_i\hat{g}(\boldsymbol{k}) \otimes D_i\hat{g}(\boldsymbol{k})) \delta(i, \boldsymbol{n}_i) \\
&+ \sum_{\boldsymbol{k} \in [K]^I} \sum_{\boldsymbol{n} \in \mathcal{D}^{\boldsymbol{k},n,i}} \sum_{j \neq i} (D_{i,j}L(\boldsymbol{k})) \delta(j, \boldsymbol{n}_j) + \sum_{\boldsymbol{n} \in \mathcal{D}^{n,i}} R(i, \boldsymbol{n})
\end{aligned} \tag{113}$$

We now rewrite or bound the four summands. First, we relate those expressions to the datapoint statistics matrices. We define

$$\boldsymbol{B}^{\boldsymbol{k},i}_n = |\mathcal{D}^{\boldsymbol{k},n,i}| = |\{\boldsymbol{n} \in \mathcal{D} : \Pi(\boldsymbol{n}) = \boldsymbol{k}, \, \boldsymbol{n}_i = n\}|. \tag{114}$$

and

$$\boldsymbol{A}^{\boldsymbol{k},i,j}_{n,n'} = |\{\boldsymbol{n} \in \mathcal{D}^{\boldsymbol{k},n,i} | \boldsymbol{n}_j = n'\}| = |\{\boldsymbol{n} \in \mathcal{D} : \Pi(\boldsymbol{n}) = \boldsymbol{k}, \, \boldsymbol{n}_i = n, \, \boldsymbol{n}_j = n'\}|. \tag{115}$$

Then the first term equals

$$\sum_{\boldsymbol{k}\in[K]^I}\sum_{\boldsymbol{n}\in\mathcal{D}^{\boldsymbol{k},n,i}}D_iL(\boldsymbol{k})=\sum_{\boldsymbol{k}\in[K]^I}\boldsymbol{B}_n^{\boldsymbol{k},i}D_iL(\boldsymbol{k}).\tag{116}$$

The second term is similarly given by

$$\sum_{\boldsymbol{k}\in[K]^I}\sum_{\boldsymbol{n}\in\mathcal{D}^{\boldsymbol{k},n,i}}\left((D_{i,i}\hat{g}(\boldsymbol{k}))\left(\hat{g}(\boldsymbol{k})-g(\boldsymbol{k})\right)+D_i\hat{g}(\boldsymbol{k})\otimes D_i\hat{g}(\boldsymbol{k})\right)\delta(i,\boldsymbol{n}_i)$$
$$=\sum_{\boldsymbol{k}\in[K]^I}\boldsymbol{B}_n^{\boldsymbol{k},i}\Big((D_{i,i}\hat{g}(\boldsymbol{k}))\left(\hat{g}(\boldsymbol{k})-g(\boldsymbol{k})\right)+D_i\hat{g}(\boldsymbol{k})\otimes D_i\hat{g}(\boldsymbol{k})\Big)\delta(i,n)\tag{117}$$

The third term is given by

$$\sum_{\boldsymbol{k}\in[K]^I}\sum_{\boldsymbol{n}\in\mathcal{D}^{\boldsymbol{k},n,i}}\sum_{j\neq i}(D_{i,j}L(\boldsymbol{k}))\delta(j,\boldsymbol{n}_j)=\sum_{\boldsymbol{k}\in[K]^I}\sum_{j\neq i}\sum_{n'\in[N]}\boldsymbol{A}_{n,n'}^{\boldsymbol{k},i,j}(D_{i,j}L(\boldsymbol{k}))\delta(j,n')$$
$$=\sum_{\boldsymbol{k}\in[K]^I}\sum_{j\neq i}\left(\left(\boldsymbol{A}^{\boldsymbol{k},i,j}\otimes D_{i,j}L(\boldsymbol{k})\right)\bar{\bar{\boldsymbol{\delta}}}(j)\right)_{nD:(n+1)D}\tag{118}$$

where $\bar{\bar{\boldsymbol{\delta}}}$ was introduced in (98). Let us summarize those findings as a proposition.

**Proposition 1.** *Assume that the bound* (15) *holds for some* $\Omega$ *and* $v(i,n)\in\Omega$ *for all* $i\in[I]$, $n\in[N]$. *Then the following expansion holds*

$$\frac{\mathrm{d}}{\mathrm{d}v(i,n)}\hat{L}(\bar{\boldsymbol{v}})=\sum_{\boldsymbol{k}\in[K]^I}\boldsymbol{B}_n^{\boldsymbol{k},i}D_iL(\boldsymbol{k})$$
$$+\sum_{\boldsymbol{k}\in[K]^I}\boldsymbol{B}_n^{\boldsymbol{k},i}\left((D_{i,i}\hat{g}(\boldsymbol{k}))\left(\hat{g}(\boldsymbol{k})-g(\boldsymbol{k})\right)+D_i\hat{g}(\boldsymbol{k})\otimes D_i\hat{g}(\boldsymbol{k})\right)\delta(i,n)$$
$$+\sum_{\boldsymbol{k}\in[K]^I}\sum_{j\neq i}\left(\left(\boldsymbol{A}^{\boldsymbol{k},i,j}\otimes D_{i,j}L(\boldsymbol{k})\right)\bar{\bar{\boldsymbol{\delta}}}(j)\right)_{nD:(n+1)D}\tag{119}$$
$$+\sum_{\boldsymbol{n}\in\mathcal{D}^{n,i}}R(i,\boldsymbol{n})$$

*Proof.* The result follows from the calculations above. □

## E   Bounds for the Loss Gradient

The goal of this section is to extract the asymptotically (with high probability) dominating terms of the expansion from the previous section as $S\to\infty$. The crucial observation is that the appearing averages can be essentially replaced by their expectation. This is a consequence of concentration properties of the datapoint statistics. These properties will be derived in Appendix G where they are collected in Theorem 9. Here, we just prove the result conditional on the bound in Theorem 9. Recall that we defined $\delta_{\max}=\max_{i\in[I],n\in[N]}|\delta(i,n)|$ in (11) as the maximal deviation norm and $r_{\max}=\max_{\boldsymbol{k}\in[K]^I}|\hat{g}(\boldsymbol{k})-g(\boldsymbol{k})|$ in (13) as the maximal residual. Finally, we introduce the notation

$$p(\boldsymbol{k},i)=\frac{|\Pi^{-1}(\boldsymbol{k})|}{N^I|\Pi_i^{-1}(\boldsymbol{k}_i)|}.\tag{120}$$

Note that $\boldsymbol{B}_n^{\boldsymbol{k},i}\sim\mathrm{Bin}(S,p(\boldsymbol{k},i))$ if $\Pi_i(n)=\boldsymbol{k}_i$ and $\boldsymbol{B}_n^{\boldsymbol{k},i}=0$ otherwise. In particular, $\mathbb{E}\boldsymbol{B}_n^{\boldsymbol{k},i}=S\cdot p(\boldsymbol{k},i)$. Note that if $\Pi$ is approximately balanced if the following bounds hold

$$Np(\boldsymbol{k},i)=N\frac{|\Pi^{-1}(\boldsymbol{k})|}{N^I|\Pi_i^{-1}(\boldsymbol{k}_i)|}=\prod_{j\neq i}\frac{|\Pi_j^{-1}(\boldsymbol{k}_j)|}{N}.\tag{121}$$

This implies

$$\left(\frac{1}{2K}\right)^I\leq Np(\boldsymbol{k},i)\leq\left(\frac{2}{K}\right)^I.\tag{122}$$

**Theorem 7.** *Assume that the bounds in Theorem 9 hold and assume $\hat{f}$ satisfies Assumption 2 for some set $\Omega$ and $v(i,n) \in \Omega$ for $i \in [I]$ and $[N]$. Then we obtain the following expansion for the loss gradient*

$$
\begin{aligned}
\frac{N}{S} \frac{\mathrm{d}}{\mathrm{d}v(i,n)} \hat{\mathcal{L}}(\bar{\boldsymbol{v}}) \\
= \sum_{\boldsymbol{k} \in [K]^I : \boldsymbol{k}_i = \Pi_i(n)} N p(\boldsymbol{k}, i) D_i L(\boldsymbol{k}) \\
+ \sum_{\boldsymbol{k} \in [K]^I : \boldsymbol{k}_i = \Pi_i(n)} N p(\boldsymbol{k}, i) \Big( (\hat{g}(\boldsymbol{k}) - g(\boldsymbol{k})) D_{i,i} \hat{g}(\boldsymbol{k}) + D_i \hat{g}(\boldsymbol{k}) \otimes D_i \hat{g}(\boldsymbol{k}) \Big) \delta(i,n) \\
+ E^F(i,n) + E^{S,1}(i,n) + E^{S,2}(i,n) + E^I(i,n) + E^T(i,n).
\end{aligned}
$$
(123)

*Here the error terms are bounded by*

$$
|E_\alpha^F(i,n)| \le 4M \min(1, r_{\max}) (2K)^{(I-1)/2} \sqrt{\ln(S)\frac{N}{S}},
$$
(124)

$$
|E_\alpha^{S,1}(i,n)| \le 4DM \min(1, r_{\max}) (2K)^{(I-1)/2} \sqrt{\ln(S)\frac{N}{S}} \delta_{\max},
$$
(125)

$$
|E_\alpha^{S,2}(i,n)| \le 4DM (2K)^{(I-1)/2} \sqrt{\ln(S)\frac{N}{S}} \delta_{\max},
$$
(126)

$$
|E^I(i,n)| \le 6DM (2K)^{(I-1)/2} \ln(S) \sqrt{\frac{N^2}{S}} \delta_{\max},
$$
(127)

$$
\left|E_\alpha^T(i,n)\right| \le MI^2 D \delta_{\max}^2 \left(1 + 4\sqrt{\frac{\ln(S)N}{S}}\right).
$$
(128)

*Proof.* The proof is a bit technical and essentially combines the assumptions and the concentration results in a straightforward fashion. First we remark that for $n \in \Pi_i^{-1}(k)$ we have as stated before that

$$
\mathbb{E}_{\mathcal{D}} \boldsymbol{B}_n^{\boldsymbol{k},i} = S \cdot p(\boldsymbol{k}, i)
$$
(129)

and $\boldsymbol{B}_n^{\boldsymbol{k},i} = 0$ otherwise. Then Proposition 1 implies that (123) holds with the definitions of the error terms given below. We now define and bound the error terms in the decomposition. Note that we have

$$
|(D_j L(\boldsymbol{k}))_\alpha| = \left| \frac{\mathrm{d}}{\mathrm{d}v_\alpha(j, \boldsymbol{k}_j)} L(\boldsymbol{k}) \right| = \left| \frac{\mathrm{d}}{\mathrm{d}v_\alpha(j, \boldsymbol{k}_j)} \hat{g}(\boldsymbol{k})(\hat{g}(\boldsymbol{k}) - g(\boldsymbol{k})) \right| \le M \min(1, r_{\max}). \quad (130)
$$

where we either bound both terms using Assumption 2 or the second term by $r_{\max}$ and the first one by Assumption 2 together with $M' \le M$. This then implies that the error term capturing the fluctuations of the occupation statistics

$$
E^F(i,n) = \frac{N}{S} \sum_{\boldsymbol{k} \in [K]^I} \left( \boldsymbol{B}_n^{\boldsymbol{k},i} - \mathbb{E}_{\mathcal{D}} \boldsymbol{B}_n^{\boldsymbol{k},i} \right) D_i L(\boldsymbol{k})
$$
(131)

can be bounded by

$$
|E_\alpha^F(i,n)| \le 4M \min(1, r_{\max}) (2K)^{(I-1)/2} \sqrt{\ln(S)\frac{N}{S}} \le C(I, K, M) \min(1, r_{\max}) \sqrt{\ln(S)\frac{N}{S}}.
$$
(132)

Here we used that there are $K^{I-1}$ non-vanishing terms in the sum (if $\Pi_i(n) \ne \boldsymbol{k}_i$ we have $\boldsymbol{B}_n^{\boldsymbol{k},i} = \mathbb{E} \boldsymbol{B}_n^{\boldsymbol{k},i} = 0$). Next, we consider the self interaction error terms

$$
E^{S,1}(i,n) = \frac{N}{S} \sum_{\boldsymbol{k} \in [K]^I} \left( \boldsymbol{B}_n^{\boldsymbol{k},i} - \mathbb{E} \boldsymbol{B}_n^{\boldsymbol{k},i} \right) (\hat{g}(\boldsymbol{k}) - g(\boldsymbol{k})) (D_{i,i} \hat{g}(\boldsymbol{k})) \delta(i,n),
$$
(133)

$$
E^{S,2}(i,n) = \frac{N}{S} \sum_{\boldsymbol{k} \in [K]^I} \left( \boldsymbol{B}_n^{\boldsymbol{k},i} - \mathbb{E} \boldsymbol{B}_n^{\boldsymbol{k},i} \right) (D_i \hat{g}(\boldsymbol{k}) \otimes D_i \hat{g}(\boldsymbol{k})) \delta(i,n).
$$
(134)

For $E^{S,1}$ we get (similar to above) and using that the operator norm of a $D \times D$ matrix $A$ is bounded by $D \max |a_{ij}|$) the bound

$$
\begin{aligned}
|E_\alpha^{S,1}(i,n)| &\leq 4DM \min(1, r_{\max}) (2K)^{(I-1)/2} \sqrt{\ln(S)\frac{N}{S}} \delta_{\max} \\
&\leq C(I,K,M,D) \min(1, r_{\max}) \delta_{\max} \sqrt{\ln(S)\frac{N}{S}}.
\end{aligned}
\tag{135}
$$

The error term $E^{S,2}$ could also be absorbed in other terms later on but since it is not dominant, we just bound it. We can obtain similarly to our treatment of $E^{S,1}$ the bound

$$
\begin{aligned}
|E_\alpha^{S,2}(i,n)| &\leq 4DM (2K)^{(I-1)/2} \sqrt{\ln(S)\frac{N}{S}} \delta_{\max} \\
&\leq C(I,K,M,D) \delta_{\max} \sqrt{\ln(S)\frac{N}{S}}.
\end{aligned}
\tag{136}
$$

Next, we consider the interaction error term

$$
E^I(i,n) = \frac{N}{S} \sum_{\boldsymbol{k} \in [K]^I} \sum_{j \neq i} \left( \left( \boldsymbol{A}^{\boldsymbol{k},i,j} \otimes D_{i,j} L(\boldsymbol{k}) \right) \bar{\boldsymbol{\delta}}(j) \right)_{nD:(n+1)D}
\tag{137}
$$

We note that $\mathbb{E}(\boldsymbol{A}_{n_1,n_2}^{\boldsymbol{k},i,j}) = c$ for some constant $c$ if $\Pi_i(n_1) = \boldsymbol{k}_i$ and $\Pi_j(n_2) = \boldsymbol{k}_j$ and $0$ otherwise. This implies that if $\Pi_i(n) = \boldsymbol{k}_i$

$$
\left( \left( \mathbb{E}\boldsymbol{A}^{\boldsymbol{k},i,j} \otimes D_{i,j} L(\boldsymbol{k}) \right) \bar{\boldsymbol{\delta}}(j) \right)_{nD:(n+1)D} = \sum_{n' \in \Pi_j^{-1}(\boldsymbol{k}_j)} L(\boldsymbol{k}) \delta(j, n') = 0
\tag{138}
$$

here we used

$$
\sum_{n \in |\Pi_i^{-1}(k)|} \delta(i,n) = 0
\tag{139}
$$

which follows from the definition (10) in the last step. If $\Pi_i(n) \neq \boldsymbol{k}_i$ then this expression is clearly zero as the corresponding row of $\boldsymbol{A}^{\boldsymbol{k},i,j}$ is zero. Thus, we find that

$$
\begin{aligned}
|E^I(i,n)| &= \left| \frac{N}{S} \sum_{\boldsymbol{k} \in [K]^I} \sum_{j \neq i} \left( \left( (\boldsymbol{A}^{\boldsymbol{k},i,j} - \mathbb{E}\boldsymbol{A}^{\boldsymbol{k},i,j}) \otimes D_{i,j} L(\boldsymbol{k}) \right) \bar{\boldsymbol{\delta}}(j) \right)_{nD:(n+1)D} \right| \\
&\leq K^{I-1} \frac{N}{S} \max_{\boldsymbol{k},i,j} \| \boldsymbol{A}^{\boldsymbol{k},i,j} - \mathbb{E}\boldsymbol{A}^{\boldsymbol{k},i,j} \| \| D_{i,j} L(\boldsymbol{k}) \| \cdot \max_j |\bar{\boldsymbol{\delta}}(j)|.
\end{aligned}
\tag{140}
$$

Here we used the submultiplicativity of the operator norm of Kronecker-products stated in (281). Now we use $|\bar{\boldsymbol{\delta}}(j)| \leq \sqrt{N}\delta_{\max}$, the concentration bound (252), and bound $\| D_{i,j} L(\boldsymbol{k}) \| \leq DM$ similar as before to find

$$
|E^I(i,n)| = 6DM (2K)^{(I-1)/2} \ln(S) \sqrt{\frac{N^2}{S}} \delta_{\max}.
\tag{141}
$$

Finally, we bound the Taylor expansion remainder term

$$
E^T(i,n) = \frac{N}{S} \sum_{\boldsymbol{n} \in \mathcal{D}^{n,i}} R(i, \boldsymbol{n}).
\tag{142}
$$

Recall that

$$
R_\alpha(i, \boldsymbol{n}) = \sum_{j_1, j_2 \in [I]} \sum_{\beta_1 \beta_2 \in [D]} R_{(j_1,\beta_1),(j_2,\beta_2)}^{n,i,\alpha}(\boldsymbol{w}(\boldsymbol{k})) \delta_{\beta_1}(j_1, \boldsymbol{n}_{j_1}) \delta_{\beta_2}(j_2, \boldsymbol{n}_{j_2})
\tag{143}
$$

and

$$
|R_{(j_1,\beta_1),(j_2,\beta_2)}^{n,i,\alpha}| \leq \max_{\boldsymbol{v} \in \Omega} \frac{1}{2} \left| \frac{\mathrm{d}}{\mathrm{d}v_{\beta_1}(j_1, \boldsymbol{n}_{j_1})} \frac{\mathrm{d}}{\mathrm{d}v_{\beta_2}(j_2, \boldsymbol{n}_{j_2})} h(\boldsymbol{v}) \right|
\tag{144}
$$

where $h = h^{n,i,\alpha}$ was introduced in (100).

Assuming the bounds (15) and (16) we find

$$|R^{n,i,\alpha}_{(j_1,\beta_1),(j_2,\beta_2)}| \le \max_{\boldsymbol{v} \in \Omega} \frac{1}{2} \left| \frac{\mathrm{d}}{\mathrm{d}v_{\beta_1}(j_1, \boldsymbol{n}_{j_1})} \frac{\mathrm{d}}{\mathrm{d}v_{\beta_2}(j_2, \boldsymbol{n}_{j_2})} h(\boldsymbol{v}) \right| \le 4 \frac{\sqrt{M}}{4} \cdot \left( \frac{\sqrt{M}}{4} + \frac{\sqrt{M}}{4} \right) \le M. \tag{145}$$

Indeed, the derivatives of $h$ can be decomposed by the product rule in 4 terms which each can bounded using (15) and (16). Then we can bound the remainder term as follows

$$|R_\alpha(i, \boldsymbol{n})| \le M \sum_{j_1, j_2 \in [I]} \sum_{\beta_1 \beta_2 \in [D]} |\delta_{\beta_1}(j_1, \boldsymbol{n}_{j_1}) \delta_{\beta_2}(j_2, \boldsymbol{n}_{j_2})| \le MID \sum_{j \in [I]} |\delta(j, \boldsymbol{n}_j)|^2. \tag{146}$$

Recalling that $\mathcal{D}^{n,i}$ defined (112) satisfies $|\mathcal{D}^{n,i}| = \mathbf{B}^i_n$ we obtain using the concentration bound (253) from Lemma 9

$$\left|E^T_\alpha(i,n)\right| = \frac{N}{S} \left| \sum_{\boldsymbol{n} \in \mathcal{D}^{n,i}} R_\alpha(i, \boldsymbol{n}) \right| \le \frac{N}{S} \mathbf{B}^i_n M I^2 D \delta^2_{\max} \le M I^2 D \delta^2_{\max} \left( 1 + 4\sqrt{\frac{\ln(S)N}{S}} \right). \tag{147}$$

$\square$

If we consider the gradient descent dynamics we obtain the following expansion for the time derivative $\dot{w}(i, n)$. Let us from now on absorb the dependence on $I$ and $D$ in generic constants $C$.

**Corollary 1.** *Under the same assumptions as in Theorem 7 we get*

$$\dot{w}(i, k) = - \sum_{\boldsymbol{k} \in [K]^I, \boldsymbol{k}_i = k} Np(\boldsymbol{k}, i) D_i L(\boldsymbol{k}) - \lambda w(i, k) + E^w(i, k) \tag{148}$$

*where $E^w$ can be bounded by*

$$|E^w_\alpha(i, k)| \le C \min(1, r_{\max}) K^{(I-1)/2} \sqrt{\ln(S)\frac{N}{S}} + C K^{(I-1)/2} \ln(S) \sqrt{\frac{N^2}{S}} \delta_{\max} + C \delta^2_{\max} \tag{149}$$

*for $S \ge N \ln(N)$.*

*Proof.* The result follows from Theorem 7, the gradient dynamics (see (8)), and the definition of $w(i, n)$ (see (9)). We also use the relation $\sum_{n \in |\Pi^{-1}_i(k)|} \delta(i, n) = 0$ (already stated in (139)) to conclude that terms involving $\delta(i, n)$ cancel. The condition $S \ge N \ln(N)$ allows us to bound $\ln(S)N/S \le 2$ to control $E^T$. Note that here we use that $E^{S,1}$ and $E^{S,2}$ can be bounded by $E^I$. Note that we absorbed all terms involving $\delta_{\max}$ into the dominating term. $\square$

We therefore get the following bound for $\dot{\delta}(i, n)$.

**Corollary 2.** *Under the same assumptions as in Theorem 7 we get*

$$\dot{\delta}(i, n) = - \sum_{\boldsymbol{k} \in [K]^I, \boldsymbol{k}_i = n} Np(\boldsymbol{k}, i) \Big( (\hat{g}(\boldsymbol{k}) - g(\boldsymbol{k})) D_{i,i} \hat{g}(\boldsymbol{k}) + D_i \hat{g}(\boldsymbol{k}) \otimes D_i \hat{g}(\boldsymbol{g}) \Big) \delta(i, n) \tag{150}$$
$$- \lambda \dot{\delta}(i, n) + E^\delta(i, n).$$

*where $E^\delta$ can be bounded by*

$$|E^\delta(i, n)| \le C \min(1, r_{\max}) K^{(I-1)/2} \sqrt{\ln(S)\frac{N}{S}} + C K^{(I-1)/2} \ln(S) \sqrt{\frac{N^2}{S}} \delta_{\max} + C \delta^2_{\max} \tag{151}$$

*for $S \ge N \ln(N)$.*

*Proof.* This follows from Theorem 7, Corollary 2, and the relation

$$\dot{\delta}(i,n) = \dot{v}(i,n) - \dot{w}(i,\Pi_i(n)). \tag{152}$$

$\square$

Now we are in the position to control the time evolution of $|\delta(i,n)|^2$. We obtain the following relation

**Lemma 1.** *Under the same assumptions as in Theorem 7 the following bound holds*

$$\frac{\mathrm{d}}{\mathrm{d}t}\frac{1}{2}|\delta(i,n)|^2 \leq \left(-\lambda - \frac{\omega}{(2K)^I} + 2^I \sup_{v_1,\ldots,v_I \in \Omega} \max_i \|D_{i,i}^2 \hat{f}(v_1,\ldots,v_n)\| r_{\max}\right) |\delta(i,n)|^2$$

$$+ C_1 \min(1, r_{\max}) K^{(I-1)/2} \sqrt{\ln(S)\frac{N}{S}} \delta_{\max} \tag{153}$$

$$+ C_1 K^{(I-1)/2} \ln(S) \sqrt{\frac{N^2}{S}} \delta_{\max}^2 + C_1 \delta_{\max}^3$$

*where $C_1$ is a constant depending on $D$, $I$, and $M$, and $\omega$ is defined by*

$$\omega = \min_{k,i} \lambda_{\min}\left(\sum_{\boldsymbol{k}\in[K]^I, \boldsymbol{k}_i=k} D_i\hat{g}(\boldsymbol{k}) \otimes D_i\hat{g}(\boldsymbol{k})\right) \geq 0. \tag{154}$$

*Proof.* We have

$$\frac{\mathrm{d}}{\mathrm{d}t}\frac{1}{2}|\delta(i,n)|^2 = \delta(i,n)\dot{\delta}(i,n)$$

$$= -\sum_{\boldsymbol{k}\in[K]^I, \boldsymbol{k}_i=n} Np(\boldsymbol{k},i)(\hat{g}(\boldsymbol{k}) - g(\boldsymbol{k}))\delta(i,n) \cdot (D_{i,i}\hat{g}(\boldsymbol{k}))\,\delta(i,n)$$

$$- \sum_{\boldsymbol{k}\in[K]^I, \boldsymbol{k}_i=n} Np(\boldsymbol{k},i)\delta(i,n) \cdot D_i\hat{g}(\boldsymbol{k}) \otimes D_i\hat{g}(\boldsymbol{k})\delta(i,n) \tag{155}$$

$$- \delta(i,n)E^\delta(i,n) - \lambda|\delta(i,n)|^2.$$

Recall that by (122) we have

$$\left(\frac{1}{2K}\right)^I \leq Np(\boldsymbol{k},i) \leq \left(\frac{2}{K}\right)^I. \tag{156}$$

We consider $\omega$ as defined in the statement of the lemma where $\omega \geq 0$ follows because we sum positive semi-definite rank one matrices. Then we find using the lower bound on the spectrum

$$\delta(i,n)\cdot \left(\sum_{\boldsymbol{k}\in[K]^I, \boldsymbol{k}_i=n} D_i\hat{g}(\boldsymbol{k}) \otimes D_i\hat{g}(\boldsymbol{k})\right)\delta(i,n)$$

$$\geq |\delta(i,n)|^2 \lambda_{\min}\left(\sum_{\boldsymbol{k}\in[K]^I, \boldsymbol{k}_i=n} D_i\hat{g}(\boldsymbol{k}) \otimes D_i\hat{g}(\boldsymbol{k})\right) \geq \omega|\delta(i,n)|^2. \tag{157}$$

Thus we find

$$\frac{\mathrm{d}}{\mathrm{d}t}\frac{1}{2}|\delta(i,n)|^2 \leq -\lambda|\delta(i,n)|^2 - \frac{\omega}{(2K)^I}|\delta(i,n)|^2$$

$$+ 2^I \sup_{v_1,\ldots,v_I \in \Omega} \max_i \|D_{i,i}^2 \hat{f}(v_1,\ldots,v_n)\| r_{\max}|\delta(i,n)|^2 - \delta(i,n)E^\delta(i,n) \tag{158}$$

where we used for the second to last contribution that we sum over $K^{I-1}$ terms which is cancelled by the $(2/K)^I$ factor. We finally bound the error term by

$$|\delta(i,n) \cdot E^\delta(i,n)|$$

$$\leq C\sqrt{D}\min(1, r_{\max})K^{(I-1)/2}\sqrt{\ln(S)\frac{N}{S}}\delta_{\max} + CK^{(I-1)/2}\sqrt{\ln(S)\frac{N^2}{S}}\delta_{\max}^2 + C\delta_{\max}^3 \tag{159}$$

$$\leq C\min(1, r_{\max})K^{(I-1)/2}\sqrt{\ln(S)\frac{N}{S}}\delta_{\max} + CK^{(I-1)/2}\sqrt{\ln(S)\frac{N^2}{S}}\delta_{\max}^2 + C\delta_{\max}^3.$$

This ends the proof. ☐

# F  Proofs of the Main Results

In this section, we prove our main results, Theorem 4 and Theorem 5. In addition, we state and prove Theorem 8 which is a variant of Theorem 5 without a-priori bound on the embeddings. Essentially, the proof strategy for both results is to rely on the groundwork from the previous two sections, in particular on Lemma 1. Indeed, we first verify that the conditions of Lemma 1 hold for all times and then application of this Lemma allows us to control the evolution of $\delta_{\max}$ for all times.

*Proof of Theorem 4.* The proof proceeds in three steps. First we use the monotonicity of the loss to deduce that $r_{\max}(t)$ can be bounded in terms of $r_{\max}(0)$ and $\delta_{\max}(t)$ and $\delta_{\max}(0)$. Then we show that choosing the variables as in the statement of the theorem allows us to bound all the terms in Lemma 1 in Appendix E. Then we apply Lemma 1 and deduce the decay of the maximum of the displacements $|\delta(i, n, t)|$. Here we need to check carefully that the assumptions of the lemma are satisfied for all $t$.

First we assume the conclusions on the concentration properties from Theorem 9 in Appendix G hold which occurs with probability at least $1 - S^{-1}$ over the randomness of the training data.

Let us consider any time $T > 0$ and we assume that $v(i, n, t) \in \Omega$ for all $i \in [I]$, $n \in [N]$ and $0 \leq \tau \leq T$. This allows us in particular to apply the Assumption 2 up to time $T$ for embeddings $v(i, n)$.

We now implement the first step where we bound $r_{\max}(t)$ for all $0 \leq t \leq T$ in terms of $r_{\max}(0)$, $\delta_{\max}(t)$, and $\delta_{\max}(t)$. The idea is to upper bound the initial sample loss $\hat{\mathcal{L}}(t = 0)$ and lower bound the loss $\hat{\mathcal{L}}(t > 0)$. First we note that by a first order Taylor expansion and using Assumption 2

$$|\hat{f}(\boldsymbol{v}(\boldsymbol{n})) - \hat{f}(\boldsymbol{w}(\Pi(\boldsymbol{n})))| \leq \sum_{i \in I} M |\delta(i, \boldsymbol{n}_i)| \leq M I \delta_{\max}. \tag{160}$$

This implies that

$$\hat{\mathcal{L}}(\bar{\boldsymbol{v}}) = \frac{1}{2} \sum_{\boldsymbol{n} \in \mathcal{D}} (\hat{f}(\boldsymbol{v}(\boldsymbol{n})) - g(\Pi(\boldsymbol{n})))^2$$

$$\leq \frac{1}{2} \sum_{\boldsymbol{n} \in \mathcal{D}} \left( |\hat{f}(\boldsymbol{w}(\Pi(\boldsymbol{n}))) - g(\Pi(\boldsymbol{n}))| + \sum_{i \in I} M |\delta(i, \boldsymbol{n}_i)| \right)^2 \leq \frac{1}{2} S (r_{\max} + I M \delta_{\max})^2. \tag{161}$$

Next, we derive a lower bound on the loss. Let $\boldsymbol{k}$ be such that $|\hat{f}(\boldsymbol{k}) - f(\boldsymbol{k})| = r_{\max}$. By assumption

$$|\Pi^{-1}(\boldsymbol{k})| \geq \left( \frac{N}{2K} \right)^I \tag{162}$$

and using concentration bounds (e.g., by summing (254) over $n$ and using the lower bound for $S$) we find that

$$|\mathcal{D}^{\boldsymbol{k}}| = |\{\boldsymbol{n} \in \mathcal{D} : \Pi(\boldsymbol{n}) = \boldsymbol{k}\}| \geq \frac{1}{2} \frac{S}{(2K)^I}. \tag{163}$$

Thus we can lower bound the loss using the same Taylor expansion as before by

$$\hat{\mathcal{L}}(\bar{\boldsymbol{v}}(t)) \geq \frac{1}{4} \frac{S}{(2K)^I} \left( \max(0, r_{\max}(t) - M I \delta_{\max}(t)) \right)^2. \tag{164}$$

Since gradient descent does not increase the loss the bounds (161) and (164) together imply for $t \geq 0$

$$\frac{1}{2} S (r_{\max}(0) + I M \delta_{\max}(0))^2 \geq \hat{\mathcal{L}}(\bar{\boldsymbol{v}}(0))) \geq \hat{\mathcal{L}}(\bar{\boldsymbol{v}}(t))$$

$$\geq \frac{1}{4} \frac{S}{(2K)^I} \left( \max(0, r_{\max}(t) - M I \delta_{\max}(t)) \right)^2. \tag{165}$$

This implies

$$r_{\max}(t) \leq 2(2K)^{I/2}(r_{\max}(0) + IM\delta_{\max}(0)) + IM\delta_{\max}(t). \tag{166}$$

Then we get the bound

$$r_{\max}(t) \leq 4(2K)^{I/2}(r_{\max}(0) + IM\max(\delta_{\max}(0), \delta_{\max}(t))). \tag{167}$$

Now we choose $C_2$ and $C_3$ such that the initialization condition (20) becomes

$$\delta_{\max}(0) \leq \min\left(\frac{\omega_0}{8C_1(2K)^I}, \frac{\omega_0}{64IM^2D(2K)^{3I/2}2^I}, \frac{1}{8(2K)^{I/2}IM}, 1\right) =: \bar{\delta} \tag{168}$$

and

$$r_{\max}(0) \leq \min\left(\frac{\omega_0}{64MD(2K)^{3I/2}2^I}, \frac{1}{8(2K)^{I/2}}\right). \tag{169}$$

Assume in addition that for all times $0 \leq t \leq T$ the bound $\delta_{\max}(t) \leq \bar{\delta}$ holds. Using (167) we then find that for $0 \leq t \leq T$

$$r_{\max}(t) \leq 4(2K)^{I/2}\left(\frac{1}{64MD(2K)^{3I/2}2^I} + IM\frac{\omega_0}{64IM^2D(2K)^{3I/2}2^I}\right) \leq \frac{\omega_0}{8MD(2K)^I2^I}. \tag{170}$$

and similarly

$$r_{\max}(t) \leq 4(2K)^{I/2}\left(\frac{1}{8(2K)^{I/2}} + IM\frac{1}{8(2K)^{I/2}IM}\right) \leq 1 \tag{171}$$

and therefore

$$r_{\max}(t) \leq \min\left(\frac{\omega_0}{8MD(2K)^I2^I}, 1\right). \tag{172}$$

The next step is to bound the various error terms appearing in Lemma 1. Using the last display we can bound for all $i \in [I]$ and $n \in [N]$ using Assumption 2 for $t \in [0,T]$

$$2^I \sup_{v_1,\ldots,v_I \in \Omega} \max_i \|D_{i,i}^2 \hat{f}(v_1,\ldots,v_n)\| r_{\max}(t)|\delta(i,n,t)|^2 \leq 2^I MD\frac{\omega_0}{8MD(2K)^I2^I}|\delta(i,n,t)|^2$$

$$\leq \frac{\omega_0}{8(2K)^I}|\delta(i,n,t)|^2. \tag{173}$$

Moreover, we can estimate for $0 \leq t \leq T$

$$C_1\delta_{\max}^3(t) \leq C_1\delta_{\max}^2(t)\frac{\omega_0}{8C_1(2K)^I} \leq \frac{\omega_0}{8(2K)^I}\delta_{\max}^2(t). \tag{174}$$

Now we observe that for $x = a\ln^2(a)$ and $a \geq 1$ we find (using $\ln(a) \leq a$) the bound

$$\frac{\ln(x)}{\sqrt{x}} \leq \frac{\ln(a^3)}{\sqrt{a}\ln(a)} = \frac{3}{\sqrt{a}}. \tag{175}$$

Hence we consider

$$S \geq S_0 = \frac{24^2C_1^2(2K)^{3I}N^2}{\omega_0^2}\ln^2\left(\frac{24^2C_1^2(2K)^{3I}N^2}{\omega_0^2}\right). \tag{176}$$

Note that tracking only the dependence on $K$ and $N$ (and using $K \leq N$) this condition reads

$$S \geq C(2K)^{3I}N^2\ln^2(N). \tag{177}$$

Then we find using (175) combined with the observation that $\ln(S)/\sqrt{S}$ is decreasing for $S \geq e^2$ for $S \geq S_0$ the bound

$$
\begin{aligned}
C_1 K^{(I-1)/2} \ln(S)\sqrt{\frac{N^2}{S}} &\leq C_1 K^{(I-1)/2} \ln(S_0)\sqrt{\frac{N^2}{S_0}} \\
&\leq \frac{3C_1 K^{I/2} N}{24 C_1 (2K)^{3I/2} N \omega_0^{-1}} \\
&\leq \frac{1}{8}\frac{\omega_0}{(2K)^I}.
\end{aligned}
\tag{178}
$$

We note that similarly for $x = a\ln(a)$ we get

$$
\frac{\ln(x)}{x} \leq \frac{2\ln(a)}{a\ln(a)} = \frac{2}{a}.
\tag{179}
$$

Thus we consider similarly

$$
S \geq S_1 = \frac{2 \cdot 16^2 N (2K)^{3I} C_1^2}{\omega_0^2 \bar{\delta}} \ln\left(\frac{2\cdot 16^2 N(2K)^{3I}C_1^2}{\omega_0^2\bar{\delta}}\right)
\tag{180}
$$

where $\bar{\delta}$ was introduced in (168). Note that tracking only $N$ and $K$ this bound becomes

$$
S \geq CN\ln(N)K^{9I/2}.
\tag{181}
$$

Then, similar to before, (using monotonicity of $\ln(S)/S$ for $S \geq e$) we find for $S \geq S_1$

$$
\begin{aligned}
C_1 K^{(I-1)/2}\max(1, r_{\max})\sqrt{\ln(S)\frac{N}{S}} &\leq C_1 K^{I/2}\max(1, r_{\max})\sqrt{\ln(S_1)\frac{N}{S_1}} \\
&\leq \frac{1}{16}\frac{\omega_0}{(2K)^I}\bar{\delta}\max(1, r_{\max}(t)).
\end{aligned}
\tag{182}
$$

After all these preliminary estimates we can now finish the proof.

Let us define $T \in \mathbb{R}_0^+ \cup \{\infty\}$ to be the maximal time such that $v(i, n, t) \in \Omega$ for all $i \in [I]$, $n \in [N]$ and $0 \leq t \leq T$ and the bound (168) holds for $\delta_{\max}(t)$ where we for convenience recall

$$
\delta_{\max}(t) \leq \bar{\delta} = \min\left(\frac{\omega_0}{8C_1(2K)^I}, \frac{\omega_0}{64IM^2D(2K)^{3I/2}2^I}, \frac{1}{8(2K)^{I/2}IM}, 1\right).
\tag{183}
$$

We want to show that $T = \infty$. We assume that $T < \infty$ and prove by contradiction that this does not hold. By (172) we know that for $t \leq T$ the bound $r_{\max}(t) \leq 1$ holds and therefore Assumption 4 implies that $w(i, k, t = 0) \in \Omega_0$ and since moreover $\delta_{\max}(t) \leq 1$ for $t \leq T$ we find that $v(i, n, t = 0) \in \Omega_0 + B_1(0) \subset\subset \Omega$. By continuity of $v(i, n, t)$ in $t$ we conclude that there is $\varepsilon > 0$ such that $v(i, n, t) \in \Omega$ for all $0 \leq t \leq T + \varepsilon$, $i \in [I]$, and $n \in [N]$. Thus, we can apply Lemma 1 for $t = T$ and then find using Assumption 3 and the bounds (173), (174), (178), and (182)

$$
\begin{aligned}
\frac{d}{dt}\frac{1}{2}|\delta(i,n,t)|^2 &\leq \left(-\lambda - \frac{\omega_0}{(2K)^I} + 2^I \sup_{v_1,\ldots,v_I\in\Omega}\max_i\|D_{i,i}^2\hat{f}(v_1,\ldots,v_n)\|r_{\max}\right)|\delta(i,n,t)|^2 \\
&\quad + C_1\max(1,r_{\max})K^{(I-1)/2}\sqrt{\ln(S)\frac{N}{S}}\delta_{\max}(t) + C_1 K^{(I-1)/2}\ln(S)\sqrt{\frac{N^2}{S}}\delta_{\max}^2 + C_1\delta_{\max}(t)^3 \\
&\leq \frac{\omega_0}{(2K)^I}\left(-|\delta(i,n,t)|^2 + \tfrac{1}{8}|\delta(i,n,t)|^2 + \frac{1}{16}\max(1,r_{\max}(t))\bar{\delta} + \tfrac{1}{8}\delta_{\max}(t)^2 + \tfrac{1}{8}\delta_{\max}(t)^2\right) \\
&\leq \frac{\omega_0}{(2K)^I}\left(-|\delta(i,n,t)|^2 + \tfrac{3}{8}\delta_{\max}(t)^2 + \frac{1}{16}\max(1,r_{\max}(t))\bar{\delta}.\right)
\end{aligned}
\tag{184}
$$

If $|\delta(i,n,t)| \geq 3/4\delta_{\max}$ we find $|\delta(i,n,t)|^2 \geq \delta_{\max}(t)^2/2$ and we conclude that

$$
\begin{aligned}
\frac{d}{dt}\frac{1}{2}|\delta(i,n,t)|^2 &\leq \frac{\omega_0}{(2K)^I}\delta_{\max}(t)(-\tfrac{1}{2}\delta_{\max}(t) + \tfrac{3}{8}\delta_{\max}(t) + \frac{1}{16}\max(1,r_{\max}(t))\bar{\delta} \\
&\leq \frac{1}{8}\frac{\omega_0}{(2K)^I}\delta_{\max}(t)(-\delta_{\max}(t) + \bar{\delta}/2).
\end{aligned}
\tag{185}
$$

Now, if $\delta_{\max}(T) \leq \bar{\delta}/2$ we conclude by continuity that there is $\varepsilon > 0$ such that $\delta_{\max}(t) < \bar{\delta}$ for $t \in [T, T + \varepsilon]$. On the other hand, if $\delta_{\max}(t) > \bar{\delta}$ we conclude that $\frac{d}{dt}\frac{1}{2}|\delta(i, n, T)|^2 < 0$ for all $i, n$ such that $|\delta(i, n, t)| \geq 3/4\delta_{\max}$. This in particular implies that $\delta_{\max}(t) < \delta_{\max}(T)$ for $t \in [T, T+\varepsilon]$ and some $\varepsilon > 0$ ($\delta_{\max}$ is non-increasing at $T$). This is a contradiction, and we conclude that $T = \infty$.

Finally, we prove the decay of $\delta_{\max}$. Assume that for $t \geq T$ the bound $\min(1, r_{\max}(t)) \leq R$ holds. Note that the function $\delta_{\max}(t)$ is not necessarily differentiable but its left and right derivative exist (as the maximum of finitely many differentiable functions). Then we obtain, similar to (184), for the right derivative the bound

$$\frac{d}{dt^+}\frac{1}{2}\delta_{\max}(t)^2 \leq -\frac{1}{8}\frac{\omega_0}{(2K)^I}\delta_{\max}(t)^2 + C_1 R K^{(I-1)/2}\sqrt{\ln(S)\frac{N}{S}}\delta_{\max}(t). \tag{186}$$

For

$$\delta_{\max}(t) > \frac{C_1 16(2K)^I K^{(I-1)/2} R}{\omega_0}\sqrt{\ln(S)\frac{N}{S}} \tag{187}$$

the previous bound simplifies to

$$\frac{d}{dt^+}\frac{1}{2}\delta_{\max}(t)^2 \leq -\frac{1}{16}\frac{\omega_0}{(2K)^I}\delta_{\max}(t)^2 \tag{188}$$

which implies that for $t \geq T$ by Gronwall's Lemma

$$\delta_{\max}(t)^2 \leq \exp\left(-\frac{1}{16}\frac{\omega_0}{(2K)^I}(t - T)\right)\delta_{\max}(T)^2. \tag{189}$$

Thus in finite time we achieve

$$\delta_{\max}(t) \leq 2\frac{C_1 16(2K)^I K^{(I-1)/2} R}{\omega_0}\sqrt{\ln(S)\frac{N}{S}} \leq CK^{3I/2}\sqrt{\ln(S)\frac{N}{S}}R. \tag{190}$$

This ends the proof. Note that we also get an exponential rate of convergence. $\qquad\square$

*Remark* 3. The exponent $9/2$ for the lower bound of $S$ in (181) is not tight because we could use that $r_{\max}$ is small, but this provides only a small improvement that does not justify the additional technicalities.

Before proving and stating Theorem 8 let us first prove that the weight decay $\lambda > 0$ allows us to derive a-priori bounds on the time evolution as stated in the following lemma.

**Lemma 2.** *Let $\Pi$ be approximately balanced, i.e., assume that Assumption 1 holds. Let $M_1 > 0$ be such that*

$$\sup_{v_1,\ldots,v_I \in B_1(0) \subset \mathbb{R}^D} |\hat{f}(v_1, \ldots, v_i)| \leq \frac{\sqrt{M_1}}{2}, \quad \max_{\boldsymbol{k}}|g(\boldsymbol{k})| \leq \frac{\sqrt{M_1}}{2}. \tag{191}$$

*Let $0 < \lambda \leq 1$ be a fixed number and assume that $v(i, n, t)$ follow the gradient dynamics (8) and are initialized such that*

$$|v(i, n, t = 0)| \leq 1 \tag{192}$$

*and assume that the dynamics exists for all times. Then for all $i \in [I]$, $k \in [K]$, and all times*

$$|w(i, k, t)| \leq R' = \sqrt{\frac{4KM_1}{\lambda} + 2IK}. \tag{193}$$

*Proof.* Note that by assumption

$$\frac{N}{S}\hat{\mathcal{L}}(\bar{\boldsymbol{v}}(t = 0)) = \frac{N}{S}\sum_{\boldsymbol{n} \in \mathcal{D}}\frac{1}{2}(\hat{f}(\boldsymbol{n}) - f(\boldsymbol{n}))^2 \leq \frac{N}{S}S\left(\frac{2\sqrt{M_1}}{2}\right)^2 = NM_1. \tag{194}$$

Then we conclude that

$$N\left(M_1 + I\frac{\lambda}{2}\right) \geq \hat{\mathcal{R}}^\lambda(\bar{\boldsymbol{v}}(0)) \geq \hat{\mathcal{R}}^\lambda(\bar{\boldsymbol{v}}(t)) \geq \frac{\lambda}{2}\max_{i\in[I]}\max_{k\in[K]}\sum_{n\in\Pi_i^{-1}(k)}|v(i,n,t)|^2$$

$$\geq \frac{\lambda}{2}\min_{i\in[I],k\in[K]}|\Pi_i^{-1}(k)|\max_{i\in[I],k\in[K]}|w(i,k,t)|^2 \geq \frac{N\lambda}{4K}\max_{i\in[I],k\in[K]}|w(i,k,t)|^2. \tag{195}$$

Thus we conclude that

$$\max_{i\in[I],k\in[K]}|w(i,k,t)|^2 \leq \frac{4KM_1}{\lambda} + 2IK. \tag{196}$$

This ends the proof. $\square$

Now we state a refined version of Theorem 5 which does not require an a-priori bound but instead proves boundedness of the embeddings $v(i,n,t)$ for all times using Lemma 2.

**Theorem 8.** *Let $\Pi$ be approximately balanced, i.e., assume that Assumption 1 holds. Assume that the function $\hat{f} : \mathbb{R}^{ID} \to \mathbb{R}$ is slot-wise linear. Assume that there is $M_1 > 0$ such that*

$$\sup_{v_1,\ldots,v_I\in B_1(0)\subset\mathbb{R}^D}|\hat{f}(v_1,\ldots,v_I)| \leq \frac{\sqrt{M_1}}{2}, \quad \max_{\boldsymbol{k}}|g(\boldsymbol{k})| \leq \frac{\sqrt{M_1}}{2}. \tag{197}$$

*Let $0 < \lambda \leq 1$ be a fixed number and assume that for*

$$R = \sqrt{\frac{4KM_1}{\lambda} + 2IK} + 3 \tag{198}$$

*Assumption 2 holds with $\Omega = B_R(0)$ and some $M > 0$. Note that $M$ depends on $R$ and thus on $K$ and $\lambda$. Let $C_1$ be the constant from Lemma 1. Assume that $v(i,n,t)$ follow the gradient dynamics (8) and are initialized such that*

$$|v(i,n,t=0)| \leq \frac{\lambda}{8C_1}. \tag{199}$$

*Then there are constants $C_2, C_3 \geq 0$ depending on $M, I, D, \lambda$, and $C_1$ such that for*

$$S \geq \max\left(C_2\frac{N^2K^{I-1}}{\lambda^2}\ln^2(N/\lambda), C_3\frac{NK^{I-1}}{\lambda^4}\ln(N/\lambda)\right) \tag{200}$$

*the bound*

$$\delta_{\max}(t) \leq \max\left(\delta_{\max}(0)e^{-\lambda t/8}, \frac{4C_1K^{(I-1)/2}}{\lambda}\sqrt{\ln(S)\frac{N}{S}}\right)$$

$$\leq \max\left(\delta_{\max}(0)e^{-\lambda t/8}, C_{I,D,\lambda,K}\sqrt{\ln(S)\frac{N}{S}}\right) \tag{201}$$

*holds for all $t \geq 0$.*

*Remark* 4. We emphasize again that since $M$ might depend on $\lambda$ we obtain no explicit rate in $\lambda$. However, the $S$ and $N$ dependence is the same as in Theorem 5.

*Proof.* The general strategy of the proof is similar to the proof of Theorem 4, but the proof is slightly simpler, as it is easier to obtain a-priori estimates on the evolution of $v(i,n,t)$. We assume that the conclusion of Theorem 9 holds, which occurs with probability at least $1 - S^{-1}$ over the randomness of $\mathcal{D}$.

Now let $T \geq 0$ be the largest time such that for all $0 \leq t \leq T$ and all $i \in [I]$ and $n \in [N]$ we have $v(i,n,t) \in \Omega$ and the bound

$$\delta_{\max}(t) \leq \frac{\lambda}{4C_1} \leq 1 \tag{202}$$

(we assume w.l.o.g. that $C_1 \geq 1$) holds for $0 \leq t \leq T$. By assumption, those relations hold at $t = 0$ (note that $\delta_{\max}(t) \leq 2\max_{i,n}|v(i,n)|$). We can bound for $t = T$ using the a-priori estimate (193) from Lemma 2 and find

$$|v(i,n,T)| \leq |w(i,\Pi_i(k),T)| + \delta_{\max}(T) \leq \sqrt{\frac{4KM_1}{\lambda} + 2IK} + 2 \leq R - 1. \qquad (203)$$

Thus, by continuity, there is $\varepsilon > 0$ such that $v(i,n,t) \in B_R(0)$ for some $\varepsilon > 0$ and all $t \in [0, T + \varepsilon]$. Now we can apply Lemma 1 for the interval $[T, T + \varepsilon]$. We note that by slot-wise linearity we have $D_{i,i}\hat{f} = 0$ and $\omega \geq 0$. Therefore, we get

$$\frac{d}{dt}\frac{1}{2}|\delta(i,n)|^2 \leq -\lambda|\delta(i,n)|^2 + C_1 K^{(I-1)/2}\sqrt{\ln(S)\frac{N}{S}}\delta_{\max}$$
$$+ C_1 K^{(I-1)/2}\ln(S)\sqrt{\frac{N^2}{S}}\delta_{\max}^2 + C_1\delta_{\max}^3. \qquad (204)$$

Now (with the same reasoning as in the proof of Theorem 4) there is a constant $C_2$ depending on $D$, $I$, and $M$ such that for

$$S \geq S_0 = \frac{(12)^2 C_1^2 N^2 K^{I-1}}{\lambda^2}\ln^2\left(\frac{(12)^2 C_1^2 N^2 K^{I-1}}{\lambda^2}\right) \qquad (205)$$

the bound

$$C_1 K^{(I-1)/2}\ln(S)\sqrt{\frac{N^2}{S}} \leq \frac{\lambda}{4} \qquad (206)$$

holds. Clearly, we get for a suitable constant $C_2$

$$S_0 \leq C_2 \frac{N^2 K^{I-1}}{\lambda^2}\ln^2(N/\lambda). \qquad (207)$$

Similarly, we get for

$$S \geq S_1 = \frac{2 \cdot 16^2 C_1^4 N K^{I-1}}{\lambda^4}\ln\left(\frac{2 \cdot 16^2 C_1^4 N K^{I-1}}{\lambda^4}\right) \qquad (208)$$

the bound

$$C_1 K^{(I-1)/2}\sqrt{\ln(S)\frac{N}{S}} \leq \frac{\lambda^2}{16C_1} \qquad (209)$$

holds. And we find

$$S_1 \leq C_3 \frac{N K^{I-1}}{\lambda^4}\ln(N/\lambda). \qquad (210)$$

Now we can continue to bound (204) for $S$ satisfying (205) and (208) and using (202) as follows

$$\frac{d}{dt}\frac{1}{2}|\delta(i,n,T)|^2 \leq -\lambda|\delta(i,n)|^2 + \frac{\lambda^2}{16C_1}\delta_{\max} + \frac{\lambda}{4}\delta_{\max}^2 + \frac{\lambda}{4}\delta_{\max}^2. \qquad (211)$$

Now, if $\delta_{\max}(T) \leq \lambda/(8C_1)$ then there is $\varepsilon > 0$ such that $\delta_{\max}(t) \leq \lambda/(4C_1)$ holds for $t \in [0, T + \varepsilon]$. Thus, we assume that $\delta_{\max}(T) \geq \lambda/(8C_1)$. Let $i \in [I]$, $n \in [N]$ be any index such that $|\delta(i,n,T)| = \delta_{\max}(T)$. Then we conclude that

$$\frac{d}{dt}\frac{1}{2}|\delta(i,n,T)|^2 \leq -\frac{\lambda}{2}\delta_{\max}^2 + \frac{\lambda^2}{16C_1}\delta_{\max} \leq -\frac{\lambda}{2}\frac{\lambda}{8C_1}\delta_{\max} + \frac{\lambda^2}{16C_1}\delta_{\max} = 0. \qquad (212)$$

This implies that

$$\frac{d}{dt^+}\frac{1}{2}\delta_{\max}(T)^2 \leq 0 \qquad (213)$$

from which we conclude that there is $\varepsilon > 0$ such that $\delta_{\max}(t) \leq \lambda/(4C_1)$ holds for $0 \leq t \leq T + \varepsilon$. In either case, we get a contradiction and thus $T = \infty$. In particular, we can apply Lemma 1 for all times $t \geq 0$. Suppose now that for some $t$

$$\delta_{\max}(t) \geq \frac{4C_1 K^{(I-1)/2}}{\lambda}\sqrt{\ln(S)\frac{N}{S}} \qquad (214)$$

then we conclude from Lemma 1 that

$$\frac{\mathrm{d}}{\mathrm{d}t}\frac{1}{2}\delta_{\max}^2(t) \leq -\frac{\lambda}{2}\delta_{\max}(t)^2 + C_1 K^{(I-1)/2}\sqrt{\ln(S)\frac{N}{S}}\delta_{\max}(t)$$
$$\leq -\frac{\lambda}{2}\delta_{\max}^2(t) + -\frac{\lambda}{4}\delta_{\max}^2(t) \leq \frac{\lambda}{4}\delta_{\max}^2(t). \tag{215}$$

Thus we conclude from Gronwall's inequality that

$$\delta_{\max}(t)^2 \leq \max\left(\delta_{\max}(0)^2 e^{-\lambda t/4}, \left(\frac{4C_1 K^{(I-1)/2}}{\lambda}\sqrt{\ln(S)\frac{N}{S}}\right)^2\right). \tag{216}$$

This ends the proof. $\qquad\square$

*Proof of Theorem 5.* The proof is the same as the proof of Theorem 8 above with the only exception that we do not need to show that $v(i, n, t) \in \Omega$ but this holds by assumption which makes the proof strictly simpler. $\qquad\square$

# G   Concentration of Datapoint Statistics Matrices

In this section, we prove high probability concentration bounds for certain matrices and vectors capturing the statistics of the dataset. They are used in Section E to extract the asymptotically dominating contribution of the loss gradient and therefore of the gradients of the embeddings under gradient flow. All concentration bounds derived in this section are a simple consequence of the general and standard matrix concentration bound stated in Lemma 7.

We start with the matrices $\boldsymbol{A}^{ij} \in \mathbb{R}^{N \times N}$ with entries

$$\boldsymbol{A}_{n_1,n_2}^{i,j} = |\{\boldsymbol{n} \in \mathcal{D} : \boldsymbol{n}_i = n_1, \boldsymbol{n}_j = n_2\}| =, \tag{217}$$

the matrix counting the appearance of a pair of tokens in slot $i$ and $j$ (this is closely related to the matrices $\boldsymbol{A}^{k,i,j}$ which we will consider below). Then we have the following result.

**Lemma 3.** *Let $\boldsymbol{A}^{i,j} \in \mathbb{R}^{N \times N}$ be as defined above. Assume that $S \geq 12N$. Then the following bound holds for $\eta > 0$*

$$\mathbb{P}_{\mathcal{D}}\left(\|\boldsymbol{A}^{i,j} - \mathbb{E}(\boldsymbol{A}^{i,j})\| \geq (1+\eta)\ln(S)\sqrt{\frac{S}{N}}\right) \leq S^{-\eta+\frac{3}{2}}. \tag{218}$$

*This implies*

$$\mathbb{P}_{\mathcal{D}}\left(\|\boldsymbol{A}^{i,j}\| \geq \frac{S}{N} + (1+\eta)\ln(S)\sqrt{\frac{S}{N}}\right) \leq S^{-\eta+\frac{3}{2}}. \tag{219}$$

*Proof.* As in the proof of Theorem 2, we use Poissonization, i.e., we consider dataset $\tilde{\mathcal{D}}$ generated by first sample $\bar{S} \sim \mathrm{Poi}(S)$ and then $\mathcal{D}_{\bar{S}} \sim \mathcal{U}([N]^I)^{\bar{S}}$. A uniform sample $\boldsymbol{n} \sim \mathcal{U}([N]^I)$ satisfies $\boldsymbol{n}_i = n_1$ and $\boldsymbol{n}_j = n_2$ with probability

$$p = \frac{1}{N^2}. \tag{220}$$

Therefore the distribution of $\boldsymbol{A}^{i,j}$ is

$$\boldsymbol{A}_{n_1,n_2}^{i,j} = \mathrm{Poi}\left(\frac{S}{N^2}\right) \tag{221}$$

and the entries are independent. We can now apply Lemma 7 where we use $\eta \ln(S)$ as $\eta$ in the statement of the lemma and find using $S \geq N$ that

$$\mathbb{P}_{\tilde{\mathcal{D}}}\left(\|\boldsymbol{A}^{i,j} - \mathbb{E}(\boldsymbol{A}^{i,j})\| \geq (1+\eta)\ln(S)\sqrt{\frac{S}{N}}\right) \leq 4NS^{-\eta}. \tag{222}$$

Let us define

$$r(S, N, \eta) = (1 + \eta) \ln(S) \sqrt{\frac{S}{N}} \tag{223}$$

Note that $\mathbb{P}(\bar{S} = S) = S^S e^{-S}/S! \geq 1/(3\sqrt{S})$. This implies

$$
\begin{aligned}
&\mathbb{P}_{\mathcal{D}} \left( \|\boldsymbol{A}^{i,j} - \mathbb{E}(\boldsymbol{A}^{i,j})\| \geq r(S, N, \eta) \right) \\
&= \mathbb{P}_{\tilde{\mathcal{D}}} \left( \|\boldsymbol{A}^{k,i,j} - \mathbb{E}(\boldsymbol{A}^{i,j})\| \geq r(S, N, \eta) | \bar{S} = S \right) \\
&= \frac{\mathbb{P}_{\tilde{\mathcal{D}}} \left( \|\boldsymbol{A}^{i,j} - \mathbb{E}(\boldsymbol{A}^{i,j})\| \geq r(S, N, \eta) \text{ and } \bar{S} = S \right)}{\mathbb{P}_{\tilde{\mathcal{D}}}(\bar{S} = S)} \\
&\leq 4 N S^{-\eta} \left( \frac{1}{3\sqrt{S}} \right)^{-1} \leq S^{-\eta + 3/2}.
\end{aligned}
\tag{224}
$$

Here we used $S \geq 12N$ in the last step. The relation (219) follows because $\|\mathbb{E}\boldsymbol{A}^{i,j}\| = NS/N^2 = S/N$. $\qquad\square$

Next we prove a similar but slightly more involved result for the matrices that capture a similar statistics as $\boldsymbol{A}^{i,j}$ but which in addition only consider $\boldsymbol{n}$ such that $\Pi(\boldsymbol{n}) = \boldsymbol{k}$. We thus consider any $\boldsymbol{k} \in [K]^I$ and any indices $i, j \in I$ which we consider fixed for now. Consider the sets $\mathcal{N}(i, \boldsymbol{k}) = \Pi_i^{-1}(\boldsymbol{k}_i), \mathcal{N}(j, \boldsymbol{k}) = \Pi_j^{-1}(\boldsymbol{k}_j) \subset [N]$. For a given dataset $\mathcal{D} = \{\boldsymbol{n}^1, \dots, \boldsymbol{n}^S\}$ we define the matrices $A^{k,i,j} \in \mathbb{R}^{\mathcal{N}(i,\boldsymbol{k}) \times \mathcal{N}(j,\boldsymbol{k})}$ by

$$A^{k,i,j}_{n_1,n_2} = |\{\boldsymbol{n} \in \mathcal{D} : \Pi(\boldsymbol{n}) = \boldsymbol{k}, \boldsymbol{n}_i = n_1, \boldsymbol{n}_j = n_2\}| \tag{225}$$

for $n_1 \in \mathcal{N}(i, \boldsymbol{k}), n_2 \in \mathcal{N}(j, \boldsymbol{k})$. Thus, the entries are the number of datapoints that are mapped to $\boldsymbol{k}$ by $\Pi$ and whose entries $i$ and $j$ are equal to $n_1$ and $n_2$. We emphasize that this essentially agrees with the definition in (115) given in Section D (except that the dimensions of the matrices do not agree, we address this below the lemma). Similar reasoning as in the previous result, Lemma 3, gives the following statement.

**Lemma 4.** *Let $A^{k,i,j} \in \mathbb{R}^{\mathcal{N}(i,\boldsymbol{k}) \times \mathcal{N}(j,\boldsymbol{k})}$ be as defined above. Assume that for all $i \in [I]$ and $k \in [K]$ the bound $N/(2K) \leq |\Pi_i^{-1}(k)| \leq 2N/K$ holds and let $S \geq 24N$. Then the following bound holds for $\eta > 0$*

$$\mathbb{P}_{\mathcal{D}} \left( \|A^{k,i,j} - \mathbb{E}(A^{k,i,j})\| \geq (1 + \eta) \ln(S) \max \left( 1, \left( \tfrac{2}{K} \right)^{(I-1)/2} \sqrt{\frac{S}{N}} \right) \right) \leq 3 S^{-\eta + 3/2}. \tag{226}$$

*Note that in particular for $K \geq 2$ we find for $S \geq 24N$ the bound*

$$\mathbb{P}_{\mathcal{D}} \left( \|A^{k,i,j} - \mathbb{E}(A^{k,i,j})\| \geq (1 + \eta) \ln(S) \sqrt{\frac{S}{N}} \right) \leq S^{-\eta + 3/2}. \tag{227}$$

*Proof.* The proof is essentially the same as the proof of Lemma 3 with some additional notational complications. Again, we consider a dataset $\tilde{\mathcal{D}}$ generated by first sampling $\bar{S} \sim \text{Poi}(S)$ and then $\mathcal{D}_{\bar{S}} \sim \mathcal{U}([N]^I)^{\bar{S}}$. Note that a uniform sample $\boldsymbol{n} \sim \mathcal{U}([N]^I)$ satisfies $\Pi(\boldsymbol{n}) = \boldsymbol{k}, \boldsymbol{n}_i = n_1$, and $\boldsymbol{n}_j = n_2$ for $n_1 \in \mathcal{N}(i, \boldsymbol{k})$ and $n_2 \in \mathcal{N}(j, \boldsymbol{k})$ with probability

$$p(\boldsymbol{k}, i, j) = \frac{1}{N^I} \frac{|\Pi^{-1}(\boldsymbol{k})|}{|\Pi_i^{-1}(\boldsymbol{k}_i)| \cdot |\Pi_j^{-1}(\boldsymbol{k}_j)|} \tag{228}$$

and thus the distribution of $A^{k,i,j}$ is

$$A^{k,i,j}_{n_1,n_2} = \text{Poi} \left( \frac{S}{N^I} \frac{|\Pi^{-1}(\boldsymbol{k})|}{|\Pi_i^{-1}(\boldsymbol{k}_i)| \cdot |\Pi_j^{-1}(\boldsymbol{k}_j)|} \right) =: \text{Poi}(\lambda(\boldsymbol{k}, i, j)) \tag{229}$$

for all $n_1 \in \mathcal{N}(i, \boldsymbol{k})$ and $n_2 \in \mathcal{N}(j, \boldsymbol{k})$ and the entries are independent. Note that the assumption $N/(2K) \leq |\Pi_i^{-1}(k)| \leq 2N/K$ implies that

$$\frac{|\Pi^{-1}(\boldsymbol{k})|}{|\Pi_i^{-1}(\boldsymbol{k}_i)| \cdot |\Pi_j^{-1}(\boldsymbol{k}_j)|} = \prod_{l \in [K] \setminus \{i,j\}} |\Pi_l^{-1}(\boldsymbol{k}_l)| \begin{cases} \leq \frac{2^{I-2} N^{I-2}}{K^{I-2}}, \\ \geq \frac{N^{I-2}}{2^{I-2} K^{I-2}} \end{cases} \tag{230}$$

which implies

$$\frac{1}{(2K)^{I-2}} \frac{S}{N^2} \leq \lambda(\boldsymbol{k}, i, j) \leq \left(\frac{2}{K}\right)^{I-2} \frac{S}{N^2}. \tag{231}$$

Now we apply Lemma 7 where we use $\eta \ln(S)$ as $\eta$ in the statement of the lemma and find that

$$\mathbb{P}_{\tilde{\mathcal{D}}} \left( \|A^{\boldsymbol{k},i,j} - \mathbb{E}(A^{\boldsymbol{k},i,j})\| \geq (1+\eta) \ln(S) \max\left(1, \left(\frac{2}{K}\right)^{(I-1)/2} \sqrt{\frac{S}{N}}\right) \right) \leq \frac{8N}{K} S^{-\eta}. \tag{232}$$

Let us set

$$r(S, N, K, I, \eta) = (1+\eta) \ln(S) \max\left(1, \left(\frac{2}{K}\right)^{(I-1)/2} \sqrt{\frac{S}{N}}\right) \tag{233}$$

Again, $\mathbb{P}(\bar{S} = S) = S^S e^{-S}/S! \geq 1/(3\sqrt{S})$. This implies

$$\begin{aligned}
\mathbb{P}_{\mathcal{D}} &\left( \|A^{\boldsymbol{k},i,j} - \mathbb{E}(A^{\boldsymbol{k},i,j})\| \geq r(S, N, K, I, \eta) \right) \\
&= \mathbb{P}_{\tilde{\mathcal{D}}} \left( \|A^{\boldsymbol{k},i,j} - \mathbb{E}(A^{\boldsymbol{k},i,j})\| \geq r(S, N, K, I, \eta) | \bar{S} = S \right) \\
&= \frac{\mathbb{P}_{\tilde{\mathcal{D}}} \left( \|A^{\boldsymbol{k},i,j} - \mathbb{E}(A^{\boldsymbol{k},i,j})\| \geq r(S, N, K, I, \eta) \text{ and } \bar{S} = S \right)}{\mathbb{P}_{\tilde{\mathcal{D}}}(\bar{S} = S)} \\
&\leq \frac{8N}{K} S^{-\eta} \left(\frac{1}{3\sqrt{S}}\right)^{-1} \leq S^{-\eta+3/2}.
\end{aligned} \tag{234}$$

Here we used $S \geq 24N$ in the last step.

$\square$

We also consider the matrix $\boldsymbol{A}^{\boldsymbol{k},i,j} \in \mathbb{R}^{IN \times IN}$ already defined in (115), which we equivalently obtain by embedding $A^{\boldsymbol{k},i,j}$ suitable, i.e, we have

$$\boldsymbol{A}^{\boldsymbol{k},i,j}_{(i_1,n_1),(j_1,n_2)} = \begin{cases} A^{\boldsymbol{k},i,j}_{n_1,n_2} & \text{if } i_1 = i, j_1 = j \text{ and } n_1 \in \mathcal{N}(i, \boldsymbol{k}), n_2 \in \mathcal{N}(j, \boldsymbol{k}) \\ 0 & \text{otherwise.} \end{cases} \tag{235}$$

Since $A^{\boldsymbol{k},i,j}$ is a submatrix of $\boldsymbol{A}^{\boldsymbol{k},i,j}$ we find

$$\|\boldsymbol{A}^{\boldsymbol{k},i,j} - \mathbb{E}\boldsymbol{A}^{\boldsymbol{k},i,j}\| = \|A^{\boldsymbol{k},i,j} - \mathbb{E}A^{\boldsymbol{k},i,j}\| \tag{236}$$

so that Lemma 4 applies to $\boldsymbol{A}^{\boldsymbol{k},i,j}$.

We also need a simpler concentration statement for the frequency of datapoints $\boldsymbol{n}^s$ such that $\boldsymbol{n}_i = n$. We define the vectors $\boldsymbol{B}^i \in \mathbb{R}^N$ by

$$\boldsymbol{B}^i_n = |\{\boldsymbol{n} \in \mathcal{D} : \boldsymbol{n}_i = n\}|. \tag{237}$$

We need a simple upper bound on the vectors $\boldsymbol{B}^i$.

**Lemma 5.** *The following bound holds for $\eta > 0$*

$$\mathbb{P}\left(\boldsymbol{B}^i_n \geq \frac{S}{N} + \max(2\sqrt{\eta \ln(S) S N^{-1}}, 4/3 \eta \ln(S))\right) \leq S^{-\eta}. \tag{238}$$

*Proof.* Note that $\boldsymbol{B}^i_n \sim \text{Bin}(S, N^{-1})$. The variance of a $\text{Ber}(p)$ variable is $p(1-p) \leq p$. Bernstein's one-sided inequality then reads

$$\mathbb{P}\left(\text{Bin}(S, N^{-1}) - SN^{-1} \geq t\right) \leq \exp\left(-\frac{t^2}{2SN^{-1} + 2t/3}\right) \leq \exp\left(-\min\left(\frac{t^2}{4SN^{-1}}, 3t/4\right)\right). \tag{239}$$

Now we apply this bound with $t = \max(2\sqrt{\eta \ln(S)SN^{-1}}, 4/3\eta \ln(S))$ which implies

$$\mathbb{P}\left(\mathrm{Bin}(S, N^{-1}) - \frac{S}{N} \geq \max\left(2\sqrt{\eta \ln(S)\frac{S}{N}}, 4/3\eta \ln(S)\right)\right) \leq S^{-\eta}. \tag{240}$$

$\square$

As before for the $A$ matrices we also need an extension of the previous result to a setting where we in addition require $\Pi(\boldsymbol{n}) = \boldsymbol{k}$ for a given $\boldsymbol{k}$ and $n \in [N]$. Recall that $\mathcal{N}(i, \boldsymbol{k}) = \Pi_i^{-1}(\boldsymbol{k}_i)$. Then we consider the vector $B^{\boldsymbol{k},i} \in \mathbb{R}^{\mathcal{N}(i,\boldsymbol{k})}$ given by

$$B_n^{\boldsymbol{k},i} = |\{\boldsymbol{n} \in \mathcal{D} : \Pi(\boldsymbol{n}) = \boldsymbol{k}, \, \boldsymbol{n}_i = n\}| = |\mathcal{D}^{\boldsymbol{k},i,n}|. \tag{241}$$

Again, this agrees with the definition of $B$ given in (114). The following Lemma holds.

**Lemma 6.** *Let $B^{\boldsymbol{k},i} \in \mathbb{R}^{\mathcal{N}(i,\boldsymbol{k})}$ be as defined above. Assume that for all $i \in [I]$ and $k \in [K]$ the bound $|\Pi_i^{-1}(k)| \leq 2N/K$ holds. Then the following bound holds for $\eta > 0$ and $n \in \mathcal{N}(i, \boldsymbol{k})$*

$$\mathbb{P}\left(|B_n^{\boldsymbol{k},i} - \mathbb{E}(B_n^{\boldsymbol{k},i})| \geq \max\left(2\left(\frac{2}{K}\right)^{(I-1)/2}\sqrt{\eta \ln(S)\frac{S}{N}}, \frac{4}{3}\eta \ln(S)\right)\right) \leq 2S^{-\eta}. \tag{242}$$

*Note that viewing $I$ and $K$ as constant we get for $S \geq N \cdot \ln(N)$ the bound*

$$\mathbb{P}_{\mathcal{D}}\left(|B_n^{\boldsymbol{k},i} - \mathbb{E}(B_n^{\boldsymbol{k},i})| \geq C_{I,K}\sqrt{\eta \ln(S)\frac{S}{N}}\right) \leq 2S^{-\eta}. \tag{243}$$

*Proof.* The proof is along the lines of Lemma 5 but slightly more technical. Note that the entries of $B^{\boldsymbol{k},i}$ are distributed according to $\mathrm{Bin}(S, p(\boldsymbol{k}, i))$ where

$$p(\boldsymbol{k}, i) = \frac{|\Pi^{-1}(\boldsymbol{k})|}{N^I|\Pi_i^{-1}(\boldsymbol{k}_i)|}. \tag{244}$$

Note that by assumption

$$p(\boldsymbol{k}, i) \leq \left(\frac{2N}{K}\right)^{I-1} N^{-I} = \frac{1}{N}\left(\frac{2}{K}\right)^{I-1}. \tag{245}$$

Since the variance of a $\mathrm{Ber}(p)$ variable is $p(1-p) \leq p$ Bernstein's inequality then implies

$$\mathbb{P}(|\mathrm{Bin}(S, p(\boldsymbol{k}, i)) - S \cdot p(\boldsymbol{k}, i)| \geq t) \leq 2\exp\left(-\frac{t^2}{2Sp(\boldsymbol{k}, i) + 2t/3}\right)$$
$$\leq 2\exp\left(-\min\left(\frac{t^2}{4Sp(\boldsymbol{k}, i)}, 3t/4\right)\right). \tag{246}$$

Now we apply this bound with $t = \max(2\sqrt{\eta \ln(S)Sp(\boldsymbol{k}, i)}, 4/3\eta \ln(S))$ which implies

$$\mathbb{P}\left(|\mathrm{Bin}(S, p(\boldsymbol{k}, i)) - S \cdot p(\boldsymbol{k}, i)| \geq \max(2\sqrt{\eta \ln(S)Sp(\boldsymbol{k}, i)}, 4/3\eta \ln(S))\right) \leq 2S^{-\eta}. \tag{247}$$

Applying (245) we find

$$\mathbb{P}\left(|\mathrm{Bin}(S, p(\boldsymbol{k}, i)) - S \cdot p(\boldsymbol{k}, i)| \geq \max\left(2\left(\frac{2}{K}\right)^{(I-1)/2}\sqrt{\eta \ln(S)\frac{S}{N}}, \frac{4}{3}\eta \ln(S)\right)\right) \leq 2S^{-\eta}. \tag{248}$$

To conclude that (243) holds, we just need to show that for $S \geq N \ln(N)$ the expression $S/(N \ln(S))$ is bounded which follows by monotonicity of $S/\ln(S)$ in $S$. $\square$

Recall that we introduced $\boldsymbol{B}^{\boldsymbol{k},i} \in \mathbb{R}^{[N]}$ in (114) and then

$$\boldsymbol{B}_n^{\boldsymbol{k},i} = \begin{cases} B_n^{\boldsymbol{k},i} & \text{if } n \in \mathcal{N}(i, \boldsymbol{k}) \\ 0 & \text{otherwise.} \end{cases} \tag{249}$$

For our analysis we want to summarize all those concentration bounds in one result. To simplify the statement, we only consider our main regime of interest.

**Theorem 9.** *Assume that the bound $N/(2K) \leq |\Pi_i^{-1}(k)| \leq 2N/K$ holds for all $i \in [I]$ and $k \in [K]$. Assume that*

$$S \geq N \max\left(24, 3(K/2)^{I-1}\ln(N), 4IK^{I/2}\right). \tag{250}$$

*Then with probability at least $1 - S^{-1}$ the following bounds hold simultaneously for all $\boldsymbol{k} \in [K]^I$, $i, j \in [I]$*

$$\|\boldsymbol{A}^{i,j} - \mathbb{E}(\boldsymbol{A}^{i,j})\| \leq 5\ln(S)\sqrt{\frac{S}{N}}, \tag{251}$$

$$\|\boldsymbol{A}^{\boldsymbol{k},i,j} - \mathbb{E}(\boldsymbol{A}^{\boldsymbol{k},i,j})\| \leq 6\ln(S)\left(\tfrac{2}{K}\right)^{(I-1)/2}\sqrt{\frac{S}{N}}, \tag{252}$$

$$\boldsymbol{B}_n^i \leq \frac{S}{N} + 4\sqrt{\frac{\ln(S)S}{N}}, \tag{253}$$

$$|\boldsymbol{B}_n^{\boldsymbol{k},i} - \mathbb{E}(\boldsymbol{B}_n^{\boldsymbol{k},i})| \leq 4\left(\frac{2}{K}\right)^{(I-1)/2}\sqrt{\ln(S)\frac{S}{N}}. \tag{254}$$

*Proof.* Applying Lemma 3 with $\eta = 4$ we obtain that the first bound holds with probability at least $1 - S^{-2}$ (here we used $S \geq 12N$). Similarly, we obtain for $S \geq (K/2)^{I-1}N$

$$\max\left(1, \left(\tfrac{2}{K}\right)^{(I-1)/2}\sqrt{\frac{S}{N}}\right) = \left(\tfrac{2}{K}\right)^{(I-1)/2}\sqrt{\frac{S}{N}}. \tag{255}$$

Setting $\eta = 5$ in Lemma 4 we get that the second bound holds (by the union bound) with probability at least $1 - 3I^2K^IS^{-3}$ for all $i, j \in [I]$ and $\boldsymbol{k} \in [K]^I$. For $S \geq \alpha N\ln(N)$ and $N \geq \alpha$ we find

$$\frac{S}{\ln(S)} \geq \frac{\alpha N\ln(N)}{\ln(\alpha\ln(N)N)} \geq \frac{\alpha N\ln(N)}{3\ln(N)} \geq \frac{\alpha}{3}N \tag{256}$$

which implies for $\eta = 2$ and $S \geq 3N\ln(N)$

$$\max\left(2\sqrt{\eta\ln(S)\frac{S}{N}}, 4/3\eta\ln(S)\right) \leq 4\sqrt{\ln(S)\frac{S}{N}} \tag{257}$$

and thus applying Lemma 5 with $\eta = 2$ implies that the third bound holds with probability at least $1 - S^{-2}$. Finally, we find for $S \geq 3(K/2)^{I-1}N\ln(N)$ and $\eta = 3$ the bound

$$\max\left(2\left(\frac{2}{K}\right)^{(I-1)/2}\sqrt{\eta\ln(S)\frac{S}{N}}, \frac{4}{3}\eta\ln(S)\right) \leq 4\left(\frac{2}{K}\right)^{(I-1)/2}\sqrt{\ln(S)\frac{S}{N}}. \tag{258}$$

Lemma 6 implies that the last bound holds with probability at least $1 - 2IK^IS^{-3}$. All bounds simultaneously then hold with probability at least

$$1 - S^{-2} - 3I^2K^IS^{-3} - S^{-2} - 2IK^IS^{-3} \geq 1 - 2S^{-2} - 2\cdot\frac{1}{4}S^{-1} \geq 1 - S^{-1}. \tag{259}$$

$\square$

# H  Spectral bounds for Random Matrices

In the derivation of the concentration bounds in the previous section, we needed concentration bounds for random matrices whose entries follow a Poisson distribution. In this section we provide the required result. Lemma 7 below should be folklore, but we did not find an exact reference so we provide a proof based on standard concentration results for random matrices. Let us first state the general result.

**Theorem 10** (Corollary 3.7 in [26]). *Let $X_k$ be a sequence of independent random symmetric matrices with dimension $d$. We assume that there is a function $g : (0, \infty) \to [0, \infty]$ and symmetric matrices $A_k$ such that*

$$\mathbb{E}e^{\theta X_k} \leq e^{g(\theta)A_k} \quad for \ \theta > 0. \tag{260}$$

*Define*

$$\rho := \lambda_{\max}(\sum_k A_k). \tag{261}$$

*Then for all $t \in \mathbb{R}$ the following bound holds*

$$\mathbb{P}\left(\lambda_{\max}\left(\sum_k X_k\right) \geq t\right) \leq d \cdot \inf_{\theta > 0} e^{-\theta t + g(\theta)\rho}. \tag{262}$$

We now apply the previous concentration bound to our specific setting of interest.

**Lemma 7.** *Consider a random matrix $A \in \mathbb{R}^{d_1 \times d_2}$ whose entries are independent random variables with $A_{ij} \sim \mathrm{Poi}(\lambda)$ for some $\lambda > 0$. Then the following bound holds for all $\eta \geq 0$*

$$\mathbb{P}\left(\|A - \mathbb{E}(A)\| \geq (\eta + 1) \max\left(\sqrt{\max(d_1, d_2)\lambda}, 1\right)\right) \leq 2(d_1 + d_2)e^{-\eta}. \tag{263}$$

*Proof.* To bound the norm of this non-symmetric matrix we use the usual approach to consider

$$Q = \begin{pmatrix} \mathbf{0}_{d_1 \times d_1} & A - \mathbb{E}(A) \\ A^\top - \mathbb{E}(A)^\top & \mathbf{0}_{d_1 \times d_1} \end{pmatrix}. \tag{264}$$

Consider the index set

$$I = \{(i, j) : i \in [d_1], \ j \in \{d_1 + 1, \ldots, d_2 + d_2\}\} \tag{265}$$

then we can write

$$Q \stackrel{\mathcal{D}}{=} \sum_{(i,j) \in I} (N_{i,j} - \lambda)(E_{i,j} + E_{j,i}) \tag{266}$$

where $E_{i,j}$ is the matrix whose entry $(i, j)$ is 1 and all other entries vanish and $N_{i,j} \sim \mathrm{Poi}(\lambda)$. Let $X_{ij} = (N_{i,j} - \lambda)(E_{i,j} + E_{j,i})$. Note that by induction one directly finds

$$(E_{i,j} + E_{j,i})^k = \begin{cases} E_{ii} + E_{jj} & \text{if } k \text{ is even,} \\ E_{i,j} + E_{j,i} & \text{if } k \text{ is odd.} \end{cases} \tag{267}$$

In any case we get $\pm(E_{i,j} + E_{j,i})^k \leq E_{i,i} + E_{j,j}$ in the sense of symmetric matrices and we set $A_{ij} = E_{i,i} + E_{j,j}$. Now we obtain for any $\theta \in \mathbb{R}$ the relation

$$\begin{aligned}
\mathbb{E}e^{\theta X_{ij}} &= A_{ij} \sum_{k=0}^{\infty} \frac{\theta^{2k}\mathbb{E}(\mathrm{Poi}(\lambda) - \lambda)^{2k}}{(2k)!} + (E_{i,j} + E_{j,i}) \sum_{k=0}^{\infty} \frac{\theta^{2k+1}\mathbb{E}(\mathrm{Poi}(\lambda) - \lambda)^{2k+1}}{(2k+1)!} \\
&\leq A_{ij}\left(\sum_{k=0}^{\infty} \frac{\theta^{2k}\mathbb{E}(\mathrm{Poi}(\lambda) - \lambda)^{2k}}{(2k)!} + \left|\sum_{k=0}^{\infty} \frac{\theta^{2k+1}\mathbb{E}(\mathrm{Poi}(\lambda) - \lambda)^{2k+1}}{(2k+1)!}\right|\right).
\end{aligned} \tag{268}$$

Note that the first summand is invariant under $\theta \to -\theta$ while the term in absolute values changes its sign. This implies that

$$\begin{aligned}
\left(\sum_{k=0}^{\infty} \frac{\theta^{2k}\mathbb{E}(\mathrm{Poi}(\lambda) - \lambda)^{2k}}{(2k)!} + \left|\sum_{k=0}^{\infty} \frac{\theta^{2k+1}\mathbb{E}(\mathrm{Poi}(\lambda) - \lambda)^{2k+1}}{(2k+1)!}\right|\right) \\
= \max\left(\sum_{k=0}^{\infty} \frac{\theta^k\mathbb{E}(\mathrm{Poi}(\lambda) - \lambda)^k}{k!}, \sum_{k=0}^{\infty} \frac{(-\theta)^k\mathbb{E}(\mathrm{Poi}(\lambda) - \lambda)^k}{k!}\right) \\
= \max\left(\mathbb{E}e^{\theta(\mathrm{Poi}(\lambda) - \lambda)}, \mathbb{E}e^{-\theta(\mathrm{Poi}(\lambda) - \lambda)}\right) \\
= \max\left(e^{\lambda(e^\theta - 1) - \lambda\theta}, e^{\lambda(e^{-\theta} - 1) + \lambda\theta}\right) = e^{\lambda(e^{|\theta|} - 1) - \lambda|\theta|}.
\end{aligned} \tag{269}$$

Here we used the moment generating function of the Poisson distribution and in the last step that $\lambda \geq 0$ and then for $\theta \geq 0$

$$e^\theta - e^{-\theta} = 2\sinh(\theta) \geq 2\theta. \tag{270}$$

Thus we infer from (268) and (269) that

$$\mathbb{E}e^{\theta X_{ij}} \leq e^{\lambda(e^{|\theta|}-1)-\lambda|\theta|}A_{ij} = e^{\left(\lambda(e^{|\theta|}-1)-\lambda|\theta|\right)A_{ij}} := e^{g_\lambda(|\theta|)A_{ij}}. \tag{271}$$

using that $A_{ij}^k = A_{ij}$ in the second step. Note that here

$$\sum_{(i,j)\in I} A_{ij} = \sum_{(i,j)\in I} E_{i,i} + E_{j,j} = \sum_{i=1}^{d_1} d_2 E_{i,i} + \sum_{i=d_1+1}^{d_2+d_2} d_1 E_{i,i} \tag{272}$$

which implies

$$\rho = \lambda \left( \sum_{(i,j)\in I} A_{ij} \right) = \max(d_1, d_2). \tag{273}$$

Thus we can apply Theorem 10 and get

$$\mathbb{P}\left( \lambda_{\max}\left( \sum_{(i,j)\in I} X_{ij} \right) \geq t \right) \leq (d_1 + d_2) \cdot \inf_{\theta > 0} e^{-\theta t + g(\theta)\rho} \tag{274}$$

$$\leq (d_1 + d_2) \cdot \inf_{\theta > 0} e^{-\theta t + \lambda \max(d_1, d_2)(e^\theta - 1 - \theta)}.$$

If $\max(d_1, d_2)\lambda \leq 1$ we apply this bound with $t = \eta + 1$ and $\theta = 1$ and find

$$\mathbb{P}\left( \lambda_{\max}\left( \sum_{(i,j)\in I} X_{ij} \right) \geq (1 + \eta) \right) \leq (d_1 + d_2) \cdot \exp\left( -(1 + \eta) + \lambda \max(d_1, d_2)(e^1 - 2) \right)$$

$$\leq (d_1 + d_2)e^{-\eta}. \tag{275}$$

If $\max(d_1, d_2)\lambda \geq 1$ we set $\theta = \sqrt{\max(d_1, d_2)\lambda}^{-1} \leq 1$ and $t = (\eta + 1)\sqrt{\max(d_1, d_2)\lambda}$ and find using that for $0 \leq \theta \leq 1$ the bound $e^\theta - 1 - \theta \leq \theta^2$ holds

$$\mathbb{P}\left( \lambda_{\max}\left( \sum_{(i,j)\in I} X_{ij} \right) \geq (\eta + 1)\sqrt{\max(d_1, d_2)\lambda} \right) \leq \exp\left( -\theta t + \lambda \max(d_1, d_2)\theta^2 \right)$$

$$\leq (d_1 + d_2) \cdot \exp\left( -(1 + \eta)\frac{\sqrt{\max(d_1, d_2)\lambda}}{\sqrt{\max(d_1, d_2)\lambda}} + \lambda \max(d_1, d_2)\sqrt{\max(d_1, d_2)\lambda}^{-2} \right) \tag{276}$$

$$= (d_1 + d_2)e^{-\eta}.$$

Thus we can combine both cases and obtain the relation

$$\mathbb{P}\left( \lambda_{\max}(Q) \geq (\eta + 1)\max\left( \sqrt{\max(d_1, d_2)\lambda}, 1 \right) \right) \leq (d_1 + d_2)e^{-\eta}. \tag{277}$$

We observe that by (271)

$$\mathbb{E}e^{\theta(-X_{ij})} = \mathbb{E}e^{-\theta X_{ij}} \leq e^{g_\lambda(|\theta|)A_{ij}}. \tag{278}$$

Thus the same reasoning shows that

$$\mathbb{P}\left( \lambda_{\min}(Q) \leq -(\eta + 1)\max\left( \sqrt{\max(d_1, d_2)\lambda}, 1 \right) \right) \leq (d_1 + d_2)e^{-\eta}. \tag{279}$$

and since $\|A - \mathbb{E}A\| = \|Q\| = \max |\lambda_i(Q)|$ the claim follows by applying the union bound over (277) and (279). $\qquad\square$

# I Kronecker Products

Let us here state some basic properties of the Kronecker product of matrices. For two matrices $A \in \mathbb{R}^{n \times m}$ and $B \in \mathbb{R}^{p \times q}$ the Kronecker product is defined by

$$A \otimes B = \begin{pmatrix} a_{11}B & \cdots & a_{1m}B \\ \vdots & \ddots & \vdots \\ a_{n1}B & \cdots & a_{nm}B \end{pmatrix} \in \mathbb{R}^{np \times mq}. \tag{280}$$

We will need the property that the operator norm satisfies

$$\|A \otimes B\| \leq \|A\| \cdot \|B\|. \tag{281}$$

Note that we use the notation $v \otimes w = vw^\top$ to denote the outer product. Formally we have $vw^\top = v \otimes w^\top$ but it is convenient and common to drop the transposed here.

