# OpenReview forum: "Learning Partitions from Context"
_NeurIPS.cc/2024/Conference — NeurIPS 2024 poster_

### Official Review · Reviewer_w8Q8 · 2024-07-08

**Soundness:** 2
**Presentation:** 2
**Contribution:** 3
**Rating:** 5
**Confidence:** 2

**Summary:**

This paper studies a learning problem where we are given sequences of tokens and the task is to predict a label. At each step of the sequence, there is a different clustering of the tokens into classes and the output label only depends on the sequence of classes corresponding to the tokens. The task is then to estimate the clusterings at each of the time steps. The authors present several information-theoretic and algorithmic results about this problem and propose an embedding-based method for learning the clusterings.

**Strengths:**

The authors propose an interesting and novel algorithmic problem.

The analysis of Section 3 is interesting. The authors derive some interesting limits regarding what is achievable in this problem.

**Weaknesses:**

The usage of the word "interact" in the abstract is rather vague.

The paper is poorly written with many language errors. Especially the paragraph from lines 69 to 81 is incomprehensible.

In Theorem 1, the $M$ is not defined. Should this be $I$?

The proposed embedding approach seems somewhat unnatural for this discrete problem.

Gradient descent seems like a rather crude approach.

**Questions:**

Is assuming knowledge of $k$ realistic?

On line 132, it is written that order $N^I$ samples are needed to learn an unstructured map. Don't you mean $N^I\cdot I\log N$? For a uniform discrete random variable with $m$ possible values, it takes $O(m\log m)$ samples before all possible variables are encountered.

From section 4, it seems that the method needs at least $O(DMI)$ memory. Is this scalable?

**Limitations:**

Assumption 1 is quite strong. In practice, one often encounters highly imbalanced clusterings.

The paper does not include experiments on real data.

---

> ### Author Rebuttal · Authors · 2024-08-06
>
> We thank the reviewer for their helpful review.
>
> Below, we comment on the weaknesses pointed out in the review and reply to the questions raised.
>
> > The usage of the word "interact" in the abstract is rather vague.
>
> - We will make the abstract a bit more verbose and clarify that
>  the interaction is through a function which has the property that it is invariant under
>  the exchange of tokens from the same class.
>
>  > The paper is poorly written with many language errors. Especially the paragraph from lines 69 to 81 is incomprehensible.
>
>  - We apologize that the writing of the paper (actually mostly section 1 and 2) could be strengthened, and we have used the time since submission to improve the writing of the paper, and we will make sure that the paper is carefully proofread before publication. In particular, we rewrote the paragraph from line 69-81 based on your feedback as follows:
>
> In this section, we illustrate our problem with an example and define the setup more formally.
> Consider the set of all animals. Those can be grouped into classes
> such as mammals, birds, or reptiles (in fact there is a rich hierarchical structure which we ignore here).
> Those groups were conceived by findings sets of animals that share many properties.
> Once these groups are found, we can predict unobserved properties by first identifying the cluster to which an animal belongs and then predict that the property is shared with animals in the same cluster.
> Note that this is a specific instance of a general problem in scientific inference where we want
> to uncover a hidden grouping of similar entities from sparse observations about these entities.
>
> Here our main motivation, however, stems from the analysis of large language models
> where a similar problem arises implicitly during training.
> They are trained by next token prediction, so we do not expect them to learn structure
> by deductive reasoning such as
> cows are mammals, and mammals have lungs, so cows have lungs.
>  Instead, their learning signal is whether a token can be replaced by another token for a given context. Thus, it is a natural question whether gradient descent-based training
> on token embeddings can uncover a hidden cluster structure of the data.
> Note that if the hidden structure is recovered, then generalization to unseen prompts is possible.
>
> We now define our formal setup...
>
> ---
> ---
> > In Theorem 1, the $M$ is not defined. Should this be $I$?
>
>
>  - Yes, you are correct that all $M$ should be $I$, thanks for pointing this out.
>
> > The proposed embedding approach seems somewhat unnatural for this discrete problem. Gradient descent seems like a rather crude approach.
>
>  - Let us clarify the motivation of this paper. One key motivation is to understand how
>  large language models learn complex relations between entities. While our setting is clearly a substantial simplification, this is necessary to make progress on the theoretical side. In this context, it is indeed natural to consider gradient-based training and
>  continuous embeddings of the discrete tokens because this is currently the main paradigm in language modelling (and also used in vision systems to some extent). On the other hand,
>  it is true that when considering the problem in isolation, it is more natural to
>  consider different approaches, such as a reduction to other combinatorial problems.
>  In our specific setting, it is possible to reduce the problem to a constraint satisfaction problem, and we refer to our response to R. `8unk` for details. Note that it is not trivial to come up with guarantees on the run-time for direct approaches (indeed, they need to be super-polynomial for general instances).
>
> > Is assuming knowledge of $k$ realistic?
>
> - It is correct that $K$ can be unknown in practice.  To deal with this in practice
> we can first start with a rather large estimate of $K$
> and then try to merge clusters to find a minimal representation. Note that the
> gradient-based approach does not require prior knowledge of $K$.
>
> > On line 132, it is written that order  $N^I$ samples are needed to ...
>
> - It is true that we here ignored the term $\ln(N^I)$ for simplicity because this is an informal argument where the logarithmic term is not relevant. To avoid confusion, we will instead use the correct expression.
>
> > From section 4, it seems that the method needs at least $O(NDI)$ memory. Is this scalable?
>
> - Memory of $DNI$ seems to be scalable to rather large settings. Taking in particular into account that large models are currently trained on huge datasets, i.e., $S$ is enormous, the memory cost of the method is much smaller than the size of the dataset.
>
> > Assumption 1 is quite strong. In practice, one often encounters highly imbalanced clusterings.
>
> This is true. We consider imbalanced datasets in Theorem 6 where we investigate the effect on the sample complexity. For the gradient-based analysis, we focused on balanced clustering for simplicity. However, an extension to an imbalanced setting would be possible but require additional notational burden.
>
> ---
> ---
> We hope that this rebuttal clarifies the motivation of our paper and addresses the concerns raised by the reviewer. If our rebuttal is satisfactory, we would appreciate it if the score can be upgraded
> to reflect this.
> We are happy to address any further questions or concerns and welcome additional feedback.

---

> > ### Comment · Reviewer_w8Q8 · 2024-08-12
> >
> > I thank the authors for their rebuttal, which addresses most of my concerns. I will increase my score. The paper still lacks experiments on real data. Adding experiments on real data would really help make the proposed problem setting more interpretable.

---

### Official Review · Reviewer_8unk · 2024-07-12

**Soundness:** 4
**Presentation:** 4
**Contribution:** 3
**Rating:** 6
**Confidence:** 4

**Summary:**

This article studies the properties of tokens, i.e word embeddings in NLP, that are grouped in a small number of clusters. It proposes and analyzes a relatively simple model which is basically a composition of a clustering and a real function, but that shares many similarities with real world complex models. The authors tackle the problem from both complexity theoretic and information theoretic viewpoints (section 3), where they showed that $O(N\log(N))$ samples are sufficient to recover the clustering map pi, and $O(N^2\log(N))$ are needed in gradient-based methods. They also show that tokens belonging to the same class interact in approximately the same way with other tokens, and that their embeddings, i.e their vector representation, have the same dynamics (section 4).

**Strengths:**

1. The article gives very important insights on why word embeddings used in Large Language Models are successful. The authors made a solid theoretical study about the stated problem, and they covered it from different viewpoints. They also proved that we can recover the partition of data with a very good precision and with a relatively small number of data samples.
2. This article can be a good starting point to investigate on the theoretical interpretability of word embeddings in LLMs for instance.
Also, the paper is coherent and the authors interpreted and commented all their results in a clear manner so that researchers from different backgrounds can read it without being lost in the formulas.

**Weaknesses:**

1. The authors also show that it is possible in theory to recover the partition map pi, but they don’t provide a clear algorithm or method on how to do so when their assumptions are satisfied. They also specified that it can be a very hard problem to determine this partition, which reduces the applicability of their findings, unfortunately.
2. Some assumptions made in this paper are quite restrictive, for example, the authors stated that one of the necessary conditions to get a stable cluster structure is to have $S > N^2 \log(N)$, which can be still a high number of samples: take the example of OpenAI’s tiktoken tokenizer “base200a” that has more than $2.10^5$ in its vocabulary size.
3. This work lacks experimentation as these findings can be applied to some simple real-world examples, like to an encoder-only model that performs sentiment analysis ($f \in \{0, 1\}$), but the authors kept it theoretical.

**Questions:**

I am curious why the authors couldn’t apply their approach to recover the clustering function $\Pi$ for a simple real-world example, for example with an Encoder-only transformer (e.g. with one attention block) that performs some binary classification for instance.

**Limitations:**

The authors have properly addressed the limitations of their work.

---

> ### Author Rebuttal · Authors · 2024-08-06
>
> We thank the reviewer for their insightful review.
> We would like to briefly comment on the weaknesses and the question raised.
>
>
> > The authors provide no algorithm
>
> - This is a valid point, and we will clarify this in the paper: It does not seem too easy to design a problem-specific algorithm. However, we can reduce the problem to other combinatorial problems for which efficient solvers exist.
> In our specific case, the problem can be reduced to a constraint satisfaction problem (see details below). Standard solvers can solve this problem, but might be slow.
> However, as Theorem 3 shows, this cannot be avoided (unless $P=NP$).
> Our results in Section 4 are in line with the general recent trend to depart from worst-case analysis, and here we show that efficient algorithms exist under the more restrictive assumptions on the problem setting. Indeed, our viewpoint here is that for typical real-world settings the data will not be adversarial, and the sample size large and in this case, the problem much simpler than the hardest problem instances.
> Moreover, we show that then gradient-based training can succeed (in polynomial time).
> We are not aware of results that would allow us to prove that the approach outlined above (i.e., applying general purpose solvers to a constraint satisfaction problem reduction) is, in fact, efficient on these simpler instances, but empirically this is often true.
>
> We will add this discussion to the updated paper.
>
> > $N^2$ can still be large if $N$ is large.
>
> - It is true that ideally we want a linear scaling (up to logarithmic terms) in $N$ which would match the information theoretic bound.  We believe that the exponent of the $N^2$ term can be slightly improved, at least for slotwise-linear
> $\hat{f}$ at the price of weaker guarantees, but this we leave for future work.
> Note that while large language models may use tokenizers with $O(10^5)$ tokens, they are
> also trained on $O(10^{12})$ unique tokens, so the scaling in our results is not too far away from real-world settings.
>
> > Missing experiments.
>
> - There is no particular reason for not applying this approach to recover clusters in real-world data, we did not try this. This was meant as a theory paper, and we feel its content is already quite dense and a solid contribution without experiments. However, the suggestion to confirm the findings with an experiment on sentiment analysis is intriguing, and we will try to implement this in the future.
>
>
> We hope that this addresses the reviewers' concerns and that they consider improving their score if they find it satisfactory.  We are happy to address any further questions or concerns and welcome additional feedback.
>
> ---
> ---
> ---
>
> Details on a general algorithm for the considered setting:
>
> Let us sketch how to reduce our problem setting to a constraint satisfaction problem which can be solved with standard solvers (which of course might have exponential runtime).
> Indeed, we introduce variables $t_{kn}$ for $k\in [K]$ and $n\in [N]$ which
> are 1 if token $n$ is in cluster $k$ and 0 otherwise. Then we need the condition
> $t_{1n}\lor t_{2n}\lor \ldots\lor t_{kn}$ to encode that
> every variable is assigned to at least one cluster. In addition, we consider variables $s_{\boldsymbol{k}}=g(\boldsymbol{k})$
> that encode whether $g(\boldsymbol{k})$ is 0 or 1 (extensions to general values are possible).
> We add for every datapoint $(\boldsymbol{n}^s,f(\boldsymbol{n}^s))$ and every $\boldsymbol{k}$ the constraint
> $$
> \lnot t\_{\boldsymbol{k}\_1\boldsymbol{n}\^s\_1}\lor
> \ldots\lor \lnot t\_{\boldsymbol{k}\_I\boldsymbol{n}\^s\_I}\lor (\lnot) s\_{\boldsymbol{k}}
> $$
> where the last term is negated if $f(\boldsymbol{n}^s)=0$ and not negated if $f(\boldsymbol{n}^s)=1$.
> Note that if $\boldsymbol{n}^s$ is in cluster $\boldsymbol{k}$ (i.e., $\Pi(\boldsymbol{n}^s)=\boldsymbol{k}$)
>  then the first part evaluates to false.
>  Thus, the condition ensures that if $\boldsymbol{n}^s$ is in cluster $\boldsymbol{k}$
>   the cluster must have the prescribed value
> $s_{\boldsymbol{k}}=g(\boldsymbol{k})=f(\boldsymbol{n}^s)$ for the clause to be true.
>
> Any satisfying assignment of the $\land$ of all these conditions gives rise to a partition induced by a
> $\Pi$ (encoded by the $t_{kn}$) and $g$ (encoded by $s_{\boldsymbol{k}}$) while proof of non-existence shows that no such partition exists.
> So far, we did not ensure that each token is assigned only to a single cluster
> but this can be achieved in postprocessing or by adding additional constraints
> (such as $\lnot t\_{k\_1n}\lor \lnot t\_{k\_1n}$).

---

> > ### Comment · Reviewer_8unk · 2024-08-10
> >
> > Thanks for the response. I will keep my original score.

---

### Official Review · Reviewer_NSnU · 2024-07-12

**Soundness:** 4
**Presentation:** 3
**Contribution:** 3
**Rating:** 7
**Confidence:** 3

**Summary:**

The paper proposes a new learning problem: learning the partitions of tokens given sample sequences of tokens. The authors first study the problem from an information-theoretical perspective, where $\tilde{O}(N)$ samples are sufficient to recover the partition for an alphabet of $N$ tokens. Then, they investigate the gradient dynamics of token embeddings and show a sample complexity $\tilde{O}(N^2)$.

**Strengths:**

-  The problem formulation is novel and clear. Though it simplifies many empirical settings, it shares some similarities with them and can serve as a starting point for theoretical analysis.
- The assumptions and main results are clearly stated. I appreciate the explanations provided after each assumption and result. While the notations, including several shorthands, are somewhat complex, they are generally easy for readers to follow. I also appreciate the efforts to condense the main results and proof ideas in the main text.
- I appreciate the inclusion of several hard case examples (including Theorem 3 and Example 1) that illustrate the hardness of the problems.

**Weaknesses:**

- I appreciate the solid proof provided by the authors. However, it is not technically novel to the community. In Section 3, the paper assumes that the sizes of clusters are almost even, under which it can be expected that the sample complexity bound can be improved from $O(N^I)$ to $\tilde{O}(K^IN)$. In Section 4, most of the analysis follows the usual path in analyzing dynamics. If there is more technical novelty, I would suggest emphasizing it explicitly.

-  The problem setting is a bit restricted. It could be interesting to explore extensions such as:

1) Allowing $f$ to be a function that depends not only on clusters but also on token embedding values, that is $f(n_1,\cdots,n_I)=f(n_1,\cdots,n_I,k_1,\cdots,k_l)=g_{\Pi}(n_1,\cdots,n_I)$. An example is that when $n_1\in k_1,n_2\in k_0$, $f(n_1,n_2)=n_1$.

2) Introducing some dependency between tokens and potentially allowing one item to belong to several clusters.


Minor Nits:
- Page 2 line 74. “such that”?
- Page 3, Line 117. Duplicates of “such as” and “e.g.,”
- Page 4, line 135 “need of the order”
- Page 5, equation (7), it should be $||u||$? u should be a vector.

**Questions:**

1. Page 4, Line 145, what is $N_0$?
2. Page 5 in the section on analysis of gradient descent. Firstly, I do not see where you use $\bar{v}(i)$ after defining it on Line 187. Secondly, what is $u$? I do not find its definition anywhere. One guess is that $u$ is essentially $v(n)$ from the context that follows. If so, I do not understand why scaling by $N/S$ is reasonable. Another guess is that $u$ is $\bar{v}(i)$, then it seems odd when you discuss ‘samples’ because $\bar{v}(i)$ is created as the concatenation across all possible N choices in the $i$-th slot, which essentially means having all possible samples at once.

**Limitations:**

The authors have adequately addressed the limitations.

---

> ### Author Rebuttal · Authors · 2024-08-06
>
> We thank the reviewer for their insightful review.
> Let us first answer the reviewer's questions and then briefly comment on the criticism raised in the review.
>
> > What is $N_0$?
>
> - We denote by $N\_0(I,K,\eta)$ a constant depending on $I$, $K$, and $\eta$, i.e., we will change the statement to 'Then there is a constant $N\_0(I,K,\eta)$ such that for all $N\geq N\_0$...'. In other words, the statement only holds for a sufficiently large number of tokens. Note that a more general statement of Theorem 1 can be found in Theorem 6 in Appendix B where this restriction is not required, and instead the condition on the sample size $S$ is that it needs to be larger than a maximum of four terms. For simplicity, we only kept the dominating term for large $N$ in Theorem 1 because this is our main setting of interest.
>
> > ... what is $\boldsymbol{u}$?...
>
> - We apologize, there is a typo (overlooked a missing macro when changing the notation) in Eq. (7) and (8): $\boldsymbol{u}$ should be $\bar{\boldsymbol{v}}$, i.e., the concatenation of all token embeddings (for all $n$ and all $i$). Note that the loss function $\hat{\mathcal{L}}$ is defined in Eq. (6) and indeed depends on all token embeddings.
> However, for one specific datapoint $\boldsymbol{n}^s$ the value of $f(\boldsymbol{n}^s)$
>  depends only on
> $v(i,n)$ if $\boldsymbol{n}^s_i=n$. This happens with probability $1/N$.
> So we will typically find that $S/N$ terms of the $S$ terms in Eq. (6) depend on the token embedding $v(i,n)$ for any $i$ and $n$. So we expect the gradient with respect to $v(i,n)$
> of $\hat{\mathcal{L}}$ to be of order $S/N$. Thus, it is natural to reweight by $N/S$ to make it of order 1.
>
> > Technical novelty
>
> - It is true that the proof itself contains no entirely new technical ingredients, however,
> this does not necessary mean that it is simple, and the proof required us to
> combine several different techniques, and it was still quite non-trivial for us to find the right decompositions and control all terms.
>
> > Problem setting is a bit restricted
>
> - As every model, our setting is a simplification of real-world settings, and we feel that the setting is already quite complex.
> Nevertheless, the suggested extensions seem very reasonable and interesting.
> Note that it is not directly clear what the 'right' representation would be for gradient descent-based algorithms in the considered extension. We will add your suggestions to the conclusion.
>
> As short remarks about the Minor Nits:
>
> - We will clarify that we use $|u|$ to denote the norm of a vector and $\lVert \cdot\rVert$ for matrix norms.
>
> - We thank the reviewer for pointing out the typos that we fixed in the current version of the manuscript.
>
> We hope that this addresses the reviewers' concerns, and we are happy to address any further questions or concerns and welcome additional feedback.

---

> > ### Comment · Reviewer_NSnU · 2024-08-07
> > **After Rebuttal**
> >
> > Thank you for the reply. I will keep my score.

---

### Official Review · Reviewer_Rx9k · 2024-07-17

**Soundness:** 3
**Presentation:** 3
**Contribution:** 3
**Rating:** 6
**Confidence:** 4

**Summary:**

This paper defines a learning problem for functions that depend only on a set of unknown partitions of the data. It establishes sample complexity, computational hardness and guarantees for GD-based algorithms under additional assumptions.

**Strengths:**

The model is simple and clean, the presentation is clear without embellishments. The progression of guarantees is natural and well-motivated.

**Weaknesses:**

The assumptions required for proving GD are many and some appear to be strong and unverifiable (?).

**Questions:**

Is it possible to (approximately) verify whether a given data set satisfies the assumptions you need to have the GD guarantee?

**Limitations:**

Yes.

---

> ### Author Rebuttal · Authors · 2024-08-06
>
> We thank the reviewer for their positive review.
>
> Regarding your question about the testability of the assumptions:
> The assumptions on the model used for learning are (in principle) testable.
> For the dataset, we can test whether samples follow a uniform distribution.
> What is difficult to test is whether a partition induced by $\Pi$ (such that $f=g\circ \Pi$) exists (see Theorem 3). It is also not possible to test whether $\Pi$ is balanced.
> Note, however, that if any algorithm such as gradient descent indeed reveals the partition induced by $\Pi$
> then we can verify post-hoc whether the training set satisfies the structural assumptions.
> Finally, we emphasize that while the criticism of lacking testability is valid, it applies to most theory results.
>
> We hope that this addresses the reviewers' concerns and that they consider improving their score if they find it satisfactory. We are happy to address any further questions or concerns and welcome additional feedback.

---

> > ### Comment · Reviewer_Rx9k · 2024-08-12
> >
> > I acknowledge the rebuttal. The authors might consider proving that their required conditions hold (unconditionally) for interesting families of inputs defined by other means. There is no need to respond to this comment.

---

### Author Rebuttal · Authors · 2024-08-06

We thank all reviewers for their careful and insightful reviews.

All reviewers agree that the problem formulation is interesting and relevant ('simple and clean' R. `Rx9k`, 'novel and clear' R. `NSnU`, 'important insights' R `8unk`, and
'interesting and novel' R. `w8Q8`), the results are acknowledged ('progression [...] natural and well-motivated' R. `Rx9k`, 'solid theoretical study' R. `8unk`) and
the presentation of the results is appreciated ('generally easy for readers to follow' R. `NSnU`, 'the authors interpreted and commented all their results in a clear manner' R. `8unk`).
There are some complaints about the language (R. `w8Q8`) and the lack of a general  algorithm
and experiments (R. `8unk`) that are addressed in the individual responses below.

---

### Decision · Program_Chairs · 2024-09-25

**Decision:**

Accept (poster)

**Comment:**

The paper proposes the problem of learning partitions of tokens from sequences of tokens. The manuscript establishes sample complexity, NP completeness and sample complexity for gradient descent, under some additional technical assumptions.

Some concerns were raised during the rebuttal phase, including for instance: restrictiveness of the assumption, lack of algorithm and experimentation. After this, the reviewers were still supportive of accepting the paper. In addition, I consider this submission a pure learning theoretic paper, and the reviewers' criticisms should be interpreted from that viewpoint (in addition to the supportive scores of the reviewers.)

For a camera-ready version, please take into the account the comments from all reviewers.